# Ferroptosis induction in host rice by endophyte OsiSh-2 is necessary for mutualism and disease resistance in symbiosis

Xianqiu Xiong[1,2], Jing Zeng[1,2], Qing Ning[1], Heqin Liu[1], Zhigang Bu[1], Xuan Zhang[1], Jiarui Zeng[1], Rui Zhuo [1], Kunpeng Cui [1], Ziwei Qin[1], Yan Gao [1]✉, Xuanming Liu [1]✉ & Yonghua Zhu [1]✉

Ferroptosis is an iron-dependent cell death that was discovered recently. For beneficial microbes to establish mutualistic relationships with hosts, precisely controlled cell death in plant cells is necessary. However, whether ferroptosis is involved in the endophyte–plant system is poorly understood. Here, we reported that endophytic *Streptomyces hygroscopicus* OsiSh-2, which established a sophisticated and beneficial interaction with host rice plants, caused ferroptotic cell death in rice characterized by ferroptosis- and immune-related markers. Treatments with ferroptosis inhibitors and inducers, different doses of OsiSh-2, and the siderophore synthesis-deficient mutant Δ*cchH* revealed that only moderate ferroptosis induced by endophytes is essential for the establishment of an optimal symbiont to enhance plant growth. Additionally, ferroptosis involved in a defence-primed state in rice, which contributed to improved resistance against rice blast disease. Overall, our study provides new insights into the mechanisms of endophyte–plant interactions mediated by ferroptosis and suggests new directions for crop yield promotion.

Plants provide a rich source of nutrients for a plethora of surrounding microbes, which in turn contribute pathogenic, beneficial, or neutral effects on plant growth and fitness. However, regardless of the kind of microbes involved, plants first recognize them as 'nonself' bodies or their components as nonplant-derived 'self' molecules, which can act as indicators of danger and then activate the plant immune system[1]. For example, rhizospheric *Bacillus subtilis*[2] and endophytic *Trichoderma*[3], chitin and flagellin 22 from pathogens[4], phosphopentomutase[5] and lipopolysaccharide[6] from beneficial microbes can induce a series of immune responses, including reactive oxygen species (ROS) burst, deposition of callose, activation of defence-related genes, and hypersensitive response (HR)-associated cell death in host plants[1,7]. Correspondingly, different microbes respond to and regulate the host defence system by various mechanisms depending on their lifestyle, i.e., invading as pathogens or colonizing as probiotics in plant tissues[8]. Cell death is a typical plant immune response to microbes[9]. Necrotrophic pathogens such as *Parastagonospora nodorum*[10] and *Zymoseptoria tritici*[11] can exploit host-mediated cell death to complete their life cycle and infect plants. The beneficial arbuscular mycorrhizal fungus *Funneliformis mosseae*[12] and rhizobacteria *Bacillus* spp[13]. can precisely regulate plant cell death to establish mutualistic symbiosis with the hosts *Cichorium intybus* and chili pepper. As important functions of plant beneficial microbes, endophytes reside symbiotically within plants and can better interact

[1]Hunan Province Key Laboratory of Plant Functional Genomics and Developmental Regulation, College of Biology, Hunan University, Changsha 410082 Hunan, PR China. [2]These authors contributed equally: Xianqiu Xiong, Jing Zeng. ✉e-mail: blessedgy@hnu.edu.cn; xml05@hnu.edu.cn; yonghuaz@outlook.com

with the host than microbes derived from soil. However, knowledge about the interactions between endophytes and the plant immune system is still sparse.

Recently, a newly recognized form of cell death named ferroptosis has been the focus of growing interest[14,15]. Distinct from other forms of cell death, such as apoptosis, necrosis, and autophagy, ferroptosis is characterized by its nonapoptotic and iron-dependent lipid hydroperoxide production[16,17]. Ferroptosis is conserved among different species: it exhibits nearly all the main biochemical and morphological hallmarks, including the accumulation of iron, ROS, and lipid peroxides, along with reduced levels of reduced glutathione (GSH)[18] and oxidized glutathione (GSSG)[19]. However, compared to the well-described roles of the ferroptosis in cancer[20], inflammation[21], and the nervous system in mammals[22], the role and regulatory mechanism of ferroptosis in plant physiological activity are still poorly understood. It was recently found that ferroptosis participates in the plant interactions with environmental factors. Heat shock treatment can trigger ferroptosis-like cell death in *Arabidopsis thaliana* root cells[18]. Disease-resistant rice can cause active ferroptosis to disrupt *Magnaporthe oryzae* infection[19]. However, the role of ferroptosis in host plants when responding to beneficial microbes, including endophytes, has not been reported.

The occurrence of ferroptosis is tightly regulated by iron and ROS accumulation. Iron is an essential micronutrient for development and growth in nearly all living organisms, but a high level of $Fe^{2+}$ can hasten the production of ROS through the Fenton reaction and leads to peroxidation of lipids, initiating cell death[23,24]. Thus, the iron chelator deferoxamine (DFO), a siderophore with a high $Fe^{3+}$ affinity constant, can inhibit ferroptosis through the prevention of iron accumulation[25]. Erastin can specifically induce ferroptosis by inhibiting the peroxide removal system, which limits ROS production in cells[26]. In contrast, ROS scavengers such as ferrostatin-1 (Fer-1) and diphenyleneiodonium (DPI) can inhibit ferroptosis. Fer-1 is a synthetic antioxidant that reduces ROS deposition through a reduction mechanism[27]. DPI preferentially inhibits the plasma membrane NADPH oxidase activity required to generate extracellular ROS in plant and mammalian cells[18]. In addition, depletion of GSH and disruption of glutathione peroxidase 4 (GPX4), two prominent components of the cellular antioxidant system that protect cells from ferroptosis-induced oxidative damage, are suggested to initiate ferroptosis[15]. The discovery and use of inducers and inhibitors of ferroptosis are of great value for elucidating the mechanism of ferroptosis-associated activity.

In our previous study, we developed a rice endophyte, *Streptomyces hygroscopicus* OsiSh-2 (hereafter OsiSh-2), which could form an optimal symbiotic relationship with host rice. OsiSh-2 exhibited remarkable antagonistic activity against rice blast disease and maintained high yields under laboratory and field conditions[28]. Further studies revealed that OsiSh-2 could effectively inhibit the growth of the pathogen *M. oryzae* by secreting siderophores for competing iron[29] and improve host disease resistance by triggering a faster and more robust activation of the immune response, i.e., defence priming, in rice[28].

In this work, we elucidated another mechanism of OsiSh-2-induced disease resistance. Detecting a series of ferroptosis- and immune-related markers in rice revealed the occurrence of ferroptosis in the OsiSh-2-rice symbiont. Then, by using different concentrations of OsiSh-2 and constructing the siderophore-knockout strain Δ*cchH*, we revealed that the appropriate level of ferroptosis regulated by OsiSh-2 is essential for establishing mutual symbiosis between OsiSh-2 and rice, which maintained a defence-primed state in the symbiont and thus enhanced disease resistance against *M. oryzae*. Our results confirm a novel endophyte–plant interaction mode related to ferroptosis induction and regulation in plants.

## Results

### Endophyte OsiSh-2 triggered a ferroptotic cell death response in rice

When OsiSh-2 ($10^8$ spores mL$^{-1}$) was sprayed evenly onto the leaves of the rice seedlings (E+), a phenotype of marked cell-death occurrence, i.e., the emergence of lesion mimics (LMs), which appear as reddish-brown spot-like lesions[30], was observed on rice leaves at 24 h post treatment (hpt) (Fig. 1a). The trypan blue staining assay confirmed the occurrence of cell death in E+ rice leaves (Fig. 1b). In addition, as the magnified images show in Supplementary Fig. 1a, the blue dots, which indicated dead plant cells, were mainly found in and around the stomata and adjacent mesophyll cells. To identify whether the formation of LMs is related to ferroptotic cell death, we measured the iron ($Fe^{3+}$) and ROS levels in the rice leaves by Prussian blue and fluorescence staining, respectively[15]. Both accumulation of $Fe^{3+}$ (blue colour) (Fig. 1c, d) and an ROS burst (green fluorescence) (Fig. 1e, f) were observed at 6 hpt in the stomata of the E+ rice leaves, indicating the occurrence of ferroptosis. However, none of the above phenotypes were observed in OsiSh-2 nontreated (E-) rice. In addition, the leaf tissues of E+ and E- rice at 6 hpt were subjected to propidium iodide (PI) and terminal deoxynucleotidyl transferase-mediated dUTP-biotin nick-end labelling (TUNEL) staining assays. As shown in Supplementary Fig. 1b, the red signals displayed in the stomata (St) of E+ rice leaves, while the stomata of E- rice leaves were not stained. Meanwhile, the green TUNEL signals were stronger in the stomata of E+ rice leaves than those in the stomata of E- rice leaves. In the merged image of PI and TUNEL signals, we observed red PI signal overlapped the green TUNEL signal at the stomata (yellow spot, white arrows), indicating that OsiSh-2 triggered the DNA damage and stomatal cell death in E+ rice stomata. Moreover, as the magnified Prussian staining images show in Supplementary Fig. 1c, the appearance of brown spots, which generally represent the cellular senescence or death, was detected in and around the stomata at 24 hpt in the E+ rice. The fact that the $Fe^{3+}$ and ROS accumulation observed at the early stage and the subsequently detected dead cells were in the same location in the rice leaves, i.e., in and around the stomata and adjacent mesophyll cells, indicates that the $Fe^{3+}$- and ROS-dependent cell death triggered by OsiSh-2 was ferroptosis. Meanwhile, the expression of key genes involved in the rice immune system, such as lipoxygenase 2 (*OsLOX2*) and probenazole-induced gene (*OsPBZ1*), was obviously induced from 4 hpt in the E+ rice (Fig. 1g, h), indicating activation of the immune response. The transcription of rice genes involved in Fe transport, including iron-related bHLH transcription factor 2 (*OsIRO2*) and its downstream gene yellow stripe-like 15 (*OsYSL15*), a $Fe^{3+}$-phytosiderophore transporter-encoding gene, was also detected. In E+ rice, expression of *OsIRO2* and *OsYSL15* was observed as early as 0.5 hpt, indicating iron mobilization in response to OsiSh-2 treatment. Until $Fe^{3+}$ accumulated in the stomata of the E+ rice at 6 hpt (Fig. 1c), the expression of these two genes correspondingly decreased (Fig. 1i, j). We also evaluated the key biochemical indexes for ferroptosis after OsiSh-2 treatment, including the levels of GSH and total GSH (GSH+GSSG). Compared with those in the E- rice, the levels of GSH and GSH+GSSG decreased by 54.24% and 50.52%, respectively, in the E+ rice (Fig. 1k, l). The GSH depletion suggested an induction of ferroptosis by OsiSh-2 in the rice plants.

To further confirm that the OsiSh-2-induced immune response in E+ rice is indeed ferroptosis, E+ rice plants were treated with the specific ferroptosis inducer erastin (E+ Erastin+) and inhibitor Fer-1 (E+ Fer-1+). Obviously, the former exhibited more LM formation, while the latter showed a reduced number of LMs compared with those exhibited by the water-treated E+ mock rice (Fig. 2a). A similar tendency was found for the accumulation of $Fe^{3+}$ (Fig. 2b, c, f, g) and ROS (Fig. 2d, e, h, i). In addition, compared with that in the E+ mock rice, the contents of

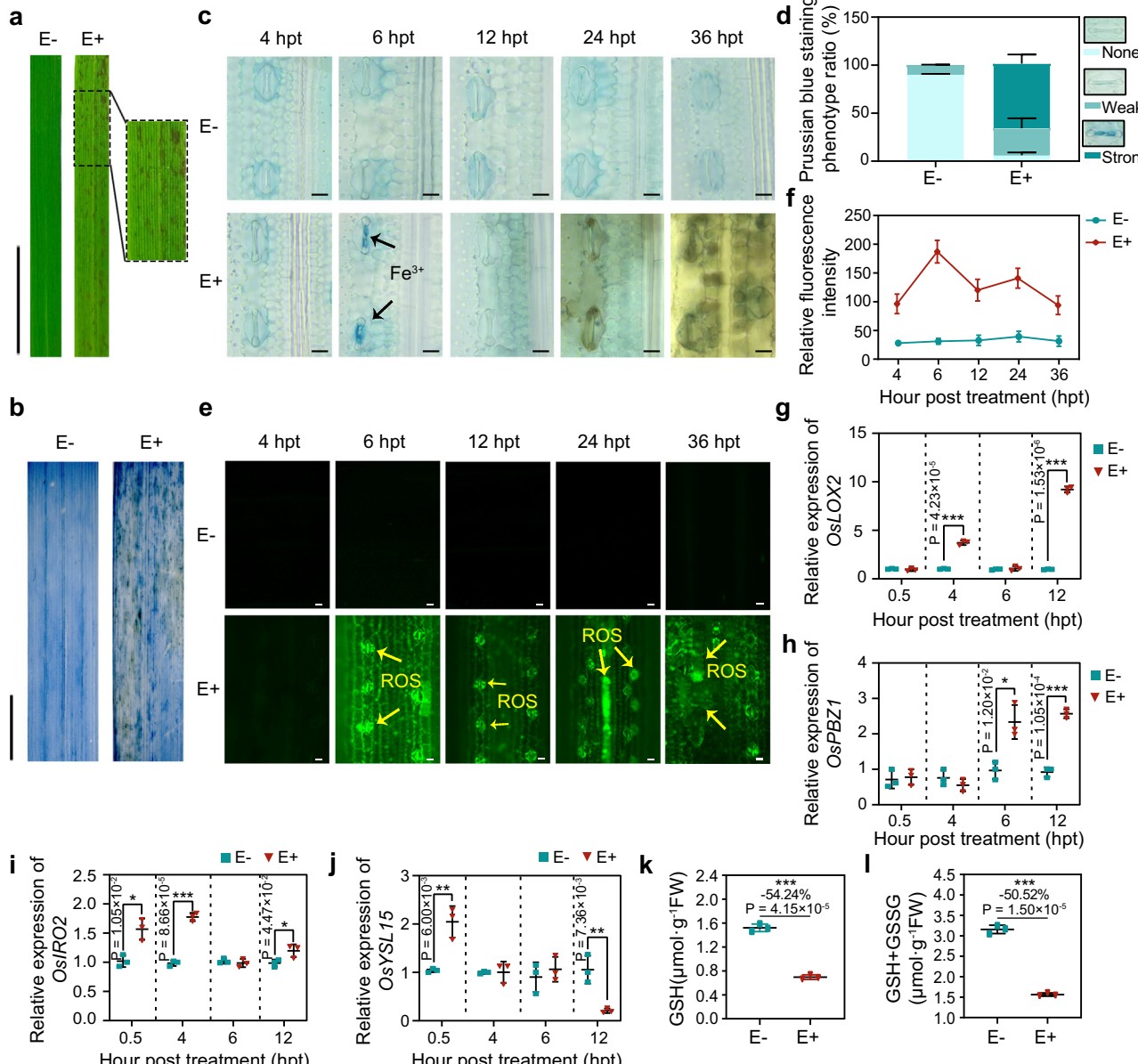

**Fig. 1 | *Streptomyces hygroscopicus* OsiSh-2 triggers an ROS- and Fe³⁺-accumulated cell death response in rice leaves. a** Images of the lesion mimic (LM) phenotype and the magnified part of rice leaves treated with OsiSh-2 at 24 h post treatment (hpt). Untreated rice leaves were used as controls. E-: rice whose leaves were sprayed with water; E+: rice whose leaves were sprayed with an OsiSh-2 spore suspension ($10^8$ spores mL$^{-1}$). Scale bars: 1 cm. **b** Trypan blue staining (dark blue colour) shows cell death in E+ rice leaves at 24 hpt. Scale bars: 1 cm. **c** Prussian blue staining (blue colour) shows the accumulation of Fe³⁺ in E+ rice leaves at 6 hpt. Scale bars: 10 μm. The black arrows indicate Fe³⁺ accumulation. **d** Relative Fe³⁺ accumulation is expressed as the Prussian blue staining phenotype ratio (%). Ratios indicate the proportions of designated staining phenotypes. Error bars indicate the mean ± SDs (n=3 for each). **e** CM-H₂DCFDA staining shows the accumulation of reactive oxygen species (ROS, green fluorescence) in E+ rice leaves at 6 hpt. Scale bars: 10 μm. The yellow arrows indicate ROS bursts. **f** Relative ROS accumulation is

expressed as the relative fluorescence intensity of CM-H₂DCFDA-stained rice cells. The relative fluorescence intensity of ROS was calculated via ImageJ. The data shown indicate the means±SDs (n = 20 for each). The transcript levels of key genes involved in the rice immune system (*OsLOX2* and *OsPBZ1*) (**g**, **h**) and Fe transport (*OsIRO2* and *OsYSL15*) (**i**, **j**) were determined by RT–qPCR at the indicated hour post treatment in E- and E+ rice. Error bars indicate the means± SDs (n = 3). The contents of GSH (**k**) and total GSH (GSH+GSSG) (**l**) at 12 hpt in E- and E+ rice were determined. Others are as in (**g**). The images shown are representative of the rice leaf samples in different treatments. The bars with asterisks are significantly different as determined by the Tukey–Kramer test (*$P < 0.05$; **$P < 0.01$; ***$P < 0.001$). Percentage changes followed by "−" above the bars were calculated via the following formula: percent change = [(value of E+ rice) - (value of E- rice)]/(value of E- rice) × 100%. All experiments were repeated independently three times with similar results. Source data are provided as the Source Data file.

GSH and total GSH (GSH+GSSG) in E+ Erastin+ rice were significantly decreased by at least 23.96% (Fig. 2j, k). Notably, although erastin is an inducer of ferroptosis, this small molecule in 10 μM did not trigger iron-dependent cell death alone in rice (Fig. 2a). On the other hand, the content of MDA, a typical lipid peroxidation marker[19], was reduced by 24.81% in E+ Fer-1+ rice compared with E+ mock rice (Fig. 2l). In addition, other ferroptosis inhibitors (DFO and DPI) and inducers (FeCl₃)

also exhibited suppression or induction effects on the accumulation of Fe³⁺ and ROS in the E+ rice, respectively (Supplementary Fig. 2).

We also added a spore suspension of OsiSh-2 (E+) to rice suspension cells at a final concentration of $10^8$ spores mL$^{-1}$ and then monitored cell death by PI staining and Fe³⁺ accumulation by Prussian blue staining. In E+ rice suspension cells, the number of dead cells increased by 213.64% compared with that of untreated (E-) rice

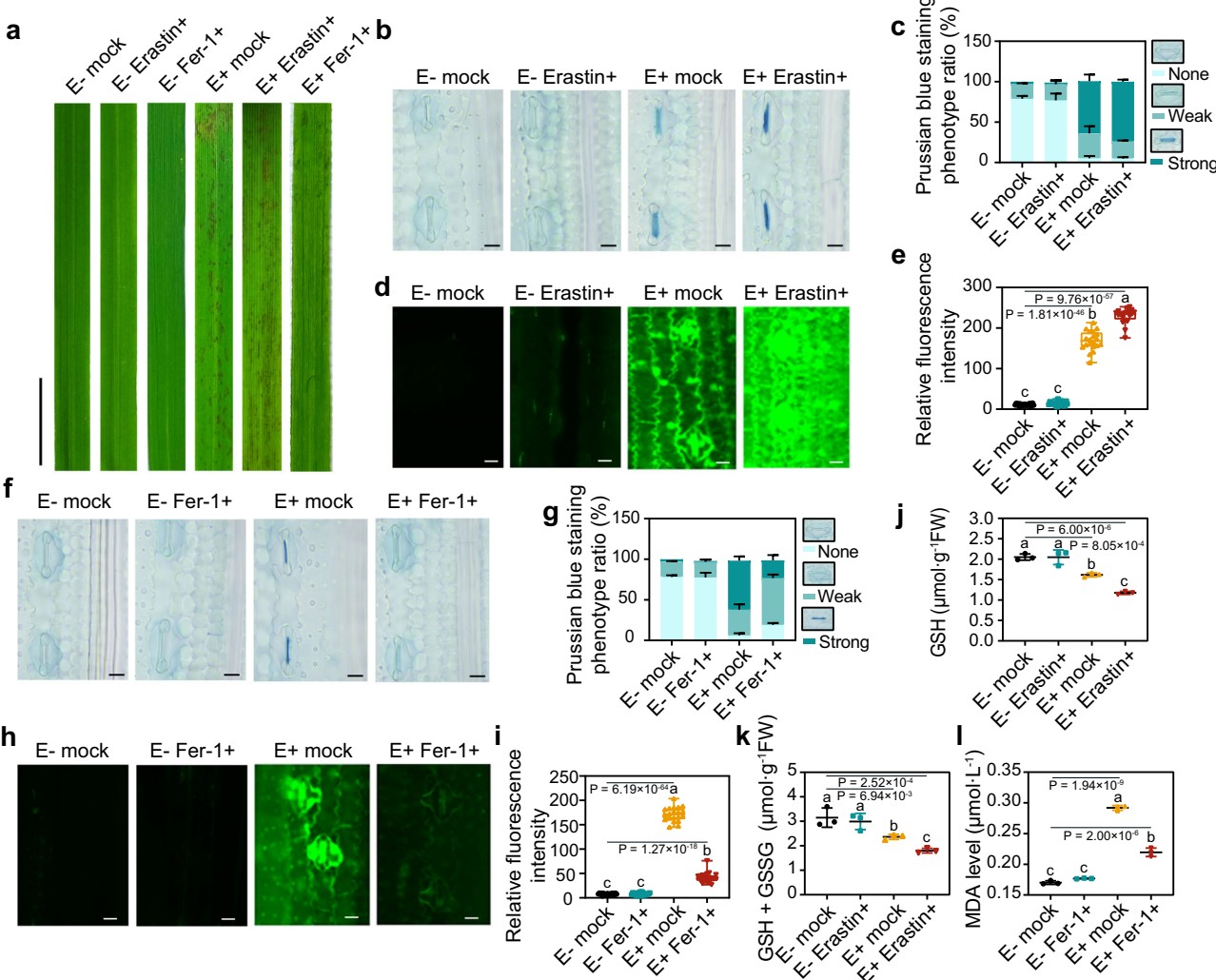

**Fig. 2 | The ferroptosis inducer erastin and inhibitor ferrostatin-1 (Fer-1) influence the ROS- and Fe$^{3+}$-accumulated cell death in OsiSh-2-treated rice leaves. a** Image of LMs of E- and E+ rice leaves treated with water (mock), erastin (Erastin+), and Fer-1 (Fer-1+) at 24 hpt. Scale bars: 1 cm. Prussian blue staining (**b**) and its phenotype ratio (%) (**c**) shows the accumulation of Fe$^{3+}$ in E- and E+ rice leaves with erastin (Erastin+) or not (mock) at 6 hpt. Scale bars: 10 μm. The Prussian blue staining phenotype ratios (%) indicate the proportions of designated staining phenotypes. Error bars indicate the mean±SDs (*n* = 3 for each). CM-H$_2$DCFDA staining (**d**) and relative fluorescence intensity (**e**) show the accumulation of ROS in rice samples same as **b**. Scale bars: 10 μm. The relative fluorescence intensity of ROS was calculated via ImageJ. Experimental repeats are displayed as box plots with individual data points. The error bars represent maximum and minimum values. Middle horizontal bars of boxplots represent the median, and the bottom and top represent the 25th and 75th percentiles (*n* = 20 for each). Prussian blue staining (**f**) and its phenotype ratio (%) (**g**) shows the accumulation of Fe$^{3+}$ in E- and E+ rice leaves with Fer-1 (Fer-1+) or not (mock) at 6 hpt. Scale bars: 10 μm. Others are as in (**c**). CM-H$_2$DCFDA staining (**h**) and relative fluorescence intensity (**i**) shows the accumulation of ROS in rice samples same as (**f**). Scale bars: 10 μm. Others are as in (**e**). The contents of GSH (**j**) and total GSH (GSH+GSSG) (**k**) in E+ and E- rice leaves with Erastin+ or mock at 12 hpt were determined. Error bars indicate the mean±SDs (*n* = 3). **l** The lipid peroxidation levels marked by the MDA content of E- and E+ rice leaves with Fer-1+ or mock at 12 hpt were measured. Others are as in (**j**). The bars with different letters are significantly different (ANOVA, *P* < 0.05) according to Duncan's multiple-range test. All experiments were repeated independently three times with similar results. Source data are provided as the Source Data file.

suspension cells at 6 hpt (Supplementary Fig. 3a, b). Meanwhile, Fe$^{3+}$ accumulation was observed only in the E+ rice suspension cells (Supplementary Fig. 3c). Moreover, the addition of the ferroptosis inducer erastin significantly increased the number of dead cells (45.32%) in E+ Erastin+ rice suspension cells, while the ferroptosis inhibitor Fer-1 alleviated cell death by 41.83% in E+ Fer-1+ rice suspension cells compared with E+ mock rice suspension cells (Supplementary Fig. 3a, b). To rule out the possibility that the observed phenomenon was due to the action of the compound erastin or Fer-1 alone in rice cells and OsiSh-2, we supplemented the treatments of erastin and Fer-1 in E- rice or culture medium of OsiSh-2 as controls. However, erastin and Fer-1 had almost no effect on rice cell viability (Supplementary Fig. 3a, b) or OsiSh-2 growth (Supplementary Fig. 3d). Overall, we concluded that OsiSh-2 can induce ferroptotic cell death in host rice.

## OsiSh-2 colonization and induced disease defence are involved in ferroptosis

Our previous study indicated that OsiSh-2 could overcome the plant immune defence for colonization and then form a mutual symbiosis with host rice. We observed that OsiSh-2 initially induced LM formation in rice leaves. However, the LMs did not continue developing or spreading to the new rice leaves, and the growth of the E+ rice gradually became better than that of the E-rice[28]. Therefore, what is the relationship between OsiSh-2 colonization and the occurrence of ferroptosis in host rice? To answer this question, we first monitored the endophytic biomass of OsiSh-2 in rice in real time. As shown in Fig. 3a, the biomass of OsiSh-2 in E+ rice rapidly increased from 2−4 hpt, indicating initial endophytic colonization. However, a sharp decrease in OsiSh-2 biomass was then observed at 6−12 hpt, suggesting that the

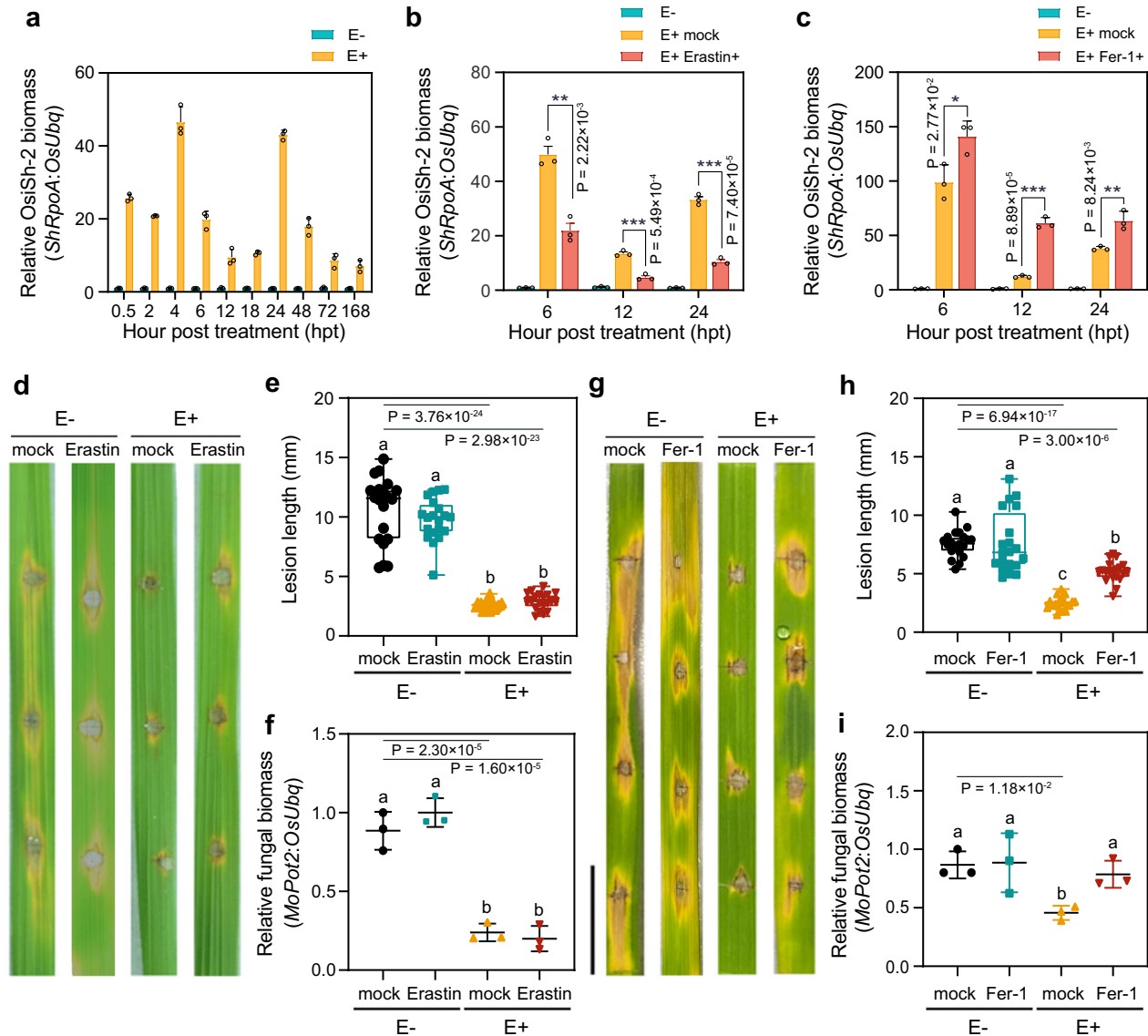

**Fig. 3 | Ferroptosis contributes to endophytic colonization and induction of disease resistance in the symbiont.** The relative biomass of OsiSh-2 was calculated via the threshold cycle value (Ct) of OsiSh-2 *ShRpoA* DNA versus the Ct of rice *OsUbq* DNA by qRT–PCR at the indicated hours post treatment in E- and E+ rice (**a**); in E-, E+ mock, and E+ Erastin+ rice (**b**); and in E-, E+ mock, E+ Fer–1+ rice (**c**). Error bars indicate the mean ± SDs (*n* = 3). Images of blast lesions on detached rice leaf segments at 5 days post-infection (dpi) of *Magnaporthe oryzae* (M+) at a concentration of 10⁵ conidia mL⁻¹ in E- M+ and E+ M+ rice leaves with Erastin+ or mock (**d**); in E- M+ and E+ M+ rice leaves with Fer-1+ or mock (**g**). Scale bars, 1.0 cm. The lesion length of detached rice leaves was measured at 5 days post-inoculation (dpi) via ImageJ in E- M+ and E+ M+ rice leaves with Erastin+ or mock (**e**); in E- M+ and E+ M+ rice leaves with Fer-1+ or mock (**h**). Experimental repeats are displayed as box plots with individual data points. The error bars represent maximum and minimum values. Middle horizontal bars of boxplots represent the median, and the bottom and top represent the 25th and 75th percentiles (*n* = 20 for each). The relative fungal biomass of *M. oryzae* was calculated via the Ct of *M. oryzae MoPot2* DNA versus the Ct of rice *OsUbq* DNA by qRT–PCR at 5 days post infection of the rice leaves in E- M+ and E+ M+ rice leaves with Erastin+ or mock (**f**); in E- M+ and E+ M+ rice leaves with Fer-1+ or mock (**i**). Others are as in (**a**). The bars with asterisks are significantly different as determined by the Tukey–Kramer test (*\**P* < 0.05; \*\**P* < 0.01; \*\*\**P* < 0.001). The bars with different letters are significantly different (ANOVA, *P* < 0.05) according to Duncan's multiple-range test. All experiments were repeated independently three times with similar results. Source data are provided as the Source Data file.

colonization of OsiSh-2 was stunted. Considering the above observation that ferroptosis was activated at 6 hpt after OsiSh-2 treatment (Fig. 1), the inhibition of endophytic colonization from 6 hpt suggested that the occurrence of ferroptosis might be responsible for the limitation of excessive endophytic proliferation. Treatment with the ferroptosis inducer erastin and inhibitor Fer-1 confirmed this hypothesis, as the addition of erastin enhanced the inhibitory effect on OsiSh-2 growth (Fig. 3b), while Fer-1 promoted OsiSh-2 colonization (Fig. 3c). Thus, the colonization of OsiSh-2 in host rice is closely related to the ferroptosis process. Intriguingly, OsiSh-2 reaccumulated at 24 hpt in the E+ rice, indicating that OsiSh-2 hijacked the activity of the immune

system of the host rice and colonized successfully. Inevitably, another decrease was found at 48 hpt and relatively lower levels until 168 hpt in the E+ rice, indicating that the overproliferation of endophytes was well inhibited in the OsiSh-2-rice symbiont.

We then investigated whether ferroptosis plays a role in OsiSh-2-induced disease resistance in host rice. For this, we added erastin and Fer-1 to E+ rice and then infected those plants with *M. oryzae*, hereafter referred to as E+ Erastin+ M+ and E+ Fer-1+ M+ rice. Erastin treatment did not further enhance the disease resistance capability of E+ mock rice, as both the disease lesion length and the relative fungal biomass of E+ Erastin+ M+ rice was similar to those of E+ mock M+ rice,

suggesting that OsiSh-2-triggered ferroptosis was sufficient to induce disease resistance (Fig. 3d–f). However, Fer-1 treatment significantly reduced the disease control effect of OsiSh-2, as the disease lesion length and relative fungal biomass of E+ Fer-1+ M+ rice increased by 35.35% and 38.08%, respectively, compared with those of E+ mock+ M+ rice (Fig. 3g–i). These results suggested that an appropriate degree of ferroptosis is essential for improving the disease resistance of the OsiSh-2-rice symbiont.

To further confirm that the appropriate induction of ferroptosis is necessary for mutualistic symbiosis between OsiSh-2 and host rice, we treated rice plants with OsiSh-2 at three different concentrations ($10^7$, $10^8$, and $10^9$ spores mL$^{-1}$), hereafter named E+ ($10^7$), E+ ($10^8$), and E+ ($10^9$) rice, respectively. This is based on our observation that the beneficial effects of OsiSh-2 depended on the size of the inoculant population. The E+ ($10^8$) rice showed the best growth-promoting and disease-resistance effects. As shown in Fig. 4a, b, the plant height, root length, and weight of the E+ ($10^8$) rice were all significantly better than those of the E+ ($10^7$) and E+ ($10^9$) rice. A similar tendency was found in the disease resistance against *M. oryzae*, as the lesion length and disease index of the E+ ($10^8$) rice were the lowest among those in response to the three treatments (Fig. 4c–e). When the relative biomass of OsiSh-2 was monitored in real time in host rice, the biomass accumulation of OsiSh-2 was consistently the lowest in the E+ ($10^7$) rice, while it was dramatically increased in the E+ ($10^9$) rice (Fig. 4f). Notably, when further analysing the ferroptosis progress indicated by the accumulation of $Fe^{3+}$ and ROS, we found that only OsiSh-2 ($10^8$ spores mL$^{-1}$) could trigger the simultaneous accumulation of $Fe^{3+}$ (59.06% strong) and ROS (105 relative fluorescent intensity) at 6 hpt (Fig. 4g–j). However, at the same time, in both E+ ($10^7$ and $10^9$) rice, the accumulation of $Fe^{3+}$ (11.19% and 20.78% strong ones, respectively) and ROS (82 and 62 relative fluorescent intensity, respectively) were decreased. Consistently, when the above three concentrations of OsiSh-2 spores were added to the rice suspension cells, PI staining revealed that, the dead cells in E+ ($10^7$ and $10^9$ spores mL$^{-1}$) rice suspension cells were significantly decreased by 37.06% and 32.07%, respectively, at 6 hpt when compared with those of E+ ($10^8$ spores mL$^{-1}$) rice suspension cells (Supplementary Fig. 4). Taken together, these results demonstrated that a moderate occurrence of ferroptosis is crucial for host rice to allow the proper colonization of OsiSh-2, thus establishing an optimal symbiont with endophytic OsiSh-2 and enhancing disease resistance against *M. oryzae*.

## Siderophore deficiency of OsiSh-2 affected the ferroptosis occurrence

Our previous studies revealed that OsiSh-2 produced siderophores for iron uptake[29]. Because the *Streptomyces*-derived siderophore DFO is a commonly used inhibitor of ferroptosis, we proposed that OsiSh-2 might also use its siderophores to regulate the degree of ferroptosis in host rice. Genome mining revealed four clusters of genes involved in the biosynthesis of siderophores in the OsiSh-2 genome. The sequences of genes in the two clusters were 100% similar to those of the siderophores coelichelin (COE) and DFO. The expression of *ShCchH* and *ShDesD*, the core synthesis genes of siderophores COE and DFO, respectively, was induced under Fe deficiency (Fe-) and *M. oryzae* cocultivation conditions (Supplementary Fig. 5), suggesting that COE and DFO participated in the stress response of OsiSh-2. In E+ rice, the expression of both genes was induced at 6–12 hpt (Supplementary Fig. 5f), suggesting that OsiSh-2 colonization was accompanied by siderophore production.

Using double homologous recombination technology, we then constructed the *ShCchH* and *ShDesD* knockout mutant strains ΔcchH and ΔdesD, respectively. PCR analysis showed that the expected linker fragment (1292 bp) of the upstream and downstream arms of *ShCchH* was observed in genomic DNA samples isolated from the ΔcchH mutant, confirming that the *ShCchH* gene was deleted (Supplemental

Fig. 6a, b, Supplemental Table 1, Supplemental Text 1). In the traditional ISP2 culture medium, the growth rate and biomass of ΔcchH were slightly worse than those of the wild-type OsiSh-2 (WT) strain. On day eight, when the biomass of the two strains tended to be the same, the amount of siderophores of ΔcchH was 32.51–36.80% lower than that of OsiSh-2 (WT) (Supplementary Fig. 6c–e). To further determine that the siderophore-producing capability of ΔcchH was weakened, OsiSh-2 (WT) and ΔcchH strains were inoculated on chrome azurol S (CAS) agar plates, which are useful in the identification of siderophores. Consistent with the solution cultivation, the diameter of the orange zone in ΔcchH was reduced by 19.34% when compared with that of OsiSh-2 (WT) after 14 days of cultivation (Supplementary Fig. 6f, g). These results confirmed that the mutant strain ΔcchH was constructed successfully. However, the obtained ΔdesD mutants could not proliferate normally, indicating that the synthesis of the siderophore DFO is crucial for microbial survival.

We then sprayed spore suspensions of OsiSh-2 (WT) and ΔcchH onto rice leaves. Considering that it is difficult to detect the amount of OsiSh-2-produced siderophores in rice tissues, we monitored the relative expression levels of *ShCchH* and *ShDesD* in E+ (WT) and E+ (ΔcchH) rice in real time. In contrast to the strong induction of *ShCchH* and *ShDesD* expression at 6–12 hpt in E+ (WT) rice, *ShCchH* failed to activate expression, and the transcriptional levels of *ShDesD* were obviously decreased in the E+ (ΔcchH) rice (Supplementary Fig. 6h, i). These results suggested that the biosynthesis of COE was indeed decreased, which might influence the production of other siderophores, such as DFO. We also found that the biomass of ΔcchH in rice was much greater than that of OsiSh-2 (WT) at 4–12 h (Fig. 5a). Accordingly, a very weak accumulation of $Fe^{3+}$ around the stomata and an ROS burst in the veins at 4 hpt were observed in the E+ (ΔcchH) rice leaves (Fig. 5b, c, d, e). When we added the spore suspension of ΔcchH ($10^8$ spores mL$^{-1}$) to the rice suspension cells, the dead cells decreased by 34.90% compared with those of OsiSh-2 (WT)-supplemented rice suspension cells at 6 hpt (Supplementary Fig. 4). In addition, ΔcchH treatment led to more accumulation of GSH and total (GSH+GSSG) in rice compared with the E+ (WT) rice (Fig. 5f, g), indicating unexhausted GSH. These results suggested that the mutant strain ΔcchH triggered only a basal level of ferroptosis in host rice. Consistent with the lower degree of ferroptosis, the disease resistance capability of the E+ (ΔcchH) M+ rice was significantly reduced with a longer disease lesion length (13.71%) and more relative fungal biomass (40.04%) than those of the E+ (WT) M+ rice (Fig. 5h–j). In addition, the dual-culture assay for OsiSh-2/ΔcchH and *M. oryzae* showed that the fungal growth inhibition rate of ΔcchH was only 26.21% lower than that of OsiSh-2 (WT) (Supplementary Fig. 6j, k). Overall, these results indicated that siderophore synthesis deficiency of ΔcchH failed to trigger effective ferroptosis, leading to excessive endophytic colonization in host rice and decreased disease resistance against pathogens compared to those of OsiSh-2 (WT).

To determine the role of siderophores in disease resistance, we added the *Streptomyces*-derived siderophore DFO to rice plants treated with OsiSh-2. The addition of DFO obviously inhibited $Fe^{3+}$ and ROS accumulation in E+ (WT) rice (Supplementary Fig. 7a–d), and the disease resistance against *M. oryzae* was significantly reduced (Supplementary Fig. 7e–g). This is consistent with the fact that DFO serves as a ferroptosis inhibitor[19]. However, when DFO was applied in ΔcchH, the $Fe^{3+}$ and ROS accumulation was partially rescued at 6 hpt in E+ (ΔcchH) DFO+ rice, and the blast resistance effect was improved close to that of E+ (WT) mock rice (Supplementary Fig. 7). This result indicated that when the occurrence of ferroptosis was restricted in E+ (ΔcchH) cells, an appropriate amount of DFO could help induce ferroptosis, although the ferroptosis level could not be rescued to that in E+ (WT) cells. Notably, the single treatment of DFO did not influence the disease resistance of the rice plants, as both lesion length and relative fungal biomass in E- DFO+ M+ rice were similar to those in E- mock M+ rice

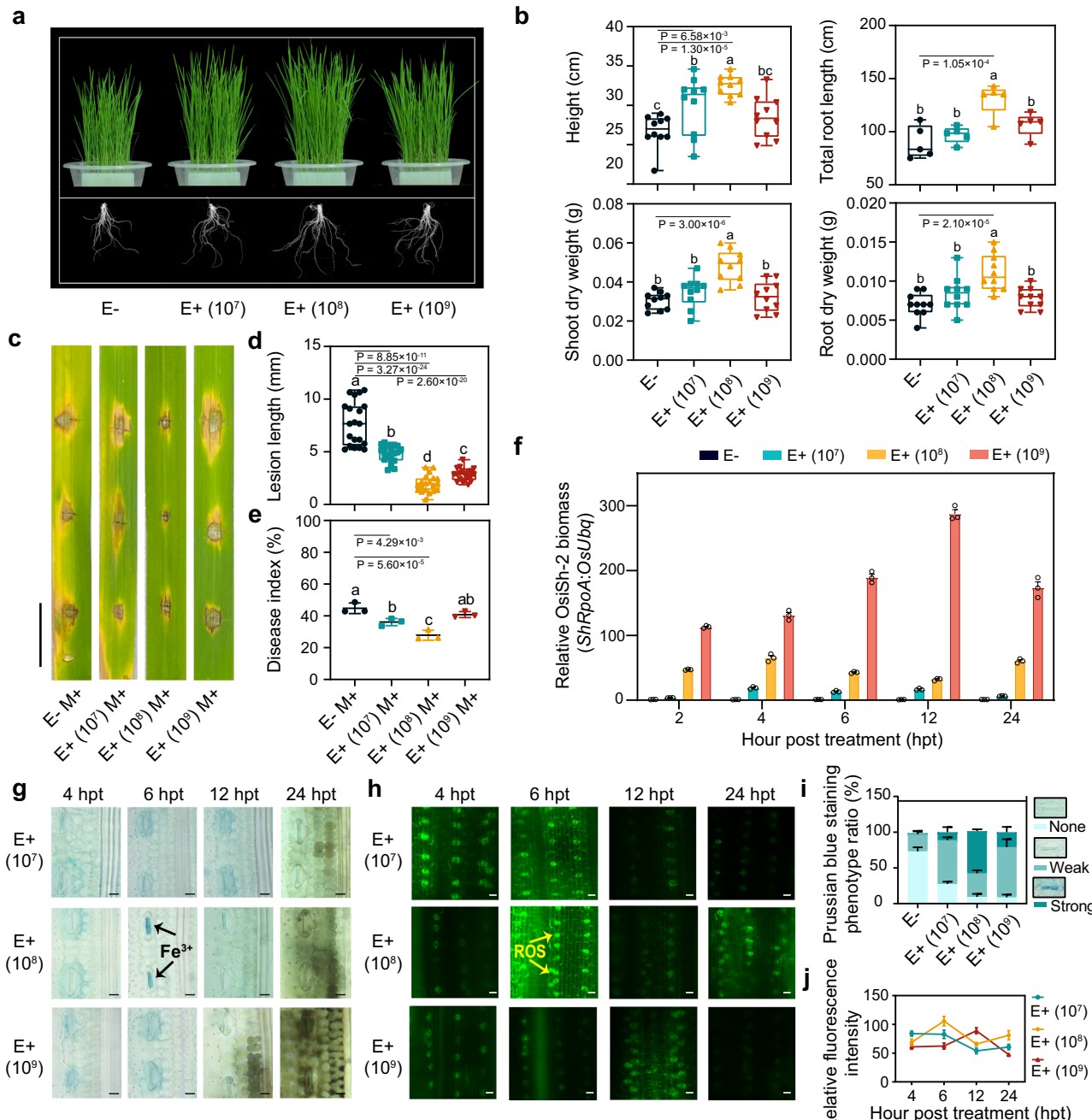

**Fig. 4 | Mutualism between OsiSh-2 and host rice requires the appropriate induction of ferroptosis. a** Representative image of 21-day-old E-, E+ (10⁸), E+ (10⁷), and E+ (10⁹) rice seedlings. E+ (10^{7/8/9}): rice whose leaves were sprayed with OsiSh-2 spore suspension in three concentrations as $10^7$, $10^8$, $10^9$ spores mL$^{-1}$. **b** Plant height (cm), total root length (cm), and shoot and root dry weights (g) were used to represent the seedling growth traits. Experimental repeats are displayed as box plots with individual data points. The error bars represent maximum and minimum values. Middle horizontal bars of boxplots represent the median, and the bottom and top represent the 25th and 75th percentiles ($n = 5$–10 for each). **c** Images of blast lesions on detached rice leaf segments at 5 days after punch inoculation with *M. oryzae* 70-15 at a concentration of $10^5$ conidia mL$^{-1}$. Scale bars: 1.0 cm. **d** The lesion length was measured at 5 days post-inoculation. Others are as in (**b**) ($n = 20$ for each). **e** The disease index was determined at 7 days post-inoculation by *M. oryzae*. Error bars indicate the mean±SDs ($n = 3$ for each). **f** The relative biomass of OsiSh-2 was determined by DNA-based qRT–PCR at the indicated hours post

treatment. Error bars indicate the mean±SDs ($n = 3$). **g** Prussian blue staining (blue colour) shows the accumulation of $Fe^{3+}$ in E+ (10^{7/8/9}) rice leaves at 6 hpt. Scale bars: 10 µm. The black arrows indicate $Fe^{3+}$ accumulation. **h** CM-H₂DCFDA staining shows the accumulation of ROS (H₂O₂, green fluorescence) in E+ (10^{7/8/9}) rice leaves at 6 hpt. Scale bars: 20 µm. The yellow arrows indicate ROS bursts. **i** Relative $Fe^{3+}$ accumulation is expressed as the Prussian blue staining phenotype ratio (%). Ratios indicate the proportions of designated staining phenotypes. Error bars indicate the mean±SDs ($n = 3$ for each). **j** Relative ROS accumulation is expressed as the relative fluorescence intensity of CM-H₂DCFDA-stained rice cells. The relative fluorescence intensity of ROS was calculated via ImageJ. The data shown indicate the means±SDs ($n = 20$ for each). The bars with different letters are significantly different (ANOVA, P < 0.05) according to Duncan's multiple-range test. All experiments were repeated independently three times with similar results. Source data are provided as the Source Data file.

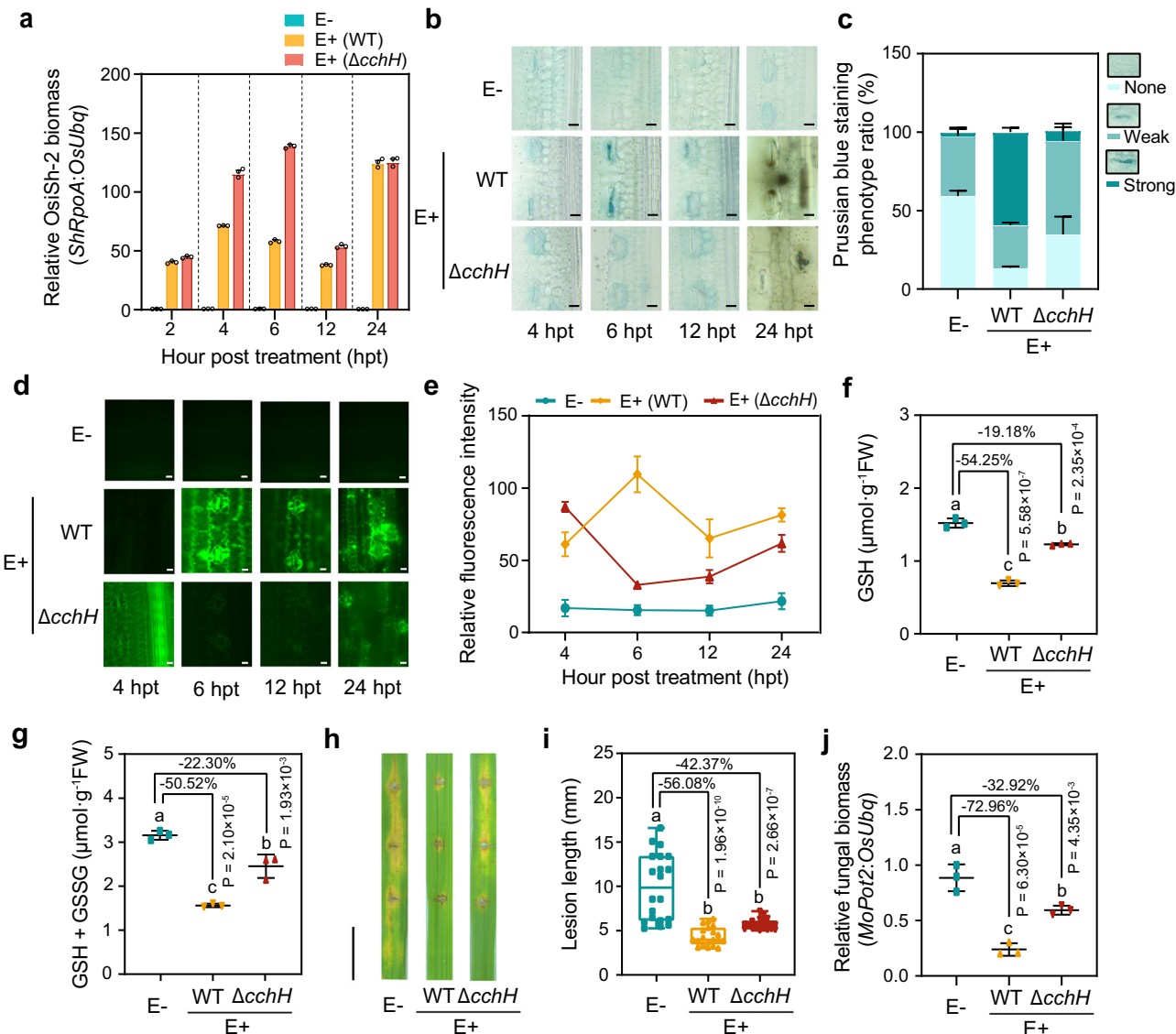

**Fig. 5 | OsiSh-2 triggers ferroptosis in a siderophore-dependent manner for mutualism with host rice. a** Endophytic biomass was calculated by DNA-based qRT–PCR in rice leaves at the indicated hour post treatment (hpt) by wild-type OsiSh-2 (WT) and Δ*cchH* (Sh*CchH* knockout mutant strains of OsiSh-2) at a concentration of $10^8$ spores mL$^{-1}$. Error bars indicate the mean±SDs ($n = 3$). **b** Prussian blue staining (blue colour) shows the accumulation of $Fe^{3+}$ in E-, E+ (WT), and E+ (Δ*cchH*) rice leaves at 6 hpt. Scale bars: 10 μm. **c** Relative $Fe^{3+}$ accumulation is expressed as the Prussian blue staining phenotype ratio (%). Ratios indicate the proportions of designated staining phenotypes. Error bars indicate the mean±SDs ($n = 3$ for each). **d** CM-H$_2$DCFDA staining shows the accumulation of ROS (green fluorescence) in rice samples same as **b**. Scale bars: 10 μm. **e** Relative ROS accumulation is expressed as the relative fluorescence intensity of CM-H$_2$DCFDA-stained rice cells. The relative fluorescence intensity of ROS was calculated via ImageJ. Error bars indicate the mean±SDs ($n = 20$ for each). The contents of GSH (**f**) and total GSH (GSH+GSSG) (**g**) in E-, E+ (WT), and E+ (Δ*cchH*) rice leaves at 12 hpt

were determined. Others are as in (**a**). **h** Images of blast lesions on detached rice leaf segments at 5 days after punch inoculation with *M. oryzae* at a concentration of $10^5$ conidia mL$^{-1}$. Scale bars: 1.0 cm. **i** The lesion length of detached rice leaves was measured via ImageJ. Experimental repeats are displayed as box plots with individual data points. The error bars represent maximum and minimum values. Middle horizontal bars of boxplots represent the median, and the bottom and top represent the 25th and 75th percentiles ($n = 20$ for each). **j** The relative fungal growth of *M. oryzae* was calculated by DNA-based qRT–PCR at 5 dpi in the rice leaves. Error bars indicate the mean±SDs ($n = 3$). The bars with different letters are significantly different (ANOVA, $P < 0.05$) according to Duncan's multiple-range test. The percentage changes followed by "+" or "-" above the bars were calculated by using the following formula: percent change = [(value of treated rice) - (value of untreated rice)] / (value of untreated rice) × 100%. All experiments were repeated independently three times with similar results. Source data are provided as the Source Data file.

(Supplementary Fig. 7e–g). That is, DFO might play a subtle role in the regulation of ferroptosis in rice rather than directly contributing to rice blast resistance. The above results further confirmed that the moderate ferroptosis triggered by OsiSh-2 (WT) is necessary for improving rice disease resistance against *M. oryzae*. Although siderophore coelichelin would have been more appropriate for this experiment, the compound is not commercially available, and we were unable to purchase it for use in this study.

## Ferroptosis induction is related to defence priming in host rice

Our previous study revealed that the induction of defence priming is an important disease resistance mechanism mediated by OsiSh-2. Priming is known as an adaptive, low-cost defensive trait activated by a stimulus, often resulting in an increased responsiveness of their immune system, thus enhancing biotic and abiotic stress tolerance[31]. We suspected that, as an important factor in the immune system, ferroptosis might be related to the primed state. Thus, the priming

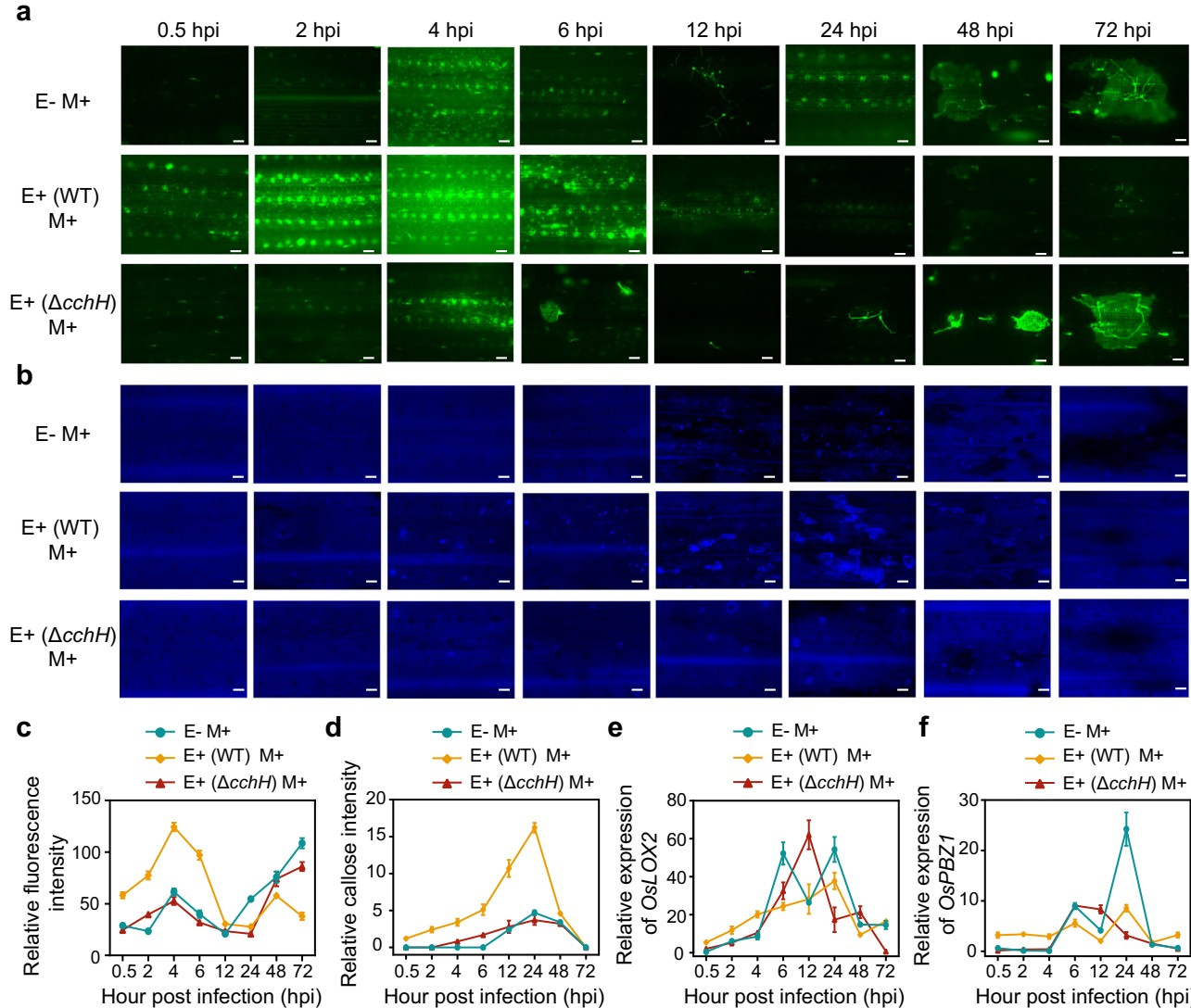

**Fig. 6 | Siderophore-dependent ferroptosis involved in OsiSh-2 to trigger defence priming against *M. oryzae* in host rice.** Fluorescence microscopy observations of ROS accumulation (as indicated by CM-H$_2$DCFDA staining) (**a**) and callose deposition (as indicated by aniline blue staining) (**b**) in E-, E+ (WT), and E+ (Δ*cchH*) rice at the indicated hours post infection (hpi) by *M. oryzae* (M+). Scale bars: 30 μm. **c** Relative ROS accumulation is expressed as the relative fluorescence intensity of CM-H$_2$DCFDA-stained rice cells. The relative fluorescence intensity of ROS was calculated via ImageJ. Error bars indicate the mean±SDs ($n$ = 20 for each).

**d** The abundances of callose intensity are represented by the percentage of relative callose intensities. Error bars indicate the mean±SDs ($n$ = 3 for each). The transcript levels of the immune response related genes *OsLOX2* (**e**) and *OsPBZ1* (**f**) were determined by RT–qPCR at different hpi in E- M+, E+ (WT) M+, and E+ (Δ*cchH*) M+ rice leaves. Error bars indicate the mean±SDs ($n$ = 3). All experiments were repeated independently three times with similar results. Source data are provided as the Source Data file.

processes, including ROS bursts, callose deposition, and the activation of defence-related genes, were determined in the E+ (WT) and E+ (Δ*cchH*) rice when *M. oryzae* was infected. A large amount of ROS accumulated in the stomata of the E+ (WT) M+ rice leaves at 0.5 to 6 h post infection (hpi) by *M. oryzae*, while the ROS in the E- M+ and E+ (Δ*cchH*) M+ rice just began to accumulate in small amounts from 4 hpi (Fig. 6, a, c). Similarly, callose deposition in the E+ (WT) M+ rice occurred at least 2 h earlier than it did in the E- M+ and E+ (Δ*cchH*) M+ rice, and the amount was also dramatically increased in the E+ (WT) M+ rice (Fig. 6b, d). Consistently, in the E+ (WT) M+ rice, the expression of two selected immune response genes, *OsLOX2* and *OsPBZ1*, was activated nearly 2-6 h earlier than that in the E-M+ and E+ (Δ*cchH*) M+ rice and then decreased to a relatively low level, which is beneficial for maintaining normal rice growth (Fig. 6e, f). These results indicated that the occurrence of effective ferroptosis in OsiSh-2-treated rice may be correlated with priming induction in the symbiont under biotic stress,

consequently enhancing disease resistance. The related mechanism still requires further study.

## Discussion

As a popular research topic on cell death, the newly identified iron- and ROS-dependent ferroptosis has been found to have multiple activities in mammals[32,33]. Only a few reports have indicated that ferroptosis is involved in plant and microbe systems, such as in wheat[34], the cyanobacterium *Synechocystis* sp. PCC 6803[35], and *M. oryzae*[36]. Intriguingly, several studies have demonstrated that pathogenic infections and control are involved in the host ferroptosis process[37]. For instance, ferroptosis is a major mechanism of necrosis during pathogenic infection by *Mycobacterium tuberculosis* in humans and animals[38]. *M. oryzae* suppress or induces ferroptosis at a basal/minimal level in blast-susceptible rice plants, thus enabling its successful invasion[19]. By inducing ferroptosis, the soilborne biocontrol agent *Pythium*

*oligandrum* has been reported to enhance soybean resistance to *Phytophthora sojae*[39]. However, the role of ferroptosis in host–microbe interactions is still underestimated. In this study, we provide a new model of ferroptosis-related plant-endophyte interactions in which the occurrence of ferroptosis plays a vital role in the mutualistic symbiosis and disease resistance of symbionts.

Endophytes are particular microbes that can occupy plant tissues without causing disease symptoms but are involved in mutualistic symbioses with plants, thus exerting multiple beneficial effects, such as growth promotion and disease resistance in host plants[40,41]. As an immune strategy that plants employ to respond to exogenous stimuli, cell death has also been reported in endophyte–host interactions. For instance, the root endophytic fungus *Piriformospora indica* requires host cell death for proliferation during mutualistic symbiosis with barley[42] and *Arabidopsis* hosts[43]. Cell death in *Arabidopsis* contributes to the successful colonization of the beneficial endophytic fungus *Serendipita indica* in the host[44]. Here, using an OsiSh-2 rice system, we determined that ferroptotic cell death participated in endophyte–plant symbiont formation, with considerable evidence. Initially, along with OsiSh-2 inoculation, $Fe^{3+}$ and ROS, two characteristic hallmarks of ferroptosis, simultaneously accumulated in the stomata at 6 hpt after OsiSh-2 was sprayed onto rice leaves (Fig. 1). In addition, corresponding to the essence of ferroptosis as an iron-dependent process, expression of the iron transport-related genes *OsIRO2* and *OsYSL15* was found in rice in response to OsiSh-2 treatment. The activation of *OsIRO2* and *OsYSL15* indicated a perturbation of iron homeostasis. For example, *OsIRO2* is critical for regulating the Fe uptake process in rice under Fe-deficient conditions[45]. *OsYSL15* is an iron-regulated $Fe^{3+}$ transporter that plays roles in Fe uptake from the rhizosphere and Fe transport and distribution in rice[46]. In addition, the biosynthesis of GSH is central to the control of ferroptosis[47], and these components are part of the most critical antioxidant pathway in living cells. For example, once GSH was depleted, cellular ferroptosis sensitivity increased in kidney[48] and tumour cells[49]. In this study, the decreased content of GSH indicated that the detoxification capacity of rice cells diminished, thus leading to lipid peroxidation (Fig. 1). Furthermore, OsiSh-2-triggered ferroptosis was effectively inhibited by ferroptosis inhibitors (Fer-1, DPI and DFO) and strengthened by inducers (erastin and $FeCl_3$) (Fig. 2, Supplementary Fig. 2).

Ferroptosis needs to be precisely regulated in the fight between plants and pathogens. The pathogen can inhibit ferroptosis in the pathogenesis process to evade the immune defence and then induce ferroptosis at the late stage of infection for progeny release[37]. The same logic applies to mutualistic symbiosis. We believe that establishing an endophyte–plant symbiotic relationship would exploit regulated ferroptosis for symbiont growth and stress resilience. Especially for the rice endophyte OsiSh-2, our previous studies have shown that the beneficial effects of OsiSh-2 are based on its harmonious symbiotic relationship with host rice[28]. Our results support this hypothesis. There were two peaks of OsiSh-2 accumulation in rice, i.e., 4 and 24 hpt. The mass of OsiSh-2 was then controlled to be decreased to an appropriate level. This is reasonable, as endophytes and plants adapt to each other and maintain a harmonious symbiosis. Otherwise, the overproliferation of OsiSh-2 would develop as a pathogenic invasion. We believe that the first regulation of OsiSh-2 mass at 6 hpt is attributable to the ferroptosis immune response, which was triggered at 6 hpt. However, it should be noted that the occurrence of ferroptosis is an energy-consuming process[50], and continuous ferroptosis in rice might inhibit the growth of plants. Indeed, as shown in Fig. 1c, the ferroptosis was suppressed after 24 hpt. Therefore, we surmised that the ferroptosis was functioning as an early immunity response model to endophytic OsiSh-2 in rice. The second regulation to inhibit the increase in OsiSh-2 after 24 hpt may be ascribed to other immune system processes but not to ferroptosis. For example, there was still a strong ROS burst in the rice leaves of E+ rice at 24 hpt (Fig. 1e). In our

previous study, we also reported that OsiSh-2 can trigger rice immune responses by secreting α-Mannosidase ShAM1 to digest the rice cell wall and release damage-associated molecular patterns, and cell death was also observed at 24 hpt[51]. In addition, we revealed that appropriate ferroptosis was an essential factor for OsiSh-2 colonization and shaping the mutual symbiont. First, ferroptosis was observed only at the optimal inoculation concentration of OsiSh-2 ($10^8$ spores $mL^{-1}$), which led to the best growth promotion and disease resistance effect on the host rice. The $10^7$ and $10^9$ spores $mL^{-1}$ of OsiSh-2 showed weaker and stronger colonization capability, but both triggered only a very low degree of ferroptosis (Fig. 4). We thought that the smaller inoculum density was insufficient to cause a certain degree of ferroptosis, while an excessive inoculant population suppressed the occurrence of ferroptosis. Similarly, when the siderophore biosynthesis-deficient mutant ΔcchH was applied to rice, compared with OsiSh-2 (WT), it showed a higher colonization density, a relatively low level of ferroptosis, and reduced disease resistance (Fig. 5). In addition, the results of ferroptosis inhibitor and inducer application also showed lower or higher levels of ferroptosis, with weakened or promoted colonization of OsiSh-2 in host rice and disease resistance of the symbionts, respectively (Fig. 3). Although the effect of ferroptosis on the mutualistic interaction between endophytes and their host plants has not yet been reported, similar sophisticated cross-talk between the symbiont partners has been thoroughly documented in the root endophytic fungus *P. indica*. These strains could improve the biomass of host plants and induce plant resistance to biotic and abiotic stress by interfering with and exploiting host programmed cell death for successful colonization but did not provoke sustained defence in host plants[42,43,52,53]. Overall, we believe that OsiSh-2 interferes with host ferroptosis to form a mutualistic interaction and benefit plants.

Therefore, how does ferroptosis mediate disease resistance promotion in this endophyte–plant system? Our previous studies have shown that OsiSh-2 can improve disease resistance by the induction of a priming response in host rice. OsiSh-2 functions as a stimulus that triggers the primed state, which can respond to the newer stress, i.e., pathogen invasion that is faster and more robust[28]. Similarly, some beneficial bacteria, such as the plant growth-promoting rhizobacteria *Pseudomonas* spp[54]., *Trichoderma* spp[55] and *Bacillus* spp[56]. can serve as priming elicitors. These bacteria can induce mild but effective immune activation, which leaves plants in a primed state to accelerate defence responses when pathogens attack[57]. However, the mechanism of priming induction by microbes remains unclear. These data indicated that OsiSh-2-induced ferroptosis plays a role in maintaining the defence-primed state in host rice. In Fig. 6a, the pattern of ROS increase observed by CM-$H_2$DCFDA exhibited a transient ROS peak at 4 hpt with virulent *M. oryzae* in E- M+ rice, followed by a typical pattern-triggered immunity (PTI) response showing low ROS levels at 24, 48, and 72 hpt. When the ΔcchH mutant strain was inoculated, it also displayed a similar transient PTI response to that observed in E- M+ rice. In contrast, in the case of E+ (WT) M+ rice, an earlier and stronger ROS peak was observed from 0.5 to 6 hpt. Similarly, callose deposition around the stomata was observed in E+ (WT) M+ rice leaves at least 4 h earlier than that in E- M+ rice. The more rapid and strong accumulation of ROS and callose in E+ (WT) M+ than in E- M+ rice indicates that defence priming was triggered by OsiSh-2. Based on our previous studies, the priming strategy used by OsiSh-2 markedly improved the pathogenic recognition and response capability in the OsiSh-2-rice symbiont[28]. To date, this is the first report on the relationship between ferroptosis and priming. On the other hand, compared to the siderophore-deficient ΔcchH mutant strain, the accumulation of ROS in E+ (WT) M+ rice might be attributed to the OsiSh-2-secreted siderophore, which enhances PTI as it is recognized as a PAMP (pathogen-associated molecular pattern) or elicitor[58]. However, when we added the DFO exogenously, a common siderophore produced in *Streptomyces*, we only observed an ROS burst in E- DFO+ rice, but the disease

resistance against *M. oryzae* was not detected. Other siderophores produced by OsiSh-2 may act as immunity elicitors, and the detailed mechanisms of action need to be further elucidated.

Notably, other modes of OsiSh-2 action against rice blast disease exist in host rice, including secreting metabolites for directly antagonizing pathogens. For example, secreting fungal cell wall lytic enzymes (such as chitinase and β−1,3-glucanase) and antibiotics such as nigericin have been found to help OsiSh-2 to directly antagonize *M. oryzae* by damaging fungal cell membrane integrity[59]. Competing for nutrition with *M. oryzae* through siderophore production in vitro can inhibit the growth of *M. oryzae*[29]. In this study, we found that moderate ferroptosis is necessary for establishing an optimal mutualistic symbiotic relationship between OsiSh-2 and rice, which consequently activates an immune resistance to rice blast disease. This induction of an immune response for indirect disease resistance is not the only contribution of ferroptosis. Our recent study showed that applying ShAM1, an α-mannosidase purified from OsiSh-2 culture supernatants, can trigger rice immune responses and improve blast resistance with a 45.9% reduction in blast disease lesion length and a 67% reduction in fungal biomass[51]. Therefore, we considered that the assistance of OsiSh-2 in disease resistance in host rice depends on multiple mechanisms, and ferroptosis is one of them.

OsiSh-2 possesses a great ability to produce siderophores, including DFO and COE. Siderophores are a class of low-molecular-weight (<1 kDa) organic compounds with a high $Fe^{3+}$ affinity constant and are thus used by microbes to effectively capture iron from the environment[58]. Siderophores are also closely involved in iron-related biological activities in host-microbe symbiosis. For instance, endophytic *Streptomyces* sp. CoT10 can promote the growth of the host *Camellia oleifera* with high P and Fe acquisition mediated by its siderophores[60]. The rhizobacteria *Pseudomonas* spp. and OsiSh-2 can effectively mitigate Fe deficiency in the hosts *Arabidopsis*[61] and rice, respectively, with the help of their secreted siderophores[62]. We hypothesized that OsiSh-2 might also participate in ferroptosis regulation by secreting siderophores, which can inhibit ferroptosis in the same way as the commonly used DFO. Compared with the exogenous application of DFO, the dynamic biosynthesis of siderophores by OsiSh-2 might be more efficient. We constructed a siderophore biosynthesis-deficient mutant of OsiSh-2 (Δ*cchH*), in which only siderophore production was reduced, and microbial growth was not affected. Δ*cchH*-treated rice failed to undergo substantial ferroptosis. Correspondingly, the colonization of Δ*cchH* increased dramatically, and its disease resistance significantly decreased compared to that of OsiSh-2 (WT) (Fig. 5). The Δ*cchH* mutant failed to induce ferroptosis, which might be due to the following reasons. Siderophores have been reported as elicitors in host plant immunity. The biocontrol effect of *Pseudomonas aeruginosa* in tomato against *Botrytis cinerea*[63] and *M. oryzae*[64] benefits from siderophore-stimulated immune defence. However, the detailed mechanism of action of siderophores has not been well elucidated[58]. Here, Δ*cchH*, with reduced production of siderophore COE, triggered only a basal level of ferroptosis in host rice (Fig. 5), confirming that the role of COE in the OsiSh-2-rice system was mainly as an immune elicitor. In addition, as an essential tool for microbes to capture iron, compared with that of OsiSh-2 (WT), the weakened siderophore production of Δ*cchH* might make the microbes more dependent on the host for iron supplementation, resulting in overproliferation in plants (Supplementary Fig. 6,f, g, Fig. 5a). Therefore, Δ*cchH* successfully inhibited the occurrence of ferroptosis instead of triggering runaway ferroptosis. However, excessive endophytic colonization inevitably affected the formation of an optimal symbiotic relationship with the host rice. Indeed, although Δ*cchH* ultimately colonized rice tissues and had no negative effect on rice growth, the disease resistance against *M. oryzae* of the symbiont was significantly reduced (Fig. 5h-j). Overall, during the interaction of OsiSh-2 and host rice, the regulatory mechanisms of ferroptosis by siderophores might be dynamic and sophisticated.

In conclusion, our findings highlighted moderate ferroptosis induced by endophytic OsiSh-2 and its essential role in promoting mutualistic symbiosis with host rice. We suggest that ferroptosis can improve disease resistance against *M. oryzae* infection by priming-induction mechanism, a notion that requires further study. Exploring the mechanism of ferroptosis in plant–microbe interactions is of great theoretical importance and practical for disease control and management of plant fitness.

## Methods
### Materials and growth conditions
Rice (*Oryza sativa* cv. indica 9311) seeds were provided by Yahua Seeds Science Research Institute, Longping High-Tech, in Changsha, Hunan, China. Surface-sterilized rice seeds were germinated and then transplanted into pots containing an International Rice Research Institute (IRRI) nutrient solution[65]. The growth chambers were set to have a 16/8 h light/dark photoperiod at 28 °C and 80% humidity.

Endophytic *Streptomyces hygroscopicus* OsiSh-2 (GenBank accession number GCA_001705785.1, China General Microbiological Collection Center accession number CGMCC-8716) was isolated from rice sheaths and routinely cultured in International Streptomyces project 2 (ISP2) solid media at 30 °C[59]. The rice blast pathogenic fungus *M. oryzae* 70−15 was routinely cultured on solid complete media at 28 °C. Spore suspensions of OsiSh-2 and 70−15 were prepared as described[28].

### CM-H$_2$DCFDA for ROS detection and callose deposition assays
CM-H$_2$DCFDA staining was performed as described[19], with slight modifications. Briefly, rice leaves were excised and immersed in Milli-Q water for 5 min at room temperature to minimize wound-induced ROS production. Then, the detached leaves were incubated in 10 μM CM-H$_2$DCFDA (Molecular Probes Life Technologies) in the dark for 30 min on a horizontal shaker. The detached leaves were then washed twice with 1 × PBS for 5 min in the dark. ROS localization inside leaf cells was observed using an EVOS™ M5000 Imaging System (Thermo Fisher Scientific, USA). To ensure the consistency of the device parameters at the same magnification, the fluorescence photos of rice leaves in different treatments were taken under the same light intensity. The relevant parameters were as follows: light:1.866; exposure:30 ms; and gain:15 dB.

The callose deposition assay was performed as described previously[28]. Briefly, the rice leaves were fixed and cleared in absolute alcohol with frequent changes of the fresh solution, and the chlorophyll was removed. The transparent leaves were washed with 70 mM sodium phosphate buffer three times and then incubated in a stain solution (70 mM sodium phosphate buffer; 0.01% aniline blue, Macklin) for 2 h in the dark. Then, the leaves were observed using an inverted microscopy under UV light (340 to 380 nm) (TS2R-FL, Nikon, Tokyo, Japan). Three replicates were included per treatment.

### Trypan blue and Prussian blue staining for cell death detection
Trypan blue staining for cell death detection was performed as described previously[66]. Briefly, leaves were submerged in trypan blue solution (0.25% trypan blue, 25% lactic acid, 23% water-saturated phenol, 25% glycerol in ddH$_2$O) in boiling water for 2 min and then incubated at room temperature overnight for staining. The trypan blue solution was replaced with chloral hydrate solution (25 g in 10 mL of H$_2$O) for destaining. After multiple exchanges of chloral hydrate solution for 3 days, the samples were equilibrated with 70% glycerol. Images were captured by an SMZ1000 stereoscope (Nikon, Tokyo, Japan) using ImagePro Plus image analysis software.

Prussian blue staining of $Fe^{3+}$ accumulation was performed as described[21], with slight modifications. Briefly, leaf tissues were cut into 5- to 7 cm lengths and incubated in 4% (w/v) potassium ferrocyanide and 4% (v/v) hydrochloric acid (1:1, v/v) for 15 h at room temperature. Then, the leaf tissues were washed with water three times and destained with chloral hydrate. Images were captured using a light microscopy equipped with a digital camera.

The TUNEL assay can sensitively detect DNA fragmentation caused by cell death signaling cascades, and the positive signal was shown with green fluorescence[67]. For the TUNEL analysis, the rice leaves were embedded in paraffin and cut into thin slices before detection. After eluting the paraffin with xylene, the corresponding slices were stained with PI and TUNEL, finally the cell death signal was observed using a fluorescence microscopy[67]. The green represents TUNEL-positive signals, the excitation wavelength is 465–495 nm and emission wavelength 515–555 nm; Red signal indicates PI staining by excitation wavelength 510–560 nm and emission wavelength 590 nm.

## Cell viability assay
The viability of suspension cells was determined by PI staining. PI passes through dead cell membranes and generates red fluorescence by forming a PI-nucleic acid conjugate. Then the treated suspension cells and 25 μM PI solution were stained in the dark for 20 min on a horizontal shaker. Stained cells were then washed twice with PBS and observed with a light/fluorescence microscopy (Olympus BX51). The experiment was conducted three times.

## Rice leaf spray and *M. oryzae* punch inoculation assays
Rice leaf inoculation assays were performed following the method described previously[28]. Briefly, we sprayed the spore suspension ($10^8$ spores mL$^{-1}$) of OsiSh-2 with 0.2% (v/v) Tween 20 evenly on the surface of rice leaves using a mini spray bottle atomizer (20 mL/50 plants). Water containing 0.2% (v/v) Tween 20 was used as a control. The sprayed rice seeds were placed in pots containing an International Rice Research Institute (IRRI) nutrient solution and the growth chambers were set to have a 16/8-h light/dark photoperiod at 28 °C and 80% humidity.

For *M. oryzae* punch inoculation assays. Rice leaf inoculation assays were performed following a previously described method[68]. Briefly, detached leaves of four-week-old rice were lightly wounded with a mouse ear punch, and then 6 μl of *M. oryzae* spore suspension (concentration of $5 \times 10^5$ conidia/mL in 0.02% Tween-20) was applied to the wounded leaves. Sterile 6-benzylaminopurine (6-BA) water (0.1%) was added to the culture dish to keep it moist. The lesion size was measured after incubation for 5–6 days at 28 °C.

For the defence priming assay, surface-sterilized rice seeds were sown into pots containing the IRRI-containing nutrient solution for 1 week. Spore suspensions of OsiSh-2 (WT) and ΔcchH ($10^8$ spores mL$^{-1}$) with 0.2% (v/v) Tween 20 were sprayed onto the rice leaves. Water containing 0.2% (v/v) Tween 20 was used as a control. After 5 days, the rice leaves were sprayed with *M. oryzae* spore suspension ($5 \times 10^5$ spores mL$^{-1}$). The cells were kept moist and cultured in darkness overnight at 26 °C, after which they transferred to normal growth conditions with a 16/8 h light/dark photoperiod.

## Antagonism assay against *M. oryzae*
To investigate the antagonism of OsiSh-2 (WT) and ΔcchH against *M. oryzae*, a dual-culture assay was performed[68]. Briefly, *M. oryzae* was inoculated in the centre of the PDA plate, and the strains were streaked one line 30 mm above the centre. Then, the *M. oryzae* colony radius was measured to assess the inhibition percentage of growth by OsiSh-2 (WT) and ΔcchH. The percent reduction in colony radius compared to the control was calculated and reported as an inhibition percentage using the following formula: inhibition percentage (%) = [(R1 − R2)/R1] × 100, where R1 is the colony radius of *M. oryzae* in the control

(water) and R2 is the colony radius of *M. oryzae* in dual culture plates towards the *Streptomyces* colony.

## Gene expression and microbial biomass analysis
The gene expression levels and the biomass of pathogenic fungi and endophytes in rice plants were quantified. Total RNA extraction was performed using a Plant Total RNA Isolation Kit (Sangon Biotech, China). The resulting RNA was reverse transcribed using a PrimeScript RT reagent kit with gDNA Eraser (TaKaRa, Japan). Gene expression was analysed by qRT–PCR as described previously[28]. Independent experiments were repeated twice, and all reactions were performed in triplicate. The primers used for qRT–PCR are listed in Table S1. The transcript data were normalized by using β-actin expression levels as internal references. Total DNA was extracted from the samples using a Dzup (plant) genomic DNA isolation reagent kit (Sangon Biotech, Shanghai, China). DNA-based quantitative real-time PCR (qRT–PCR) was performed using a CFX96 real-time system instrument (Bio-Rad, USA). The relative biomass of OsiSh-2 in rice was calculated using the threshold cycle ($C_t$) of *ShRpoA* (a DNA-directed RNA polymerase subunit alpha of OsiSh-2) DNA against the $C_t$ of *OsUbq* (a rice genomic ubiquitin gene) DNA as a ratio (*ShRpoA/OsUbq*), represented by the equation $2^{Ct\,(OsUbq)-CT(ShRpoA)}$. The relative fungal biomass of *M. oryzae* in rice was calculated using the threshold cycle ($C_t$) of *MoPot2* (an inverted repeat transposon of *M. oryzae*) DNA against the $C_t$ of *OsUbq* (a rice genomic ubiquitin gene) DNA as a ratio (*MoPot2/OsUbq*), represented by the equation $2^{Ct\,(OsUbq)-CT(MoPot2)}$.

## Treatment with Fer-1, DFO, DPI, FeCl₃ and erastin
For Fer-1 treatment, the rice leaves sprayed with the OsiSh-2 spore suspension were incubated in the chambers for 6 h at 28 °C and then vacuum infiltrated[19] in water (mock) or 10 μM Fer-1 solution for 10 min, followed by incubation under the same conditions in the chambers. For the DFO and DPI treatments, the rice leaves sprayed with the OsiSh-2 spore suspension were incubated in the chambers for 4 h at 28 °C and then sprayed with 100 μM DFO and 10 μM DPI solution, respectively, followed by incubation under the same conditions in the chambers.

For $FeCl_3$ and erastin treatment, rice leaves sprayed with the OsiSh-2 spore suspension were incubated in the chambers for 4 h at 28 °C and then sprayed with a 5 μM $FeCl_3$ and 10 μM erastin solution, followed by incubation under the same conditions in the chambers.

## Lipid peroxidation, GSH content and enzyme activity assays
As a product of unsaturated fatty peroxidation present in plant cells, malondialdehyde (MDA) was quantified according to its reaction with thiobarbituric acid, as described[19], with slight modifications. Briefly, the leaf tissues were collected and ground in liquid nitrogen. The resulting fine tissue powder was mixed with 10% (w/v) trichloroacetic acid and centrifuged at $1685 \times g$ for 10 min The supernatant was added to 0.6% (w/v) thiobarbituric acid (1:1, v/v). The mixture was then incubated in boiling water for 15 min, cooled in an ice bath for 5 min, and then centrifuged at $12,400 \times g$ for 10 min at 4 °C. The absorbances of the resultant supernatant were measured at 450, 532, and 600 nm. The MDA concentration (C) was calculated according to the equation C = 6.45 (OD$_{532}$–OD$_{600}$) · (0.56 OD$_{450}$).

For GSH and GSH peroxidase measurements, rice leaves (0.05 g) were collected from different treatments at a series of time points. The GSH content was determined by the M0301A Reduced Glutathione Kit (enzyme-conjugated method)/M0302A Oxidized Glutathione Kit (enzyme-conjugated method) according to the kit instructions. The GSH/GSSG contents in the extraction solution were measured by using a microporous plate reader at 412 nm.

## Construction and screening of siderophore deletion mutants
Plasmid construction and intergeneric conjugation between *Escherichia coli* and *Streptomyces* were performed as described previously[68].

Briefly, the plasmid pYH7-*cchH* was introduced into the wild-type strain OsiSh-2, with the helper plasmid pUZ8002 in *E. coli* ET12567 through conjugation on the MS agar plates. Single colonies of the exconjugants were selected by inoculating in the MS plates with 25 μg/mL apramycin and 30 μg/mL nalidixic acid. The colonies with apramycin resistance were reinoculated onto MS agar plates for relaxation. The apramycin sensitive colonies were then selected as candidates for double-crossover mutants for PCR screenings using test-F and -R as the primers (Supplementary Table 1). The plasmid pYH7 and plasmid were kind gifts from Prof. Yuhui Sun, School of Pharmaceutical Sciences, Wuhan University.

### Siderophore production assay

To assess the siderophore producing capability, OsiSh-2 strains were inoculated into a modified Fe-deficient hydroponic solution[60] (soluble starch 20 g, NaCl 0.5 g, $KNO_3$ 1 g, $K_2HPO_4 \cdot 3H_2O$ 0.655 g, $MgSO_4 \cdot 7H_2O$ 1.025 g, deionized water volume to 1 L, and pH = 7.0) for 12 days. Culture broth was collected each day and the siderophore content was determined by using a commonly used chrome azurol S (CAS) method[29]. The absorbance at 630 nm of each CAS solution was determined to quantify the siderophore activity. Here, we used two calculation methods to represent the siderophore activity.

1) Siderophore units (%), containing all kinds of siderophores that can be detected by the CAS method[69].

$$\text{Siderophore unis}(\%) = \frac{Ar - As}{Ar}$$

where $A_r$=absorbance at 630 nm of the reference sample (CAS solution and uninoculated broth), and $A_s$=absorbance at 630 nm of the sample (CAS solution and supernatant of sample).

2) For the method of relatively quantitative of siderophore. The standard curve ($R^2$å 0.99) was established for relative quantitative analysis by setting a series of known concentrations of desferrioxamine B (DFO-B) to measure the absorbance at 630 nm[70,71]. The siderophore content in the sample was calculated by the standard curve.

The siderophore production was assessed by Chrome Azurol S (CAS) solid media as described[29]. Briefly, inoculate 2-week- old mycelium of OsiSh-2 and Δ*cchH* with same amount on the medium, the plates were incubated at 30 °C conditions for 14 days and then monitored for the appearance of an orange halo zone around the actinomyces on blue coloured agar media, and the zone area was then measured, each group of five parallel.

### Statistical analysis

Statistical parameters are reported in the figures and figure legends. Normally, the statistical analysis of the data was performed by one-way repeated-measures analysis of variance (ANOVA) using SPSS software (Chicago, IL, USA), followed by Tukey's post hoc test or Duncan's multiple range tests, which were used to compare the means for the multiple groups. The $P$-value < 0.05 was considered to indicate statistical significance.

### Reporting summary

Further information on research design is available in the Nature Portfolio Reporting Summary linked to this article.

## Data availability

The authors declare that all data supporting the findings of this study are available within the paper and the supplementary files. Source date are provided with this paper. Source data are provided with this paper.

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

## Acknowledgements

This research is financially supported by the National Natural Science Foundation of China (32172497 and 32100246), the Key research and development program of Hunan province (2022NK2050 and 2023NK2024), Natural Science Foundation of Hunan Province (2021JJ40051), Postgraduate Scientific Research Innovation Project of Hunan Province (CX20220426).

## Author contributions

Y.Z., Y.G., and X.X. designed the experiments. X.X., Y.G., and J.Z. performed most of experiments, analyzed the data, and wrote the manuscript. Q.N. performed bioinformatic analyses and mutant construction. H.L. cultured bacteria and plants. R.Z. reviewed the manuscript. Z.B., X.Z., J.Z., K.C., and Z.Q. assisted in experiments and discussed the results. Y.Z. and X.L. guided the whole process of experiments. All authors read and approved the final manuscript.

## Competing interests

The authors declare no competing interests.
