## [Peer Review File · Nature Communications]

Ferroptosis induction in host rice by endophyte OsiSh-2 is necessary for mutualism and disease resistance in symbiosisReviewer #1 (Remarks to the Author):

λ Comments for author

This paper is an intriguing research study that aims to elucidate the role of ferroptosis, observed during the symbiotic establishment between endophytic *Streptomyces hygroscopicus* (OsiSh-2) and rice, in providing disease resistance against virulent *M. oryzae*. The key issue is whether ferroptosis is limited to the symbiotic relation between rice and OsiSh-2 or if it also contributes to the induction of disease resistance against virulent *M. oryzae* 70-15.

I have identified several areas in the paper that require logical revisions, and I have listed them below.

Major Points)

1. In Fig. 1a and 1b, it is not clear from the images alone whether the spots on the rice leaves under the E+ condition (OsiSh-2 inoculated) represent typical ferroptosis accompanied by iron- and ROS-dependent cell death with lipid oxidation, or if they indicate just a ferroptosis-like response. This ambiguity arises because even in processes like PTI where ROS accumulation occurs, there is a possibility of iron accumulation and cell death in some cells, even if it is not typical ferroptosis. Therefore, in addition to the current Trypan blue images in Fig. 1b, it would be helpful to include microscopic images with Prussian assay showing iron accumulation in the cells where ferroptotic cell death occurred, similar to Fig. 1c style.
2. In Fig. 1c, the image demonstrates that when OsiSh-2 is inoculated, there is a higher accumulation of iron at 6 hpt compared to the non-inoculated condition. However, at 12 and 24 hpt, the iron levels decrease and become similar to those of the non-inoculated rice. This finding makes it difficult to conclude that it leads to typical ferroptosis and suggests a higher possibility of cell recovery rather than the occurrence of typical ferroptotic cell death. If ferroptosis occurs after 24 hpt (eg. 36, 48 hpt), please add microscopic evidences to support this claim.
3. In Fig. 1k and 1i, the authors wrote in Line 107, 'Compared with that in the (E-) rice, the expression of the GPX4-encoding gene OsGPX4 in the (E+) rice was significantly downregulated at 6 hpt (Fig. 1i), and the enzyme activity of GPX4 decreased by 23.9%.' However, in animals, inhibiting GPX4, a glutathione peroxidase 4, leads to the induction of glutathione-deficiency. In contrast, it is known that plant GPX4 lacks the essential selenocysteine residue, which is crucial for glutathione oxidation. Consequently, plant GPX4 is incapable of efficiently oxidizing GSH (glutathione) (Iqbal et al., 2006; Bela et al., 2022). Therefore, this line of reasoning requires modification.
-Iqbal, A., Yabuta, Y., Takeda, T., Nakano, Y., & Shigeoka, S. (2006). Hydroperoxide reduction by thioredoxin-specific glutathione peroxidase isoenzymes of *Arabidopsis thaliana*. *The FEBS journal*, 273(24), 5589–5597. <https://doi.org/10.1111/j.1742-4658.2006.05548.x>
-Bela, K., Riyazuddin, R., & Csiszár, J. (2022). Plant Glutathione Peroxidases: Non-Heme Peroxidases with Large Functional Flexibility as a Core Component of ROS-Processing Mechanisms and Signalling. *Antioxidants* (Basel, Switzerland), 11(8), 1624. <https://doi.org/10.3390/antiox11081624>
4. In Figure 2b and 2c, when erastin is co-treated with OsiSh-2, a more pronounced increase in ferric ions and ROS burst occurs compared to OsiSh-2 treatment alone. This can be attributed to the blockade of cysteine influx by erastin, leading to glutathione-depletion due to inhibition of glutathione synthesis. However, there are no typical ferroptotic cell death images presented after erastin treatment in this Figure. Please add photographic evidence demonstrating the occurrence of ferroptosis.
5. In the case of (E+ mock) in Fig. 3b, the biomass of OsiSh-2 decreased significantly at 12 hpt compared to 6 hpt, but then increased again at 24 hpt. The authors explained this as 'OsiSh-2 hijacked the activity of the immune system of the host rice and colonized successfully' (Line 135). Please provide an explanation of the observed data regarding whether the number of ferroptotic cells in rice continues to increase after 36 or 48 hpt or if the increase in rice ferroptosis cells ceases after successful colonization of OsiSh-2.
6. In Figure 3h graph, when the ferroptosis inhibitor Fer-1 was applied to completely suppress ferroptosis in (E+, Fer-1, M+), it resulted in only a 35.35% increase in blast disease lesion length compared to (E+, M+), while the fungal biomass increased by only 38% (Line 150). However, if ferroptosis induced by E+ is indeed a key factor in disease resistance against *M. oryzae*, then treating with Fer-1 to completely inhibit ferroptosis should lead to a further increase in lesion length and fungal biomass, approaching the levels of E-, but instead, we observe a limited increase

of only 35-38%. This indicates that ferroptosis induced by OsiSh-2 (E+) only partially contributes to the increase in rice disease resistance against *M. oryzae*. These results suggest that factors other than ferroptosis significantly contribute to rice disease resistance. One such factor could be iron chelation by siderophores secreted by OsiSh-2 (E+) in the symbiotic relationship between OsiSh-2 (E+) and rice. In fact, Figure 5h shows that even with a mere 32.5% reduction in siderophore secretion compared to the WT OsiSh-2 strain, when the Δ cchH mutant strain was inoculated, the *M. oryzae* fungal biomass increased significantly by 40.04% compared to WT OsiSh-2 treatment (Line 201). If the Δ cchH mutant strain, which has a 100% reduction in siderophore secretion compared to the WT OsiSh-2 strain, were used, it can be anticipated that the *M. oryzae* fungal biomass would increase even more significantly than the current 40.04%. Therefore, when considering the results of these two experiments (Fig. 3h, i and Fig. 5g, h) together, they collectively demonstrate that siderophores secreted by OsiSh-2 strain contribute significantly more to rice disease resistance against *M. oryzae* than the ferroptosis formed during the symbiotic process of the WT OsiSh-2 strain in rice. This indicates a flaw in the logic proposed by the authors. Please provide a response to this comment.

7. In Figure 5b, the result of Δ cchH mutant strain inoculation suggests that the reduction in chelation due to siderophore mutation leads to increased iron uptake by *M. oryzae*, resulting in the restoration of its pathogenicity. In other words, the observed increase in disease resistance against *M. oryzae*, as claimed in the paper, may be partially attributed to ferroptosis, but it could also be the result of the stronger iron chelation effect exerted by the siderophore secreted by OsiSh-2. To demonstrate this, additional experiments should be conducted by treating the Δ cchH mutant strain, which is a siderophore mutant, with exogenous coelichelin treatment and observing the changes that occur. Please add these experimental results to the Figures.

8. The results in Figure 5f demonstrate a strong correlation between the extent of iron uptake inhibition by siderophore into *M. oryzae*, as observed in Shen et al., 2020 (New Phytologist 227: 1831-1846), and the decrease in fungal biomass (32.92%) (Figure 5h), which aligns closely with the siderophore production levels of OsiSh-2 (WT: 100%; Δ cchH strain: 32.5% lower than WT siderophore secretion). This finding suggests that rice disease resistance increases proportionally with siderophore production. Additionally, in Lane 316, the authors also hypothesize that siderophore itself could serve as an elicitor, activating plant immunity. Despite this, it remains unclear why the authors attribute defense priming to ferroptosis. Please provide an explanation for this inconsistency.

9. In Figure 5f, the increased lesion length of *M. oryzae* in rice leaves infected with Δ cchH mutant strain compared to OsiSh-2 WT can also be attributed to the siderophore produced by WT OsiSh-2, which prevents the uptake of ferric ions by *M. oryzae*, leading to subsequent reduction in *M. oryzae* virulence. In other words, this is unrelated to rice ferroptosis induced by OsiSh-2. This finding aligns with a previous study by the same group in 2018 (Zeng et al., 2018 Microbial Ecology, <https://doi.org/10.1007/s00248-018-1189-x>), where they explained the exceptional iron acquisition competitiveness of OsiSh-2 compared to *M. oryzae*, resulting in the effective inhibition of *M. oryzae* in vitro and in vivo by the transfer of iron to OsiSh-2 as an antagonistic action. What are your thoughts on these previous conclusions?

10. In Figure 6a, the pattern of ROS increase observed by CM-H2DCFDA exhibited a transient ROS peak at 4 hours post-inoculation (hpt) with virulent *M. oryzae* 70-15 in *Oryza sativa* cv. indica 9311 (E-, M+), followed by a typical PTI (pattern-triggered immunity) response showing low ROS level at 24, 48, and 72 hpt. When Δ cchH mutant strain was inoculated, it also displayed a similar transient PTI response as observed in (E-, M+). In contrast, in the case of WT (E+, M+), a stronger and persistent ROS peak was observed from 2 to 6 hpt. This can be attributed to the siderophore secreted by WT OsiSh-2, which enhances PTI as it is recognized as a PAMP (pathogen-associated molecular pattern) or elicitor, unlike the siderophore-deficient Δ cchH mutant strain (reference: Aznar et al., 2015 J. Ex Bot 66: 3001-3010). If you believe that the difference in ROS peak observed in Figure 6a can contribute to increased disease resistance against *M. oryzae*, please explain the reasons behind it.

Reviewer #2 (Remarks to the Author):

This study focuses on a very interesting question, based on previous observations by this research group, that found that the rice endophyte OsiSh-2 showed antagonistic activity against rice blast

disease.

In this report, the authors investigated if OsiSh is triggering moderate ferroptosis, which in turn provides resistance to the disease through an immune priming mechanism.

Although I found the idea very interesting, I have several questions to the authors that I am summarizing below:

1- To identify whether the formation of LMs observed upon OsiSh-2 treatment was related to ferroptosis, the authors state that they measured iron (Fe^{3+}) and ROS levels in the leaves (Fig. 1). However, the experiment seems not clear as i- The region sprayed was not indicated, ii-The nature of the lesions observed is not evident, a magnified image should be shown (is there chlorosis?/yellowish or only brown lesions?) iii- The presence of iron (Fe^{3+}) and ROS was assessed by staining and there is not quantification, so it is not possible to talk about iron or ROS levels here. The same applies for Fig. 2 and Fig. 1S. Also, fluorescence snaps seem to have very different exposure times. These details should be specified in the methods section and exposure should be the same for all treatments. Also cell dead needs to be quantified in each case.

2- Fig. 3 a and b, why are values obtained for E+ and E+ mock so different? As they scales are of different magnitude it is difficult to interpret the figures.

3- The authors state that the erastin treatment did not further enhance the disease resistance capability of E+ mock rice, suggesting that OsiSh-2-triggered ferroptosis was sufficient to induce disease resistance. Here several question emerges, i- Is erastin having an effect on OsiSh-2? (this control is missing), and ii- is ferroptosis sufficient to induce disease resistance? Treatment with erastin without adding OsiSh-2- should answer this question.

4- Authors wanted to confirm that the appropriate induction of ferroptosis is necessary for mutualistic symbiosis between OsiSh-2 and host rice. So rice plants were treated with OsiSh-2 at three different concentrations and ferroptosis progression was monitored by Fe^{3+} and ROS accumulation. Again, no quantitative data and importantly, no data on cell death progression. This need to be done.

5- Fig. 5, again, without assessing cell death there is not confirmation that ferroptosis is taking place. ROS/ Fe^{3+} accumulation are early steps in the pathways but cell death extent needs to be quantified.

6- The conclusion "We demonstrated that, by maintaining the immune-primed state in symbionts, ferroptosis can improve disease resistance against *M. oryzae* infection" seems an overstatement in the context of the studies performed. Ferroptosis is a cell death mechanism and was never measured. The immune-primed status needs further investigation, as ROS accumulation was not quantified and callose deposition seems higher but not earlier.

Reviewer #3 (Remarks to the Author):

The work by Xiong and co-authors entitled "Ferroptosis induction in host rice by endophytic *Streptomyces hygrosopicus* OsiSh-2 is need for mutualism and disease resistance in symbiosis" refers to the description of a new beneficial plant-bacterial endophyte interaction mechanism, called Ferroptosis.

Ferroptosis is a mechanism for the induction of cell death depending on Iron that has been described in detail in mammals, while in plants or other microorganisms it has been little studied, as justified by the authors. Only a few references exist, as far as I know. Therefore, describing this mechanism of interaction between a bacterial endophyte such as *Streptomyces* is something new in this area of research.

Additionally, the article is very well written and the experiments they perform are very strong, with

robust statistical and methodological analysis.

Additionally, working with endophytes is methodologically complicated, so the authors do a very elegant job of detecting and corroborating ferroptosis in their plant-endophyte model.

I just have a few questions and suggestions for the authors:

The first doubt is about the construction of the mutant. As shown in Supplementary Figure no. 3, the Δ cchH mutant strain of *Streptomyces hygroscopicus* OsiSh-2 continues to produce siderophores. The authors did not explain well, or I did not see, what media they used to detect production. They only mention liquid medium. I would suggest that the authors showed another type of evidence to know exactly what percentage of the mutant continues to produce siderophores, and how this affects their results, since much of their work claims that there is an important role of these iron-chelating compounds in the process of ferroptosis.

The second doubt is about the confirmation of the deletion in strain OsiSh-2, please include sequencing results in supplementary data. Also, describe the conjugating *E. coli* strain used and a brief description in Methods, not only to mention it in a reference.

Another suggestion is to use CAS medium if the strain grows well and visualize the production of siderophores. Also, it would be nice to know what kind of siderophore the OsiSh-2 strain produces. This would allow us to understand, at least up to now, and explore in the future whether the molecular diversity of bacterial siderophores could have a more general role in other bacterial endophytes in ferroptosis, or if this is a unique mechanism of *Streptomyces* and its type of siderophore(s).

Minor comments:

Please correct citation in Line 56 (Distéfano et al., 2017).

Another suggestion to better visualize some graphs I suggest deleting the 'arc lines' where there are only two treatments and leaving the percentage that shows the difference. For example, Fig. 2, panels f-i.

Please include the following recent references:

Distéfano, A. M., López, G. A., Bauer, V., Zabaleta, E., & Pagnussat, G. C. (2022). Ferroptosis in plants: regulation of lipid peroxidation and redox status. *Biochemical Journal*, 479(7), 857-866.

Hao, Y. J., Zou, Z. B., Xie, M. M., Zhang, Y., Xu, L., Yu, H. Y., ... & Yang, X. W. (2023). Ferroptosis Inhibitory Compounds from the Deep-Sea-Derived Fungus *Penicillium* sp. MCCC 3A00126. *Marine Drugs*, 21(4), 234.

Reviewer #4 (Remarks to the Author):

Ferroptosis is a type of cell death that is distinct from apoptosis and necrosis, typically accompanied by the accumulation of iron, ROS, and lipid peroxides. Ferroptosis is conserved among different species including animals, protozoan parasites, and bacteria. Emerging evidence shows that ferroptosis is also involved in plant immune and stress responses. The manuscript by Xiong et al., reported a novel mechanism of an endophytic *Streptomyces hygroscopicus*-induced disease resistance, which is linked to ferroptosis in rice plants. Authors showed that upon inoculation with *S. hygroscopicus*, cell death responses, accumulation of ROS and Fe³⁺, and expression of plant immune- and ferroptosis-related genes were induced in rice leaves. Treatments with ferroptosis inducer (erastin) and inhibitor (ferrostatin-1) either enhanced or diminished the accumulation of ROS and Fe³⁺ and the cell death response. Their results further showed that *S. hygroscopicus* triggered-ferroptosis promoted plant growth and prevented rice blast disease development in a siderophore-dependent manner. Siderophore-dependent ferroptosis triggered by *S. hygroscopicus* colonization contributed to the defense priming against the rice blast pathogen infection.

In my view, this is an interesting contribution that uncovers a novel mechanism underlying the endophytic interactions between *S. hygroscopicus* and plant rice, revealing an involvement of a ferroptotic process during *S. hygroscopicus* colonization. The topic is interesting and fits the scope of the journal. Experimental replicates appear adequate, and statistical analyses were performed where appropriate. The work will be of interest to a broad audience with interest in plant-microbe interactions. However, I have several comments for the authors to address. Specific comments on each experimental design and data are described below:

Major comments:

The main argument of this manuscript is that the endophytic colonization by *S. hygroscopicus* triggers a ROS- and Fe³⁺-accumulated cell death response in rice leaves. In Figure 1a and 1b and Figure 2b, visible cell death shown on rice leaves upon *S. hygroscopicus* inoculation appears as randomly distributed brown necrotic spots or blotches (Figure 1a and Figure 2a) that were further revealed by Trypan blue staining (Figure 1b). However, the histological Fe³⁺ staining by Prussian blue is associated with the stomatal apparatus (Figure 1c; Figure 2b, 2d; Figure 4f; and Figure 5b). Similarly, ROS accumulation revealed by staining with a fluorescent probe CM-H₂DCFDA (Figure 1d, Figure 2c, 2e; Figure 4g, Figure 5c; Figure 6a) seems to be mostly associated with the stomatal cell areas. As the ferroptosis is strictly associated with the hallmark of iron and ROS accumulation within the dead cells, further cautious analysis is needed to confirm whether the cell death process and the Fe³⁺ and ROS accumulation are simultaneously occurring within the same cells/tissues.

Furthermore, while the Prussian blue staining is mostly associated with the stomatal apparatus, strong blue staining (Figures 2b and 2d) clearly appears at the pores of stomata, rather than in the guard cells or subsidiary cells. The Prussian blue staining protocol used in this study doesn't provide sufficient resolution and specificity of Fe³⁺ accumulation within the cells/tissues that underlie the cell death triggered by *S. hygroscopicus* colonization. Additional imaging approaches, such as the synchrotron radiation micro X-ray fluorescence (SXRF), will help to examine and identify intracellular iron pools at the cell/tissue level in rice leaves.

Although cell/tissue death on rice leaves was evident by the appearance of visible necrotic lesions or by the detection of trypan blue staining, there was no data to show whether/how these cells/tissues are endophytically infected/colonized by *S. hygroscopicus*. Authors should provide a direct comparison between the *S. hygroscopicus* colonization, accumulation of Fe³⁺ and ROS, and the cell death event in a spatiotemporal manner at the single cell level. Using a fluorescent protein-labeled *S. hygroscopicus* strain or selective staining of *S. hygroscopicus* would help the analysis.

Rice leaves at 48 hours post inoculation (hpi) with *S. hygroscopicus* spores exhibited the lesion mimic symptoms (Figure 1a and 1b; Figure 2a). However, detection for both Fe³⁺ and ROS was performed within 24 hpi where the peak for Fe³⁺ level was detected at 6 hpi. At the similar inoculation stages, transcript profiling for immune-related genes (Figure 1e-h) and ferroptosis gene (Figure 1i) were performed, and GPX4 enzyme activity (Figure 1j) and GSH/GSSG levels (Figure 1k, 1l) were measured at 12 hpi. *Streptomyces* spp. are filamentous organisms that undergo several distinct stages of growth, involving spore germination, aerial hyphae emerge, and production of spore chains. At these early stages within 12 hours post inoculation, does *S. hygroscopicus* infect and colonize the host leaf cells/tissues? What are the developmental stages/forms present within the leaf tissues?

Minor comments:

In Figure 6, the fluorescence signals for ROS detection by CM-H₂DCFDA staining at the late time points (48 and 72 hpi) are likely derived from the background noise, as the subject in focus is out of leaf tissues.

In evaluating the defense responses triggered in rice leaves, authors have chosen OsLOX2 and OsWRKY70 in Figure 1e and 1f and OsPBZ1 and OsLOX2 in Figure 6d and 6e for transcript profiling by RT-qPCR. Why were different genes used in these similar analyses? In addition, the rationale for choosing these specific immune-related genes for qPCR analysis was not well justified.

Transcriptional profiling by choosing more candidate defense genes that decode multiple defense response/signaling pathways or a genome-wide transcriptome profiling would be helpful to interpret the functional readout of the defense responses in the rice leaves upon *S. hygroscopicus* colonization.

In the main text (line 89), it states that "a cell-death phenotype; i.e., the emergence of lesion mimics (LMs), was observed on rice leaves at 24 hours post treatment (hpt) (Fig. 1a)", whereas in the Figure 1 legend, it says that "Image of LMs on rice leaves treated with OsiSh-2 at 48 hpt". Further clarification is needed with regard to the time points on these investigations.

Response to Reviewers

To Reviewer #1:

Comment [1]: In Fig. 1a and 1b, it is not clear from the images alone whether the spots on the rice leaves under the E+ condition (OsiSh-2 inoculated) represent typical ferroptosis accompanied by iron- and ROS-dependent cell death with lipid oxidation, or if they indicate just a ferroptosis-like response. This ambiguity arises because even in processes like PTI where ROS accumulation occurs, there is a possibility of iron accumulation and cell death in some cells, even if it is not typical ferroptosis. Therefore, in addition to the current trypan blue images in Fig. 1b, it would be helpful to include microscopic images with Prussian assay showing iron accumulation in the cells where ferroptotic cell death occurred, similar to Fig. 1c style.

Response [1]: Thank you very much for this comment. We have made many attempts to show iron (Fe^{3+}) accumulation in the rice leaf cells where ferroptotic cell death occurred. However, we realized that the Fe^{3+} and ROS accumulation are early steps of ferroptosis formation, while a large amount of cell death in rice leaves requires a continuous accumulation process. Thus, the phenotype of marked cell death occurrence can only be observed later. This means that it is difficult to show Fe^{3+} accumulation with Prussian staining and cell death with trypan blue staining in rice leaves at the same time. Even so, we have supplemented the magnified images of trypan blue staining as Supplementary Fig. 1a, and described them in the revised manuscript on page 3, lines 91-93 as “In addition, as the magnified images show in Supplementary Fig. 1a, the blue dots, which indicated dead plant cells, were mainly found in and around the stomata and adjacent mesophyll cells.”

In addition, when we monitored Fe^{3+} accumulation in E+ rice leaves in real-time by Prussian staining, we observed the Fe^{3+} accumulation in the stomata at 6 hpt. In the same location, we observed the appearance of brown spots, which can generally represent the cellular senescence or death from 24 hpt (Supplementary Fig. 1b). We added the magnified images of Prussian staining with different magnifications in Supplementary Fig. 1b. The relative description as “Moreover, as the magnified

Prussian staining images show in Supplementary Fig. 1b, the appearance of brown spots, which generally represent the cellular senescence or death, was detected in and around the stomata at 24 hpt in the E+ rice.” was added in the revised manuscript on page 3, lines 98-101. Combining the above supplemented evidence and the results of Fe³⁺ and ROS accumulation associated with the stomatal apparatus (Fig. 1c), the conclusion that OsiSh-2 can trigger ferroptosis in host rice was added as “The fact that the Fe³⁺ and ROS accumulation observed at the early stage and the subsequently detected dead cells were in the same location in the rice leaves, i.e., in and around the stomata and adjacent mesophyll cells, indicates that the Fe³⁺- and ROS-dependent cell death triggered by OsiSh-2 was ferroptosis.” in the revised manuscript on pages 3-4, lines 101-104.

To further verify our conclusion, we used the coculture system of OsiSh-2 and rice suspension cells. Compared with the rice leaves, the suspension cells are the single cells and more sensitive to the external stimuli (i.e., treatment with OsiSh-2), and the occurrence of cell death can be observed more rapidly. We added this experiment and results to the revised manuscript on pages 4-5, lines 128-141 as “We also added a spore suspension of OsiSh-2 (E+) to rice suspension cells at a final concentration of 10⁸ spores mL⁻¹ and then monitored cell death by propidium iodide (PI) staining and Fe³⁺ accumulation by Prussian blue staining. In E+ rice suspension cells, the amount of dead cells increased by 213.64% compared with those of untreated (E-) rice suspension cells at 6 hpt (Supplementary Fig. 3a, b). Meanwhile, Fe³⁺ accumulation was observed only in the E+ rice suspension cells (Supplementary Fig. 3c). Moreover, the addition of the ferroptosis inducer erastin significantly increased the number of dead cells (45.32%) in E+ Erastin+ rice suspension cells, while the ferroptosis inhibitor Fer-1 alleviated cell death by 41.83% in E+ Fer-1+ rice suspension cells compared with E+ mock rice suspension cells (Supplementary Fig. 3a, b). To rule out the possibility that the observed phenomenon was due to the action of the compound erastin or Fer-1 alone in rice cells and OsiSh-2, we supplemented the treatments of erastin and Fer-1 in E- rice or culture medium of OsiSh-2 as controls. However, erastin and Fer-1 had almost no effect on rice cell viability (Supplementary Fig. 3a, b) or OsiSh-2 growth (Supplementary Fig. 3d). Overall, we concluded that OsiSh-2 can induce ferroptotic cell death in host rice.”.

Comment [2]: In Fig. 1c, the image demonstrates that when OsiSh-2 is inoculated, there is a higher accumulation of iron at 6 hpt compared to the non-inoculated condition. However, at 12 and 24 hpt, the iron levels decrease and become similar to those of the non-inoculated rice. This finding makes it difficult to conclude that it leads to typical ferroptosis and suggests a higher possibility of cell recovery rather than the occurrence of typical ferroptotic cell death. If ferroptosis occurs after 24 hpt (eg. 36, 48 hpt), please add microscopic evidences to support this claim.

Response [2]: Thank you very much for this comment. As we mentioned in response [1], the Fe^{3+} and ROS accumulation are early steps in ferroptosis pathways, and the phenotype of cell death requires a continuous accumulation process for observation. Therefore, until 24 hpt, we detected the obvious cell death around the stomata where Fe^{3+} and ROS accumulation were induced at 6 hpt. Meanwhile, we confirmed that cell death and Fe^{3+} accumulation simultaneously occurred at 6 hpt using rice suspension cells treated with OsiSh-2 with or without the ferroptosis inhibitor Fer-1 and inducer erastin (Supplementary Fig. 2). Thus, we have reason to conclude that the Fe^{3+} - and ROS-dependent cell death triggered by OsiSh-2 was typical ferroptosis.

According to the reviewer's suggestion, we added information on Fe^{3+} and ROS accumulation at 36 hpt in Fig. 1c, e and f. At this time, the levels of Fe^{3+} and ROS decreased, and Fe^{3+} accumulation around the stomata almost disappeared. Considering that the cell death, i.e., brown spots that interfere with the images of Fe^{3+} accumulation, was observed from 24 hpt, we did not show the images at 48 hpt. All these results indicated that ferroptosis was suppressed after 24 hpt.

Comment [3]: In Fig. 1k and 1i, the authors wrote in Line 107, 'Compared with that in the (E-) rice, the expression of the GPX4-encoding gene OsGPX4 in the (E+) rice was significantly downregulated at 6 hpt (Fig. 1i), and the enzyme activity of GPX4 decreased by 23.9%.' However, in animals, inhibiting GPX4, a glutathione peroxidase 4, leads to the induction of glutathione-deficiency. In contrast, it is known that plant GPX4 lacks the essential selenocysteine residue, which is crucial for glutathione

oxidation. Consequently, plant GPX4 is incapable of efficiently oxidizing GSH (glutathione)^{1, 2}. Therefore, this line of reasoning requires modification.

Reference:

1. Iqbal, A. et al. Hydroperoxide reduction by thioredoxin-specific glutathione peroxidase isoenzymes of *Arabidopsis thaliana*. *The FEBS journal*. **24**, 5589–5597 (2006).
2. Bela, K., Riyazuddin, R. & Csiszár, J. Plant Glutathione Peroxidases: Non-Heme Peroxidases with Large Functional Flexibility as a Core Component of ROS-Processing Mechanisms and Signalling. *Antioxidants (Basel, Switzerland)*. **8**, 1624 (2022).

Response [3]: This comment is greatly appreciated. Based on the results of the current research, plant GPX4 is indeed incapable of efficiently oxidizing GSH (glutathione). We referred to some relevant literatures^{1, 2}, and found that the index of GSH depletion in plants could be employed as key evidence for the occurrence of ferroptosis in rice. Therefore, we deleted the relevant results and discussions about the GPX4 gene and enzyme activity in our revised manuscript. We retained the results for GSH and GSH+GSSG (oxidized glutathione) (Fig. 1k, l), which showed the decreased levels of GSH and GSH+GSSG in E+ rice and indicated that OsiSh-2 treatment led to an induction of glutathione deficiency.

Reference:

1. Distéfano, A. M. et al. Heat stress induces ferroptosis-like cell death in plants. *J Cell Biol*. **216**, 463-476 (2017).
2. Dangol, S., Chen, Y., Hwang, B. K. & Jwa, N. S. Iron- and Reactive Oxygen Species-Dependent Ferroptotic Cell Death in Rice-*Magnaporthe oryzae* Interactions. *Plant Cell*. **31**, 189-209 (2019).

Comment [4]: In Figure 2b and 2c, when erastin is co-treated with OsiSh-2, a more pronounced increase in ferric ions and ROS burst occurs compared to OsiSh-2 treatment alone. This can be attributed to the blockade of cysteine influx by erastin, leading to glutathione-depletion due to inhibition of glutathione synthesis. However, there are no typical ferroptotic cell death images presented after erastin treatment in this Figure. Please add photographic evidence demonstrating the occurrence of ferroptosis.

Response [4]: Thank you very much for this comment. To present the level of

ferroptotic cell death after erastin treatment, we used the rice suspension cells cocultured with OsiSh-2 (E+) and detected the cell death by PI staining. As we mentioned in response [1], after erastin was added to the E+ rice suspension cells, the number of dead cells significantly increased by 45.32% compared with that of the E+ mock rice suspension cells. We also determined the effect of the ferroptosis inhibitor Fer-1 on the cell death of E+ rice suspension cells and found that Fer-1 treatment alleviated cell death by 41.83%. The related results were added to Supplementary Fig. 3a, b and described in the revised manuscript on pages 4-5, lines 133-141. To rule out the possibility that the observed phenomenon was due to the action of the compound erastin or Fer-1 alone in rice cells and OsiSh-2, we supplemented the treatments of E-Erastin+ and E- Fer-1+ rice in Fig. 2, OsiSh-2 (Erastin+) and OsiSh-2 (Fer-1+) in Supplementary Fig. 3a, d as controls.

Comment [5]: In the case of (E+ mock) in Fig. 3b, the biomass of OsiSh-2 decreased significantly at 12 hpt compared to 6 hpt, but then increased again at 24 hpt. The authors explained this as 'OsiSh-2 hijacked the activity of the immune system of the host rice and colonized successfully' (Line 135). Please provide an explanation of the observed data regarding whether the number of ferroptotic cells in rice continues to increase after 36 or 48 hpt or if the increase in rice ferroptosis cells ceases after successful colonization of OsiSh-2.

Response [5]: Thank you very much for this comment. We have monitored the endophytic biomass of OsiSh-2 in rice in real-time from 0.5-168 hours post treatment (hpt), and only the data from 2-24 hpt were presented in the first version of our manuscript. In this revised manuscript, we present the data of relative OsiSh-2 biomass at 0.5-168 hpt in Fig. 3a. The relevant results were added as “Inevitably, another decrease was found at 48 hpt and relatively lower levels until 168 hpt in the E+ rice, indicating that the overproliferation of endophytes was well inhibited in the OsiSh-2-rice symbiont.” in the revised manuscript on page 5, lines 159-161.

To explain the observed data, we added the following discussion: “There were two peaks of OsiSh-2 accumulation in rice, i.e., 4 and 24 hpt. The mass of OsiSh-2 was then

controlled to be decreased to an appropriate level. This is reasonable, as endophytes and plants adapt to each other and maintain a harmonious symbiosis. Otherwise, the overproliferation of OsiSh-2 would develop as a pathogenic invasion. We believe that the first regulation of OsiSh-2 mass at 6 hpt is attributable to the ferroptosis immune response, which was triggered at 6 hpt. However, it should be noted that the occurrence of ferroptosis is an energy-consuming process, and continuous ferroptosis in rice might inhibit the growth of plants. Indeed, as shown in Fig. 1c, the ferroptosis was suppressed after 24 hpt. Therefore, we surmised that the ferroptosis was functioning as an early immunity response model to endophytic OsiSh-2 in rice. The second regulation to inhibit the increase in OsiSh-2 after 24 hpt may be ascribed to other immune system processes but not to ferroptosis.” in the revised manuscript on page 10, lines 325-336. The study of the other delicate regulation processes between OsiSh-2 and rice is still needed and will be explored in the future.

Comment [6]: In Figure 3h graph, when the ferroptosis inhibitor Fer-1 was applied to completely suppress ferroptosis in (E+ Fer-1 M+), it resulted in only a 35.35% increase in blast disease lesion length compared to (E+ M+), while the fungal biomass increased by only 38% (Line 150). However, if ferroptosis induced by E+ is indeed a key factor in disease resistance against *M. oryzae*, then treating with Fer-1 to completely inhibit ferroptosis should lead to a further increase in lesion length and fungal biomass, approaching the levels of E-, but instead, we observe a limited increase of only 35-38%. This indicates that ferroptosis induced by OsiSh-2 (E+) only partially contributes to the increase in rice disease resistance against *M. oryzae*. These results suggest that factors other than ferroptosis significantly contribute to rice disease resistance. One such factor could be iron chelation by siderophores secreted by OsiSh-2 (E+) in the symbiotic relationship between OsiSh-2 (E+) and rice. In fact, Figure 5h shows that even with a mere 32.5% reduction in siderophore secretion compared to the WT OsiSh-2 strain, when the $\Delta cchH$ mutant strain was inoculated, the *M. oryzae* fungal biomass increased significantly by 40.04% compared to WT OsiSh-2 treatment (Line 201). If the $\Delta cchH$ mutant strain, which has a 100% reduction in siderophore secretion compared to the

WT OsiSh-2 strain, were used, it can be anticipated that the *M. oryzae* fungal biomass would increase even more significantly than the current 40.04%. Therefore, when considering the results of these two experiments (Fig. 3h, i and Fig. 5g, h) together, they collectively demonstrate that siderophores secreted by OsiSh-2 strain contribute significantly more to rice disease resistance against *M. oryzae* than the ferroptosis formed during the symbiotic process of the WT OsiSh-2 strain in rice. This indicates a flaw in the logic proposed by the authors. Please provide a response to this comment.

Response [6]: This comment is greatly appreciated. First, as a commonly used ferroptosis inhibitor, Fer-1 obviously inhibited the ferroptosis occurrence triggered by OsiSh-2. However, Fer-1 did not completely suppress ferroptosis in E+ Fer-1+ rice, as 1) small amounts of Fe³⁺ and ROS accumulation were still detected (Fig. 2f-i); 2) the content of MDA, a typical lipid peroxidation marker, was reduced by 24.81% in E+ Fer-1+ rice compared with E+ mock rice but was still present at higher levels than that in E- Fer-1+ by 24.11% (Fig. 2l); and 3) when compared with that in E- rice suspension cells, the number of dead cells in E+ mock rice and in E+ Fer-1+ rice significantly increased by 213.64% and 82.46% respectively, and the latter was still increased by 74.20% compared with that in E- Fer-1+ rice. (Supplementary Fig. 3b). Thus, it was reasonable that only a 35.35% increase in blast disease lesion length and a 38.22% increase in fungal biomass were detected in E+ Fer-1+ M+ rice when compared to E+ mock M+ rice.

In addition, for animal cells, 0.5-2 μM is a commonly used concentration range for Fer-1, and even lower concentrations, for instance, 60 nM Fer-1 can suppress the erastin-induced ferroptosis in HT-1080 cells¹. For plant and microbe cells, the working concentration of Fer-1 is normally approximately 1-10 μM . Treatment with 10 μM Fer-1 has been found to suppress lipid peroxidation in rice leaf sheaths infected with avirulent *M. oryzae* with the MDA content decreasing by approximately 50%². In cyanobacteria, 1 μM Fer-1 decreased the cell death triggered by heat stress by approximately 40%³. In *M. oryzae*, treatment with 5-10 μM Fer-1 triggered the formation of normal hyphal structures, and the number of dead conidial cells was reduced by approximately 30%⁴. In this study, the working concentration of Fer-1 was

10 μ M, which had no effect on OsiSh-2 growth (Supplementary Fig. 3d). Perhaps a concentration of Fer-1 greater than 10 μ M might completely suppress ferroptosis in E+ rice. However, to avoid a negative effect on the normal growth of OsiSh-2, we only used 10 μ M Fer-1.

For the disease control mode of OsiSh-2, we fully agree with the reviewer's opinion that disease resistance in host rice induced by OsiSh-2 depends on multiple mechanisms, and ferroptosis is just one of them. Thus, we supplemented the discussion as "Notably, other modes of OsiSh-2 action against rice blast disease exist in host rice, including secreting metabolites for directly antagonizing pathogens. For example, secreting fungal cell wall lytic enzymes (such as chitinase and β -1,3-glucanase) and antibiotics such as nigericin have been found to help OsiSh-2 to directly antagonize *M. oryzae* by damaging fungal cell membrane integrity. Competing for nutrition with *M. oryzae* through siderophore production in vitro can inhibit the growth of *M. oryzae*. In this study, we found that moderate ferroptosis is necessary for establishing an optimal mutualistic symbiotic relationship between OsiSh-2 and rice, which consequently activates an immune resistance to rice blast disease. This induction of an immune response for indirect disease resistance is not the only contribution of ferroptosis. Our recent study showed that applying ShAM1, an α -mannosidase purified from OsiSh-2 culture supernatants, can trigger rice immune responses and improve blast resistance with a 45.9% reduction in blast disease lesion length and a 67% reduction in fungal biomass. Therefore, we considered that the assistance of OsiSh-2 in disease resistance in host rice depends on multiple mechanisms, and ferroptosis is one of them." in the revised manuscript on page 12, lines 382-395.

To further investigate whether the siderophores secreted by the OsiSh-2 strain contribute significantly more to rice disease resistance against *M. oryzae* than the ferroptosis formed during the symbiotic process of OsiSh-2 in rice, we added the *Streptomyces*-derived siderophore deferoxamine (DFO) when OsiSh-2 was used to treat rice plants. The results were added as "To determine the role of siderophores in disease resistance, we added the *Streptomyces*-derived siderophore deferoxamine (DFO) to rice plants treated with OsiSh-2. The addition of DFO obviously inhibited Fe^{3+} and ROS accumulation in E+ (WT) rice (Supplementary Fig. 7a-d), and the disease resistance

against *M. oryzae* was significantly reduced (Supplementary Fig. 7e-g). This is consistent with the fact that DFO serves as a ferroptosis inhibitor. However, when DFO was applied in $\Delta cchH$, the Fe^{3+} and ROS accumulation was partially rescued at 6 hpt in E+ ($\Delta cchH$) DFO+ rice, and the blast resistance effect was improved close to that of E+ (WT) mock rice (Supplementary Fig. 7). This result indicated that when the occurrence of ferroptosis was restricted in E+ ($\Delta cchH$) cells, an appropriate amount of DFO could help induce ferroptosis, although the ferroptosis level could not be rescued to that in E+ (WT) cells. Notably, the single treatment of DFO did not influence the disease resistance of the rice plants, as both lesion length and relative fungal biomass in E- DFO+ M+ rice were similar to those in E- mock M+ rice (Supplementary Fig. 7e-g). That is, DFO might play a subtle role in the regulation of ferroptosis in rice rather than directly contributing to rice blast resistance. The above results further confirmed that the moderate ferroptosis triggered by OsiSh-2 (WT) is necessary for improving rice disease resistance against *M. oryzae*. Although siderophore coelichelin would have been more appropriate for this experiment, the compound is not commercially available, and we were unable to purchase it for use in this study.” in the revised manuscript on page 8, lines 246-262.

Reference:

1. Dixon, S. J. et al. Ferroptosis: an iron-dependent form of nonapoptotic cell death. *Cell*. **149**, 1060-1072 (2012).
2. Dangol, S., Chen, Y., Hwang, B. K. & Jwa, N. S. Iron- and Reactive Oxygen Species-Dependent Ferroptotic Cell Death in Rice-*Magnaporthe oryzae* Interactions. *Plant Cell*. **31**, 189-209 (2019).
3. Distéfano, A. M. et al. Heat stress induces ferroptosis-like cell death in plants. *J Cell Biol*. **216**, 463-476 (2017).
4. Shen, Q., Liang, M. L., Yang, F., Deng, Y. Z. & Naqvi, N. I. Ferroptosis contributes to developmental cell death in rice blast. *New Phytol*. **227**, 1831-1846 (2020).

Comment [7]: In Figure 5b, the result of $\Delta cchH$ mutant strain inoculation suggests that the reduction in chelation due to siderophore mutation leads to increased iron uptake by *M. oryzae*, resulting in the restoration of its pathogenicity. In other words, the observed increase in disease resistance against *M. oryzae*, as claimed in the paper, may be partially attributed to ferroptosis, but it could also be the result of the stronger iron

chelation effect exerted by the siderophore secreted by OsiSh-2. To demonstrate this, additional experiments should be conducted by treating the $\Delta cchH$ mutant strain, which is a siderophore mutant, with exogenous coelichelin treatment and observing the changes that occur. Please add these experimental results to the Figures.

Response [7]: We agree with the reviewer's viewpoint about the contribution of disease resistance against *M. oryzae* by the strong iron chelation effect exerted by the siderophore secreted by OsiSh-2. In our previous study, we confirmed that OsiSh-2 effectively inhibits the growth of the pathogenic *M. oryzae* by secreting siderophores for competing iron *in vitro*¹. According to the reviewer's suggestion, we added exogenous siderophore treatment and observed the changes that occurred. OsiSh-2 can produce the same type of siderophores, i.e., DFO and coelichelin. We obtained a coelichelin synthetic deletion mutant of OsiSh-2 but failed to obtain a DFO-related mutant, as DFO might be essential for the normal growth of OsiSh-2. Unfortunately, coelichelin is not commercially available, and we were therefore unable to purchase it. The *Streptomyces*-derived siderophore DFO was chosen as an alternative. As we mentioned in response [6], the single treatment of DFO only activated a low level of ROS accumulation, but no accumulation of Fe³⁺ was detected, and the disease resistance of E- DFO+ M+ rice plants was similar to that in E- mock M+ rice (Supplementary Fig. 7e, f). In addition, if the disease resistance against *M. oryzae* was mainly attributed to the strong iron chelation effect exerted by the siderophore secreted by OsiSh-2, the addition of DFO would further enhance the disease resistance of E+ (WT) M+ rice. However, we did not observe this phenomenon. In contrast, in E+ (WT) DFO+ M+ rice, both Fe³⁺ and ROS accumulation and the disease resistance against *M. oryzae* were significantly reduced (Supplementary Fig. 7). Under this condition, DFO serves as an inhibitor of ferroptosis. Thus, our current study suggested that the role of DFO in directly contributing to rice blast resistance is limited. The effect of other siderophores produced by OsiSh-2 may be stronger than that of DFO in the inhibition of rice disease. We are now trying to isolate and identify siderophores from OsiSh-2.

Reference:

1. Zeng, J. R. et al. The Role of Iron Competition in the Antagonistic Action of the Rice Endophyte *Streptomyces sporocinereus* OsiSh-2 Against the Pathogen *Magnaporthe oryzae*. *Microb Ecol.* **76**, 1021-1029 (2018).

Comment [8]: The results in Figure 5f demonstrate a strong correlation between the extent of iron uptake inhibition by siderophore into *M. oryzae*, as observed in Shen et al., 2020 (*New Phytologist* 227: 1831-1846), and the decrease in fungal biomass (32.92%) (Figure 5h), which aligns closely with the siderophore production levels of OsiSh-2 (WT: 100%; $\Delta cchH$ strain: 32.5% lower than WT siderophore secretion). This finding suggests that rice disease resistance increases proportionally with siderophore production. Additionally, in Line 316, the authors also hypothesize that siderophore itself could serve as an elicitor, activating plant immunity. Despite this, it remains unclear why the authors attribute defense priming to ferroptosis. Please provide an explanation for this inconsistency.

Response [8]: Thank you very much for this comment. As we mentioned in response [6], various modes of action OsiSh-2 exist when against the rice blast disease in host rice, including secreting metabolites to directly antagonize pathogens and evoking an immune response to indirectly enhance disease resistance. Among these factors, siderophores are important for antagonizing *M. oryzae* by iron competition and plant immunity elicitation. In this study, the siderophore production level of $\Delta cchH$ decreased by 32.51% compared with that of OsiSh-2(WT) (Supplementary Fig. 6d). However, the disease resistance capability of the E+ ($\Delta cchH$) M+ rice was reduced with a longer disease lesion length (13.71%) and a higher relative fungal biomass (40.04%) than those of the E+ (WT) M+ rice (Fig. 5i, j). In addition, the dual-culture assay of OsiSh-2/ $\Delta cchH$ and *M. oryzae* showed that the fungal growth inhibition rate of $\Delta cchH$ was only 26.21% lower than that of OsiSh-2(WT) (Supplementary Fig. 6j, k). Therefore, the increase in rice disease resistance increment was not proportionally correlated with the siderophore production, indicating that other defence mechanisms simultaneously functioned. The result of the dual-culture assay was supplemented as Supplementary Fig. 6j, k and described as “In addition, the dual-culture assay for

OsiSh-2/ $\Delta cchH$ and *M. oryzae* showed that the fungal growth inhibition rate of $\Delta cchH$ was only 26.21% lower than that of OsiSh-2 (WT) (Supplementary Fig. 6j, k).” in the revised manuscript on page 8, lines 241-243. Another piece of evidence is that when we treated rice plants with OsiSh-2 at three different concentrations (10^7 , 10^8 , and 10^9 spores mL⁻¹), the biomass of OsiSh-2 in rice was correspondingly enhanced with increasing inoculation concentration. Normally, more OsiSh-2 indicated more siderophores produced, however, the disease resistance effect was best in E+ (10^8) M+ rice, not in E+ (10^9) M+ rice. Therefore, we still considered that, in addition to siderophores, the ferroptosis triggered by OsiSh-2 is necessary for improving rice disease resistance against *M. oryzae*.

In this study, we hypothesized that defence priming is related to the OsiSh-2-triggered ferroptosis in host rice. In our previous study, the induction of defence priming was shown to be an important disease resistance mechanism mediated by OsiSh-2, through which the symbiont can effectively and low-costly resist the invasion of *M. oryzae*, and maintain the balance of defence and normal growth¹. Considering that ferroptosis is also an important mechanism in immune regulation, we wondered whether there is a relationship between defence priming and ferroptosis. The normal development of *M. oryzae* conidia at approximately 2-12 hpi on the surface of rice leaves is a critical preparatory period for successful pathogenic infection in the rice cells. *M. oryzae* conidia need to quickly germinate (1-2 hpi) and then form a dome-shaped infection structure at the tip called an appressorium with high internal turgor (generally 4-12 hpi), which is required for cuticle penetration. In this study, we showed that in E+ (WT) M+ rice, the ROS and callose accumulation, which indicate the early defence response against *M. oryzae*, occurred at 0.5 hpi and 2 hpi, respectively, while ROS and callose accumulation occurred at 4 hpi and 12 hpi in both E- M+ and E+ ($\Delta cchH$) M+ rice (Fig. 6a-d). The rapid accumulation of ROS and callose in E+ (WT) M+ rice presented a typical defence priming response pattern. Compared with the later recognition of *M. oryzae* by E- M+ or E+ ($\Delta cchH$) M+ cells after the conidia germinate or even the appressorium forms, the activation of priming should be a more timely and cost-efficient strategy to defend against pathogen attack. Therefore, the fact that the

siderophore mutant strain $\Delta cchH$, which could not trigger the occurrence of ferroptosis in rice, failed to initiate the priming state confirmed the relationship between ferroptosis and priming. Even so, we are not sure that there is a direct connection between ferroptosis and priming. Perhaps siderophores can also trigger priming. The mechanisms need to be further studied. The conclusion “These results indicated that the occurrence of effective ferroptosis in OsiSh-2-treated rice may be correlated with priming induction in the symbiont under biotic stress, consequently enhancing disease resistance. The related mechanism still requires further study.” in the revised manuscript on page 9, lines 278-280.

Reference:

1. Gao, Y. et al. Endophytic *Streptomyces hygroscopicus* OsiSh-2-Mediated Balancing between Growth and Disease Resistance in Host Rice. *mBio*. **12**, e0156621 (2021).

Comment [9]: In Figure 5f, the increased lesion length of *M. oryzae* in rice leaves infected with $\Delta cchH$ mutant strain compared to OsiSh-2 WT can also be attributed to the siderophore produced by WT OsiSh-2, which prevents the uptake of ferric ions by *M. oryzae*, leading to subsequent reduction in *M. oryzae* virulence. In other words, this is unrelated to rice ferroptosis induced by OsiSh-2. This finding aligns with a previous study by the same group in 2018 (Zeng et al., 2018 *Microbial Ecology*, <https://doi.org/10.1007/s00248-018-1189-x>), where they explained the exceptional iron acquisition competitiveness of OsiSh-2 compared to *M. oryzae*, resulting in the effective inhibition of *M. oryzae* in vitro and in vivo by the transfer of iron to OsiSh-2 as an antagonistic action. What are your thoughts on these previous conclusions?

Response [9]: Here, we also agree with the reviewer’s viewpoint about the important contribution of siderophores in antagonizing *M. oryzae*. However, we also consider that the disease resistance improved by OsiSh-2 is partially related to rice ferroptosis induced by OsiSh-2. The reasons are listed below: 1) As we mentioned in response [8], the increases in rice disease resistance were not proportionally correlated with siderophore production, indicating that other defence mechanisms (including ferroptosis) were simultaneously at play. 2) Applying the specific ferroptosis inducer

erastin (E+ Erastin+) and inhibitor Fer-1 (E+ Fer-1+) confirmed that the OsiSh-2-induced immune response in E+ rice is ferroptosis (Fig. 2). Further *M. oryzae* infection revealed that the disease resistance was related to the OsiSh-2-induced appropriate ferroptosis. As in E+ Erastin+ M+ rice, the disease resistance improvement was similar to that of E+ mock M+ rice (Fig. 3d-f), while in E+ Fer-1+ M+ rice, the disease resistance was significantly lower than that of E+ mock M+ rice (Fig. 3g-i). 3) As we mentioned in response [6], disease-resistant results from exogenous siderophore DFO-treated E+ rice suggested that this kind of siderophore might play a subtle role in the regulation of ferroptosis in rice rather than directly contributing to rice blast resistance.

Comment [10]: In Figure 6a, the pattern of ROS increase observed by CM-H₂DCFDA exhibited a transient ROS peak at 4 hours post-inoculation (hpt) with virulent *M. oryzae* 70-15 in *Oryza sativa* cv. indica 9311 (E- M+), followed by a typical PTI (pattern-triggered immunity) response showing low ROS level at 24, 48, and 72 hpt. When $\Delta cchH$ mutant strain was inoculated, it also displayed a similar transient PTI response as observed in (E- M+). In contrast, in the case of WT (E+ M+), a stronger and persistent ROS peak was observed from 2 to 6 hpt. This can be attributed to the siderophore secreted by WT OsiSh-2, which enhances PTI as it is recognized as a PAMP (pathogen-associated molecular pattern) or elicitor, unlike the siderophore-deficient $\Delta cchH$ mutant strain (reference: Aznar et al., 2015 J. Ex Bot 66: 3001-3010). If you believe that the difference in ROS peak observed in Figure 6a can contribute to increased disease resistance against *M. oryzae*, please explain the reasons behind it.

Response [10]: Thank you very much for this comment. The early recognition of pathogens is beneficial for plants to make a timely defence response. When exposed to *M. oryzae*, a more rapid accumulation of ROS and callose in E+ M+ rice than in E- M+ rice indicated that defence priming was triggered by OsiSh-2. The priming strategy used by OsiSh-2 obviously improved the pathogenic recognition and response capability in the OsiSh-2-rice symbiont. Based on our previous studies, this defence model represents a typical endophyte-mediated modulation of disease resistance and fitness in the host plant. As in priming-state rice, the continuous expression of energy-consuming

defence-related proteins was controlled, and the saved energy might be used to maintain the growth of rice under the pathogen stress.

We agree with the reviewer's comment that the stronger and persistent ROS peak observed from 2 to 6 hpt can be attributed to the siderophore secreted by OsiSh-2(WT), but we have no direct evidence to reveal this. As we mentioned in response [7], when we exogenously added DFO, a common siderophore produced in *Streptomyces*, we only observed a basal ROS burst in E- DFO+ rice, but the disease resistance against *M. oryzae* was not detected. However, this does not rule out the possibility that the siderophore secreted by OsiSh-2(WT) might enhance PTI, as it is recognized as a PAMP (pathogen-associated molecular pattern) or elicitor. The related mechanism needs to be further studied. We added the related discussion as follows: "In Fig. 6a, the pattern of ROS increase observed by CM-H₂DCFDA exhibited a transient ROS peak at 4 hpt with virulent *M. oryzae* in E- M+ rice, followed by a typical pattern-triggered immunity (PTI) response showing low ROS levels at 24, 48, and 72 hpt. When the $\Delta cchH$ mutant strain was inoculated, it also displayed a similar transient PTI response to that observed in E- M+ rice. In contrast, in the case of E+ (WT) M+ rice, an earlier and stronger ROS peak was observed from 0.5 to 6 hpt. Similarly, callose deposition around the stomata was observed in E+ (WT) M+ rice leaves at least 4 h earlier than that in E- M+ rice. The more rapid and strong accumulation of ROS and callose in E+ (WT) M+ than in E- M+ rice indicates that defence priming was triggered by OsiSh-2. Based on our previous studies, the priming strategy used by OsiSh-2 markedly improved the pathogenic recognition and response capability in the OsiSh-2-rice symbiont. To date, this is the first report on the relationship between ferroptosis and priming. On the other hand, compared to the siderophore-deficient $\Delta cchH$ mutant strain, the accumulation of ROS in E+ (WT) M+ rice might be attributed to the OsiSh-2-secreted siderophore, which enhances PTI as it is recognized as a PAMP (pathogen-associated molecular pattern) or elicitor. However, when we added the DFO exogenously, a common siderophore produced in *Streptomyces*, we only observed an ROS burst in E- DFO+ rice, but the disease resistance against *M. oryzae* was not detected. Other siderophores produced by OsiSh-2 may act as immunity elicitors, and the detailed mechanisms of action need to

be further elucidated” in the revised manuscript on pages 11-12, lines 364-381.

To Reviewer #2:

Comment [1]: To identify whether the formation of LMs observed upon OsiSh-2 treatment was related to ferroptosis, the authors state that they measured iron (Fe^{3+}) and ROS levels in the leaves (Fig. 1). However, the experiment seems not clear as i- The region sprayed was not indicated, ii-The nature of the lesions observed is not evident, a magnified image should be shown (is there chlorosis?/yellowish or only brown lesions?) iii- The presence of iron (Fe^{3+}) and ROS was assessed by staining and there is not quantification, so it is not possible to talk about iron or ROS levels here. The same applies for Fig. 2 and Fig. 1S. Also, fluorescence snaps seem to have very different exposure times. These details should be specified in the methods section and exposure should be the same for all treatments. Also cell dead needs to be quantified in each case.

Response [1]: Thank you very much for this comment. We sprayed the spore suspension (10^8 spores mL^{-1}) of OsiSh-2 evenly on the surface of rice leaves using a mini spray bottle atomizer (20 mL/50 plants). Thus, the LMs observed were randomly distributed on the OsiSh-2-treated (E+) rice leaves. We added the description “We sprayed the spore suspension (10^8 spores mL^{-1}) of OsiSh-2 with 0.2% (v/v) Tween 20 evenly on the surface of rice leaves using a mini spray bottle atomizer (20 mL/50 plants).” in the “Methods” in our revised manuscript on page 15, lines 480-482. In addition, we modified Fig. 1a, which shows the LMs distributed on the whole rice leaf and a magnified image of LMs. As shown in Fig. 1a, we can see that most of the LMs were reddish-brown spot-like lesions. We revised the description of LM formation as “When OsiSh-2 (10^8 spores mL^{-1}) was sprayed evenly onto the leaves of the rice seedlings (E+), a phenotype of marked cell-death occurrence, i.e., the emergence of lesion mimics (LMs), which appear as reddish-brown spot-like lesions, was observed on rice leaves at 24 hours post treatment (hpt) (Fig. 1a).” in the revised manuscript on page 3, lines 87-90.

According to the reviewer’s suggestion, the presence of Fe^{3+} and ROS assessed by staining in each case were quantified and added to revised Fig. 1, Fig. 2, Fig. 4, Fig. 5,

Fig.6, Supplementary Fig. 2, and Supplementary Fig. 7. For instance, in Fig. 1, Fe^{3+} accumulation is indicated by the Prussian blue staining intensity (Fig. 1c), and the phenotype ratio (%) indicates the proportions of designated staining phenotypes (Fig. 1d). The ROS burst is indicated by the CM- H_2DCFDA staining intensity (Fig. 1e), and the relative fluorescence intensity indicates the average relative fluorescence intensity units of twenty parts calculated via ImageJ (Fig. 1f).

Considering that the cell death in rice leaves is a gradual process and difficult to quantify, we quantified the cell death triggered by OsiSh-2 by using the rice suspension cells after propidium iodide (PI) staining. PI staining can indicate whether the cells are alive (cell wall staining) or dead (nuclear staining). PI signals in nuclei of stained cells (%) indicate the proportions of dead cells in all observed cells. The related results were added to Supplementary Fig. 3a, b. The related description was added as follows: “We also added a spore suspension of OsiSh-2 (E+) to rice suspension cells at a final concentration of 10^8 spores mL^{-1} and then monitored cell death by propidium iodide (PI) staining and Fe^{3+} accumulation by Prussian blue staining. In E+ rice suspension cells, the amount of dead cells increased by 213.64% compared with those of untreated (E-) rice suspension cells at 6 hpt (Supplementary Fig. 3a, b). Meanwhile, Fe^{3+} accumulation was observed only in the E+ rice suspension cells (Supplementary Fig. 3c).” in the revised manuscript on page 4, lines 128-133.

For the fluorescence exposure times, we used an EVOS™ M5000 Imaging System (Thermo Fisher Scientific, USA) for fluorescence observation in rice leaves. To ensure the consistency of the device parameters at the same magnification, fluorescence photos of rice leaves in different treatments were taken under the same light intensity. The relevant parameters are as follows: light:1.866; exposure:30 ms; gain:15 dB. Therefore, although the fluorescence intensities of E+ Erastin+ and E+ FeCl_3+ rice in Fig. 2 and Supplementary Fig. 2 were much higher than those of E+ rice, we did not decrease the light intensity or change the exposure times. The related description was added to the “Method” as follows: “To ensure the consistency of the device parameters at the same magnification, the fluorescence photos of rice leaves in different treatments were taken under the same light intensity. The relevant parameters were as follows: light:1.866;

exposure:30 ms; and gain:15 dB.” in the revised manuscript on page 14, lines 451-454.

Comment [2]: Fig. 3a and b, why are values obtained for E+ and E+ mock so different? As they scales are of different magnitude it is difficult to interpret the figures.

Response [2]: This comment is greatly appreciated. The relative biomass of OsiSh-2 in rice was calculated using the threshold cycle (C_T) of *ShRpoA* (a DNA-directed RNA polymerase subunit alpha of OsiSh-2) DNA against the C_T of *OsUbq* (a rice genomic ubiquitin gene) DNA as a ratio ($ShRpoA/OsUbq$), represented by the equation $2^{CT(OsUbq) - CT(ShRpoA)}$. This method referred to the commonly used DNA-based qPCR assay for relative quantification of the endophyte or pathogen within its host plants. The assay has been confirmed to be highly specific and suitable to reliably detect amounts of endophytes in host plants^{1,2}. Because this method is a relatively quantitative measurement, the values of the same treatment might be different among different experimental replications. We repeated the experiment three times, and the results are shown in the figure below. The values of relative OsiSh-2 biomass in E+ mock and E+ Erastin+ rice from different experimental replications were different. However, the tendency of change in relative OsiSh-2 biomass in the same monitoring period from different experimental replications was consistent, especially at 6-24 hpt, which showed a similar trend to that in Fig.3a-c, Fig. 4f, and Fig. 5a. To ensure clarity, we replaced Fig. 3b with the figure below (c) instead of the figure below (a) in the revised manuscript. The methods for determining the biomass of OsiSh-2 were added as “The relative biomass of OsiSh-2 in rice was calculated using the threshold cycle (C_T) of *ShRpoA* (a DNA-directed RNA polymerase subunit alpha of OsiSh-2) DNA against the C_T of *OsUbq* (a rice genomic ubiquitin gene) DNA as a ratio ($ShRpoA/OsUbq$), represented by the equation $2^{CT(OsUbq) - CT(ShRpoA)}$.” in the revised manuscript on page 16, lines 519-522.

Fig. Endophytic colonization in the OsiSh-2-rice symbiont in three independent experiments.

The relative biomass of OsiSh-2 was calculated via the threshold cycle value (CT) of *ShRpoA* DNA (a DNA-directed RNA polymerase subunit alpha of OsiSh-2) versus the CT of *OsUbq* DNA (a rice genomic ubiquitin gene) by DNA-based qRT-PCR at the indicated hours post treatment in E-, E+ mock, and E+ Erastin+ rice.

Reference

1. Park, C. H. et al. The *Magnaporthe oryzae* Effector AvrPiz-t Targets the RING E3 Ubiquitin Ligase APIP6 to Suppress Pathogen-Associated Molecular Pattern-Triggered Immunity in Rice. *Plant Cell*. **24**, 4748–4762 (2012).
2. Kawano Y. et al. Activation of a Rac GTPase by the NLR family disease resistance protein pit plays a critical role in rice innate immunity. *Cell host & microbe*. **7**, 362-375 (2010).

Comment [3]: The authors state that the erastin treatment did not further enhance the disease resistance capability of E+ mock rice, suggesting that OsiSh-2-triggered ferroptosis was sufficient to induce disease resistance. Here several question emerges, i- Is erastin having an effect on OsiSh-2? (this control is missing), and ii- is ferroptosis sufficient to induce disease resistance? Treatment with erastin without adding OsiSh-2- should answer this question.

Response [3]: According to the reviewer’s suggestion, we supplemented the erastin and Fer-1 control to reveal whether they have effects on OsiSh-2. We collected the mycelia of OsiSh-2 daily after inoculation into ISP2 liquid medium without or with erastin/Fer-1 for 10 days and assessed the growth of OsiSh-2. We plotted the growth curves of OsiSh-2 in culture media and found that the addition of erastin and Fer-1 had almost no effect on OsiSh-2 growth (Supplementary Fig. 3d). When we treated the seedlings and

suspension cells of E- rice with erastin/Fer-1 at the same concentration as that in E+ rice, no phenotype of ferroptosis was detected in E- Erastin+/Fer-1+ rice leaves (added in Fig. 2a) or suspension cells (added as Supplementary Fig. 1a, b). The description of that “To rule out the possibility that the observed phenomenon was due to the action of the compound erastin or Fer-1 alone in rice cells and OsiSh-2, we supplemented the treatments of erastin and Fer-1 in E- rice or culture medium of OsiSh-2 as controls. However, erastin and Fer-1 had almost no effect on rice cell viability (Supplementary Fig. 3a, b) or OsiSh-2 growth (Supplementary Fig. 3d).” was added to our revised manuscript on pages 4-5, lines 136-140.

After further infection with *M. oryzae*, the lesion length and relative fungal biomass of E- Erastin+/Fer-1+ M+ rice were also not significantly different from those of E- mock M+ rice (added in Fig. 3d-i). However, in E+ Erastin+/Fer-1+ M+ rice, we found that erastin treatment did not further enhance the disease resistance capability of E+ mock M+ rice, while Fer-1 treatment significantly reduced the disease control effect of OsiSh-2. These results indicated that appropriate ferroptosis was sufficient to induce disease resistance.

For the disease control mode of OsiSh-2, we believe that disease resistance in host rice induced by OsiSh-2 depends on multiple mechanisms, and ferroptosis is just one of them. We supplemented the discussion as follows: “Notably, other modes of OsiSh-2 action against rice blast disease exist in host rice, including secreting metabolites for directly antagonizing pathogens. For example, secreting fungal cell wall lytic enzymes (such as chitinase and β -1,3-glucanase) and antibiotics such as nigericin have been found to help OsiSh-2 to directly antagonize *M. oryzae* by damaging fungal cell membrane integrity. Competing for nutrition with *M. oryzae* through siderophore production in vitro can inhibit the growth of *M. oryzae*. In this study, we found that moderate ferroptosis is necessary for establishing an optimal mutualistic symbiotic relationship between OsiSh-2 and rice, which consequently activates an immune resistance to rice blast disease. This induction of an immune response for indirect disease resistance is not the only contribution of ferroptosis. Our recent study showed that applying ShAM1, an α -mannosidase purified from OsiSh-2 culture supernatants,

can trigger rice immune responses and improve blast resistance with a 45.9% reduction in blast disease lesion length and a 67% reduction in fungal biomass. Therefore, we considered that the assistance of OsiSh-2 in disease resistance in host rice depends on multiple mechanisms, and ferroptosis is one of them.” in the revised manuscript on page 12, lines 382-395.

Comment [4]: Authors wanted to confirm that the appropriate induction of ferroptosis is necessary for mutualistic symbiosis between OsiSh-2 and host rice. So rice plants were treated with OsiSh-2 at three different concentrations and ferroptosis progression was monitored by Fe^{3+} and ROS accumulation. Again, no quantitative data and importantly, no data on cell death progression. This need to be done.

Response [4]: Thank you very much for this comment. As we mentioned in response [1], we have added the quantitative data of Fe^{3+} and ROS in all cases. In the case of rice plants treated with OsiSh-2 at three different concentrations, we added the quantitative data of Fe^{3+} and ROS as Fig. 4i, j in our revised manuscript. We have replaced “Notably, when further analysing the ferroptosis progress indicated by the accumulation of Fe^{3+} and ROS, we found that only OsiSh-2 (10^8 spores mL^{-1}) could trigger the simultaneous accumulation of Fe^{3+} (59.06% strong) and ROS (105 relative fluorescent intensity) at 6 hpt (Fig. 4g-j). However, at the same time, in both E+ (10^7 and 10^9) rice, the accumulation of Fe^{3+} (11.19% and 20.78% strong ones, respectively) and ROS (82 and 62 relative fluorescent intensity, respectively) were decreased. Consistently, when the above three concentrations of OsiSh-2 spores were added to the rice suspension cells, PI staining revealed that, the dead cells in E+ (10^7 and 10^9 spores mL^{-1}) rice suspension cells were significantly decreased by 37.06% and 32.07%, respectively, at 6 hpt when compared with those of E+ (10^8 spores mL^{-1}) rice suspension cells (Supplementary Fig. 4).” in the revised manuscript on page 6, line 184-193.

For the qualification of cell death, we also quantified OsiSh-2-triggered cell death using rice suspension cells as we mentioned in response [1]. We added the related results as Supplementary Fig. 4, and described them as follows “Consistently, when the above three concentrations of OsiSh-2 spores were added to the rice suspension cells,

PI staining revealed that, the dead cells in E+ (10^7 and 10^9 spores mL^{-1}) rice suspension cells were significantly decreased by 37.06% and 32.07%, respectively, at 6 hpt when compared with those of E+ (10^8 spores mL^{-1}) rice suspension cells (Supplementary Fig. 4).” in revised manuscript on page 6, lines 189-193.

Comment [5]: Fig. 5, again, without assessing cell death there is not confirmation that ferroptosis is taking place. ROS/ Fe^{3+} accumulation are early steps in the pathways but cell death extent needs to be quantified.

Response [5]: Thank you very much for this suggestion. We realized that Fe^{3+} and ROS accumulation are early steps of ferroptosis formation, while a large amount of cell death in rice leaves requires a continuous accumulation process. Thus, the phenotype of cell death can only be observed latter. This means that it is difficult to show and quantify Fe^{3+} accumulation with Prussian staining and cell death with trypan blue staining in rice leaves at the same time. To quantify the extent of cell death, we used the coculture system of OsiSh-2/ $\Delta cchH$ and rice suspension cells. Compared with rice leaves, suspension cells are single cells and are more sensitive to external stimuli (i.e., treatment with OsiSh-2 and $\Delta cchH$), and the occurrence of cell death can be observed more rapidly and easily quantified. We added the related results to Supplementary Fig. 4, and described them as follows: “When we added the spore suspension of $\Delta cchH$ (10^8 spores mL^{-1}) to the rice suspension cells, the dead cells decreased by 34.90% compared with those of OsiSh-2 (WT)-supplemented rice suspension cells at 6 hpt (Supplementary Fig. 4).” in the revised manuscript on page 7, lines 233-235. This is consistent with the much lower Fe^{3+} and ROS accumulation in E+ ($\Delta cchH$) compared to E+(WT), as shown in Fig 5. All these results indicated that siderophore synthesis deficiency in $\Delta cchH$ failed to trigger effective ferroptosis.

Comment [6]: The conclusion “We demonstrated that, by maintaining the immune-primed state in symbionts, ferroptosis can improve disease resistance against *M. oryzae* infection” seems an overstatement in the context of the studies performed. Ferroptosis is a cell death mechanism and was never measured. The immune-primed status needs

further investigation, as ROS accumulation was not quantified and callose deposition seems higher but not earlier.

Response [6]: Thank you very much for this comment. As we mentioned in responses [1], [4] and [5], the accumulation of Fe^{3+} , ROS and cell death in each case have been quantified and supplemented in the revised manuscript. Therefore, from the quantified data shown in Fig. 6c, d, we can see that an earlier ROS peak was observed in E+ (WT) M+ rice leaves from 0.5 to 6 hpt. Similarly, callose deposition around the stomata was observed in E+ (WT) M+ rice leaves at least 4 h earlier than that in E- M+ rice. At these times, the strength of ROS and callose intensity in E+ (WT) M+ rice was stronger than that in E- M+ and E+ ($\Delta cchH$) M+ rice. The rapid and strong accumulation of ROS and callose in E+ (WT) M+ rice presented a typical defence priming response pattern. The fact that the siderophore mutant strain $\Delta cchH$, which could not trigger the occurrence of ferroptosis in rice, failed to initiate the priming state confirmed the relationship between ferroptosis and priming. Even so, we are not sure that there is a direct connection between ferroptosis and priming. Perhaps siderophores can also trigger priming. The mechanisms need to be further studied. Therefore, we agree with the reviewer's comment that our conclusion about ferroptosis and the immune-primed state in symbionts is overstated. The conclusion in the results was revised to "These results indicated that the occurrence of effective ferroptosis in OsiSh-2-treated rice may be correlated with priming induction in the symbiont under biotic stress, consequently enhancing disease resistance. The related mechanism still requires further study." in the revised manuscript on page 9, lines 278-280. The related discussion was also revised to "We suggest that ferroptosis can improve disease resistance against *M. oryzae* infection by priming-induction mechanism, a notion that requires further study." in the revised manuscript on page 13, lines 428-430.

To Reviewer #3:

Comment [1]: The first doubt is about the construction of the mutant. As shown in Supplementary Figure no. 3, the $\Delta cchH$ mutant strain of *Streptomyces hygroscopicus* OsiSh-2 continues to produce siderophores. The authors did not explain well, or I did

not see, what media they used to detect production. They only mention liquid medium. I would suggest that the authors showed another type of evidence to know exactly what percentage of the mutant continues to produce siderophores, and how this affects their results, since much of their work claims that there is an important role of these iron-chelating compounds in the process of ferroptosis.

Response [1]: Thank you very much for this comment. To assess the siderophore producing capability, OsiSh-2 strains were inoculated into a modified Fe-deficient hydroponic solution (soluble starch 20 g, NaCl 0.5 g, KNO₃ 1 g, K₂HPO₄·3H₂O 0.655 g, MgSO₄·7H₂O 1.025 g, deionized water volume to 1 L, and pH = 7.0) for 12 days. Culture broth was collected each day and the siderophore content was determined by using a commonly used chrome azurol S (CAS) method. The absorbance at 630 nm of each CAS solution was determined to quantify the siderophore activity. In the first version of our manuscript, we used a commonly used method for determining the production of siderophores, as listed below:

Method 1: Siderophore units (%), containing all kinds of siderophores that can be detected by the CAS method (as shown in Supplementary Fig. 6d).

$$\text{Siderophore units (\%)} = \frac{Ar - AS}{Ar} \times 100$$

where Ar = absorbance at 630 nm of the reference sample (CAS solution and uninoculated broth), and As = absorbance at 630 nm of the sample (CAS solution and supernatant of sample).

To be sure what percentage of the mutant continues to produce siderophores, we added another commonly used calculation methods to represent the siderophore activity in the revised manuscript:

Method 2: DFO-B (equiv.)/OD value, which is mainly calculated by measuring the contents of a commercially available siderophore desferrioxamine B (DFO-B) after establishing a standard curve of DFO-B ($R^2 > 0.99$), indicates the relative quantification of siderophores in cells. The results have been added as Supplementary Fig. 6e.

The results showed that on day eight, when the biomass of the two strains tended to be the same, the amount of siderophores of $\Delta cchH$ was 32.51% (Method 1) and 36.80% (Method 2) lower than that of OsiSh-2 (WT) (Supplementary Fig. 6d, e). Based on the

results of these two calculation methods, we considered that the siderophore activity of $\Delta cchH$ was lower than that of OsiSh-2 (WT).

According to the reviewer's suggestion, we added another piece of evidence to further confirm that the siderophore producing capability of $\Delta cchH$ was weakened. Wild-type OsiSh-2 and $\Delta cchH$ strains were inoculated on CAS agar plates, where the colour of the CAS-iron complex changed from blue to orange after chelation of bound iron by siderophores. Therefore, the activity of siderophores can be indicated by the diameter of the orange zone in culture plates. Consistent with the liquid solution cultivation, the diameter of the orange zone in $\Delta cchH$ was reduced by 19.34% when compared with that of OsiSh-2 (WT) after 14 days of cultivation. The result has been added to Supplementary Fig. 6f, g, and all of the above supplemented data have been described as follows: "On day eight, when the biomass of the two strains tended to be the same, the amount of siderophores of $\Delta cchH$ was 32.51%-36.80% lower than that of OsiSh-2 (WT) (Supplementary Fig. 6c-e). To further determine that the siderophore-producing capability of $\Delta cchH$ was weakened, OsiSh-2 (WT) and $\Delta cchH$ strains were inoculated on chrome azurol S (CAS) agar plates, which are useful in the identification of siderophores. Consistent with the solution cultivation, the diameter of the orange zone in $\Delta cchH$ was reduced by 19.34% when compared with that of OsiSh-2 (WT) after 14 days of cultivation (Supplementary Fig. 6 f, g)." in the revised manuscript on page 7, lines 213-219. All this detailed experimental information has been added to the "Methods" section of the revised manuscript on pages 17-18, lines 560-580.

In rice tissues, it is difficult to detect the amount of OsiSh-2-produced siderophores. Hence, we monitored the relative expression levels of *ShCchH* and *ShDesD*, the core synthesis genes of siderophores COE and DFO of OsiSh-2, respectively, in E+ (WT) and E+ ($\Delta cchH$) rice in real time. We added the related results as "We then sprayed spore suspensions of OsiSh-2 (WT) and $\Delta cchH$ onto rice leaves. Considering that it is difficult to detect the amount of OsiSh-2-produced siderophores in rice tissues, we monitored the relative expression levels of *ShCchH* and *ShDesD* in E+ (WT) and E+ ($\Delta cchH$) rice in real time. In contrast to the strong induction of *ShCchH* and *ShDesD* expression at 6-12 hpt in E+ (WT) rice, *ShCchH* failed to activate expression, and the

transcriptional levels of *ShDesD* were obviously decreased in the E+ ($\Delta cchH$) rice (Supplementary Fig. 6h, i). These results suggested that the biosynthesis of COE was indeed decreased, which might influence the production of other siderophores, such as DFO.” in the revised manuscript on page 7, lines 223-230. In combination, the results showing that the siderophore production of OsiSh-2 is synchronously activated with the occurrence of OsiSh-2-triggered ferroptosis (Fig. 1) and the stunted growth of OsiSh-2 at 6-12 hpt in E+(WT) rice (Fig. 3a), we considered that the biosynthesis of siderophores might affect the regulatory process of ferroptosis by OsiSh-2 in host rice.

Comment [2]: The second doubt is about the confirmation of the deletion in strain OsiSh-2, please include sequencing results in supplementary data. Also, describe the conjugating *E. coli* strain used and a brief description in Methods, not only to mention it in a reference.

Response [2]: According to the reviewer’s suggestion, the sequencing results of the $\Delta cchH$ mutant strain are listed below and have been added to Supplementary Text 1. In our study, the *E. coli* ET12567 (pUZ8002) strain was a kind gift from Prof. Yuhui Sun, School of Pharmaceutical Sciences, Wuhan University. *E. coli* ET12567, as a DNA donor to avoid methylated DNA restriction systems of actinomycetes, has been successfully applied for the intergeneric transfer of plasmids from *E. coli* to *Streptomyces* species. Plasmid pUZ8002 is often used for cloning and expression of genes of interest, as well as for DNA sequencing and other molecular biology techniques and it is capable of mobilizing other plasmids efficiently^{1,2}. Thus, *E. coli* ET12567, into which the helper plasmid pUZ8002 was cloned, we used as a donor strain for conjugation into *Streptomyces sp.* strain OsiSh-2. The description of *E. coli* strains that we used for conjugational transfer was added to the “Methods” in the revised manuscript on page 17, lines 550-559 and listed in Supplementary Table 2.

The method that we used for the intergeneric conjugation between *E. coli* and *Streptomyces* was performed as described previously³. Briefly, the plasmid pYH7-*cchH* was introduced into the wild-type strain OsiSh-2, with the helper plasmid pUZ8002 in *E. coli* ET12567 through conjugation on MS agar plates. Single colonies of the

exconjugants were selected by inoculating MS plates with 25 µg/mL apramycin and 30 µg/mL nalidixic acid. The colonies with apramycin resistance were reinoculated onto MS agar plates for relaxation. The apramycin sensitive colonies were then selected as candidates for double-crossover mutants for PCR screenings using test-F and -R as the primers (Supplementary Table 1), which should give a 1292-bp PCR product, followed by sequencing confirmation (Supplementary Text 1). Detailed information on intergeneric conjugation has been added to the revised manuscript on page 17, lines 550-559.

Reference:

1. Deng, L. et al. Dissection of 3D chromosome organization in *Streptomyces coelicolor* A3(2) leads to biosynthetic gene cluster overexpression. *Proceedings of the National Academy of Sciences of the United States of America*. **120**, e2222045120 (2023).
2. Paranthaman. et al. Intergeneric: Conjugation in *Streptomyces peucetius* and *Streptomyces* sp. Strain C5: Chromosomal Integration and Expression of Recombinant Plasmids Carrying the *chiC* Gene. *Applied & Environmental Microbiology*. **69**, 84-91 (2003).
3. Liu, Y. et al. Editorial: Role of endophytic bacteria in improving plant stress resistance. *Frontiers in plant Sci*. **13**, 1106701 (2022).

Comment [3]: Another suggestion is to use CAS medium if the strain grows well and visualize the production of siderophores. Also, it would be nice to know what kind of siderophore the OsiSh-2 strain produces. This would allow us to understand, at least up to now, and explore in the future whether the molecular diversity of bacterial siderophores could have a more general role in other bacterial endophytes in ferroptosis, or if this is a unique mechanism of *Streptomyces* and its type of siderophore(s).

Response [3]: This comment is greatly appreciated. As we mentioned in response [1], we have supplemented the CAS method to detect the production of siderophores. Regarding the kind of siderophores produced by the OsiSh-2 strain, siderophores can be classified into three main categories: hydroxamates, catecholates, and carboxylates depending on the oxygen ligands for Fe (III) coordination. Based on the genome annotation analysis in our previous study, there are four secondary metabolite-synthesizing gene clusters in the OsiSh-2 genome involved in the biosynthesis of

siderophores. Among them, two showed 100% similarity to those of the siderophores desferrioxamine B (DFO-B) and coelichelin (COE), which have been commonly found in many *Streptomyces* species¹. In addition, combined with the results of the FeCl₃ test for siderophores, the ferric perchlorate assay for the hydroxamate type and the Arnow test for the catecholate type, we confirmed that there were abundant siderophores secreted by OsiSh-2, which might contain two types of siderophores: hydroxamates and catecholates².

We also tried to identify the siderophore type secreted by OsiSh-2. The isolation of siderophores was carried out by extraction with different solvents. The obtained active siderophore candidates confirmed by the CAS assay were then identified by liquid chromatography tandem mass spectrometry (LC–MS) (Figure below). Four potential siderophore molecules, including m/z 601.3, m/z 1058, m/z 566.3, and m/z 973.3, were detected in the active substances extracted from the culture filtrate of OsiSh-2. The compound at m/z 601.3 is quite close to the molecular weight of DFO-B (600.705), and the substance at m/z 566.3 is fairly near the molecular weight of COE (565.5747). Therefore, the siderophores secreted by OsiSh-2 might be DFO-B and COE. Further purification, separation, and validation are currently underway.

Hydroxamate-type siderophores comprise the most common group of siderophores found in nature, such as DFO and COE. These siderophores are produced by microorganisms including bacteria and fungi, such as *Streptomyces* spp.³, *Trichoderma* spp.⁴, and *Pseudomonas fluorescens*⁵. Catecholate-type siderophores, such as enterobactin and salmochelin, are mostly produced by certain bacteria, such as *Escherichia coli* and *Klebsiella pneumoniae*⁶. We obtained the COE synthetic deletion mutant of OsiSh-2 but failed to obtain the DFO-related mutant, as DFO might be essential for the normal growth of OsiSh-2. As COE is not commercially available for purchase, we only tested a *Streptomyces*-derived commercial DFO in this study to show its role in ferroptosis and disease resistance. The results were supplemented as “we added the *Streptomyces*-derived siderophore deferoxamine (DFO) to rice plants treated with OsiSh-2. The addition of DFO obviously inhibited Fe³⁺ and ROS accumulation in E+ (WT) rice (Supplementary Fig. 7a-d), and the disease resistance against *M. oryzae*

was significantly reduced (Supplementary Fig. 7e-g). This is consistent with the fact that DFO serves as a ferroptosis inhibitor. However, when DFO was applied in $\Delta cchH$, the Fe^{3+} and ROS accumulation was partially rescued at 6 hpt in E+ ($\Delta cchH$) DFO+ rice, and the blast resistance effect was improved close to that of E+ (WT) mock rice (Supplementary Fig. 7). This result indicated that when the occurrence of ferroptosis was restricted in E+ ($\Delta cchH$) cells, an appropriate amount of DFO could help induce ferroptosis, although the ferroptosis level could not be rescued to that in E+ (WT) cells. Notably, the single treatment of DFO did not influence the disease resistance of the rice plants, as both lesion length and relative fungal biomass in E- DFO+ M+ rice were similar to those in E- mock M+ rice (Supplementary Fig. 7e-g). That is, DFO might play a subtle role in the regulation of ferroptosis in rice rather than directly contributing to rice blast resistance. The above results further confirmed that the moderate ferroptosis triggered by OsiSh-2 (WT) is necessary for improving rice disease resistance against *M. oryzae*. Although siderophore coelichelin would have been more appropriate for this experiment, the compound is not commercially available, and we were unable to purchase it for use in this study.” in the revised manuscript on page 8, lines 246-262. From these results, we are not sure that the molecular diversity of bacterial siderophores could have a more general role in other bacterial endophytes in ferroptosis.

Fig. Preliminary separation process of OsiSh-2 siderophore material. **a** An OsiSh-2 fermentation supernatant was subjected to adsorption and elution by macroporous resin AmberliteXAD-16 and preliminary purification by hydroxypropyl dextran gel Sephadex LH-20. **b** Methanol extract of ISP2 solid medium of OsiSh-2 was subjected to (chloroform, ethyl acetate, n-butanol) systematic solvent extraction followed by adsorption and elution with macroporous resin D101. **c** LC–MS assay results of sample No. 1 of the preliminary separation of OsiSh-2 siderophores. **d** LC–MS assay results of sample No. 2 of the preliminary separation of OsiSh-2 siderophores.

References

1. Liu, Y. et al. The vital role of ShTHIC from the endophyte OsiSh-2 in thiamine biosynthesis and blast resistance in the OsiSh-2-rice symbiont. *J Agric Food Chem.* **70**, 6993-7003 (2022).
2. Zeng, J. R. et al. The Role of Iron Competition in the Antagonistic Action of the Rice Endophyte *Streptomyces sporocinereus* OsiSh-2 Against the Pathogen *Magnaporthe oryzae*. *Microb Ecol.* **76**, 1021-1029 (2018).
3. Xia, K. Y. et al. Aromatic Polyketides and Hydroxamate Siderophores from a Marine-Algae-Derived *Streptomyces* Species. *J. Nat. Prod.* **84**, 1550–1555 (2021).
4. Vinale, F. et al. Harzianic acid: a novel siderophore from *Trichoderma harzianum*. *FEMS Microbiology Letters.* **347**, 123-129 (2013).
5. Grobelak, A. & Hiller, J. Bacterial siderophores promote plant growth: screening of catechol and hydroxamate siderophores. *International Journal of Phytoremediation.* **19**, 825-833 (2017).
6. Wilson, B.R. et al. Siderophores in Iron Metabolism: From Mechanism to Therapy Potential. *Trends Mol Med.* **12**, 1077-1090 (2016).

Comment [4]: Please correct citation in Line 56 (Distéfano et al., 2017).

Response [4]: We apologize for our carelessness. The superfluous “(Distéfano et al., 2017)” has been deleted in the revised manuscript in line 54.

Comment [5]: Another suggestion to better visualize some graphs I suggest deleting the ‘arc lines’ where there are only two treatments and leaving the percentage that shows the difference. For example, Fig. 2, panels f-i.

Response [5]: As the reviewer kindly suggested, we have deleted the ‘arc lines’ where there are only two treatments and leaving the percentage that shows the difference in revised manuscript.

Comment [6]: Please include the following recent references:

Distéfano, A. M., López, G. A., Bauer, V., Zabaleta, E., & Pagnussat, G. C. (2022). Ferroptosis in plants: regulation of lipid peroxidation and redox status. *Biochemical Journal*, 479(7), 857-866.

Hao, Y. J., Zou, Z. B., Xie, M. M., Zhang, Y., Xu, L., Yu, H. Y., ... & Yang, X. W. (2023).

Ferroptosis Inhibitory Compounds from the Deep-Sea-Derived Fungus *Penicillium* sp. MCCC 3A00126. *Marine Drugs*, 21(4), 234.

Response [6]: Thank you very much for this comment. We have added the relevant references in the revised manuscript on page 2, lines 60.

To Reviewer #4:

Comment [1]: The main argument of this manuscript is that the endophytic colonization by *S. hygroscopicus* triggers a ROS- and Fe³⁺-accumulated cell death response in rice leaves. In Figure 1a and 1b and Figure 2b, visible cell death shown on rice leaves upon *S. hygroscopicus* inoculation appears as randomly distributed brown necrotic spots or blotches (Figure 1a and Figure 2a) that were further revealed by trypan blue staining (Figure 1b). However, the histological Fe³⁺ staining by Prussian blue is associated with the stomatal apparatus (Figure 1c; Figure 2b, 2d; Figure 4f; and Figure 5b). Similarly, ROS accumulation revealed by staining with a fluorescence probe CM-H₂DCFDA (Figure 1d, Figure 2c, 2e; Figure 4g, Figure 5c; Figure 6a) seems to be mostly associated with the stomatal cell areas. As the ferroptosis is strictly associated with the hallmark of Fe³⁺ and ROS accumulation within the dead cells, further cautious analysis is needed to confirm whether the cell death process and the Fe³⁺ and ROS accumulation are simultaneously occurring within the same cells/tissues.

Response [1]: Thank you very much for this comment. To confirm whether the cell death process and the accumulation of Fe³⁺ and ROS simultaneously occur within the same cells/tissues, we made many attempts. However, we realized that Fe³⁺ and ROS accumulation are early steps of ferroptosis formation, while a large amount of cell death in rice leaves requires a continuous accumulation process. Thus, the phenotype of cell death can only be observed later. This means that it is difficult to show Fe³⁺ accumulation with Prussian staining and cell death with trypan blue staining at the same time in rice leaves.

Although in Figure 1b, visible cell death shown on rice leaves upon OsiSh-2 inoculation appears as randomly distributed blotches revealed by trypan blue staining, from the magnified images of trypan blue staining that we added in Supplementary Fig.

1a, we noticed that the blue dots that indicated dead plant cells were mainly found in and around the stomata and adjacent mesophyll cells. This is consistent with the results of Fe^{3+} accumulation monitored by Prussian staining, i.e., the Fe^{3+} accumulated in the stomata cells (Fig 1c). The Fe^{3+} accumulation observed at the early stage and the subsequently- detected dead cells (blue spots) were at the same location of rice leaves indicating that Fe^{3+} accumulation is accompanied by cell death in some rice cells after OsiSh-2 treatment.

Corresponding to the added magnified images of trypan blue staining with different magnifications in Supplementary Fig. 1a, we described this as follows: “The trypan blue staining assay confirmed the occurrence of cell death in E+ rice leaves (Fig. 1b). In addition, as the magnified images show in Supplementary Fig. 1a, the blue dots, which indicated dead plant cells, were mainly found in and around the stomata and adjacent mesophyll cells.” in the revised manuscript on page 3, lines 90-93.

In addition, when we monitored Fe^{3+} accumulation in E+ rice leaves in real time by Prussian staining, we observed Fe^{3+} accumulation in the stomata at 6 hpt. In the same location, we observed the appearance of brown spots, which can generally represent cellular senescence or death from 24 hpt (Supplementary Fig. 1b). We added magnified images of Prussian staining with different magnifications in the Supplementary Fig. 1b. The following description was added to the revised manuscript on page 3, lines 98-101: “Moreover, as the magnified Prussian staining images show in Supplementary Fig. 1b, the appearance of brown spots, which generally represent cellular senescence or death, was detected in and around the stomata at 24 hpt in the E+ rice”.

Combining the above supplemented evidence and the results that Fe^{3+} and ROS accumulation are associated with the stomatal apparatus (Fig. 1c, e), the conclusion that OsiSh-2 can trigger ferroptosis in host rice was added as “The fact that the Fe^{3+} and ROS accumulation observed at the early stage and the subsequently detected dead cells were in the same location in the rice leaves, i.e., in and around the stomata and adjacent mesophyll cells, indicates that the Fe^{3+} - and ROS-dependent cell death triggered by OsiSh-2 was ferroptosis.” in the revised manuscript on pages 3-4, lines 101-104.

To further verify our conclusion, we used the coculture system of OsiSh-2 and rice

suspension cells. Compared with the rice leaves, the suspension cells are single cells and more sensitive to external stimuli (i.e., treatment with OsiSh-2), and the occurrence of cell death can be observed more rapidly. We added this experiment and results to the revised manuscript on pages 4-5, lines 128-141 as follows: “We also added a spore suspension of OsiSh-2 (E+) to rice suspension cells at a final concentration of 10^8 spores mL^{-1} and then monitored cell death by propidium iodide (PI) staining and Fe^{3+} accumulation by Prussian blue staining. In E+ rice suspension cells, the amount of dead cells increased by 213.64% compared with those of untreated (E-) rice suspension cells at 6 hpt (Supplementary Fig. 3a, b). Meanwhile, Fe^{3+} accumulation was observed only in the E+ rice suspension cells (Supplementary Fig. 3c). Moreover, the addition of the ferroptosis inducer erastin significantly increased the number of dead cells (45.32%) in E+ Erastin+ rice suspension cells, while the ferroptosis inhibitor Fer-1 alleviated cell death by 41.83% in E+ Fer-1+ rice suspension cells compared with E+ mock rice suspension cells (Supplementary Fig. 3a, b). To rule out the possibility that the observed phenomenon was due to the action of the compound erastin or Fer-1 alone in rice cells and OsiSh-2, we supplemented the treatments of erastin and Fer-1 in E- rice or culture medium of OsiSh-2 as controls. However, erastin and Fer-1 had almost no effect on rice cell viability (Supplementary Fig. 3a, b) or OsiSh-2 growth (Supplementary Fig. 3d). Overall, we concluded that OsiSh-2 can induce ferroptotic cell death in host rice.”.

Comment [2]: Furthermore, while the Prussian blue staining is mostly associated with the stomatal apparatus, strong blue staining (Figures 2b and 2d) clearly appears at the pores of stomata, rather than in the guard cells or subsidiary cells. The Prussian blue staining protocol used in this study doesn't provide sufficient resolution and specificity of Fe^{3+} accumulation within the cells/tissues that underlie the cell death triggered by *S. hygrosopicus* colonization. Additional imaging approaches, such as the synchrotron radiation micro-X-ray fluorescence (SXRF), will help to examine and identify intracellular iron pools at the cell/tissue level in rice leaves.

Response [2]: According to the reviewer's suggestion, we tried to detect Fe^{3+} accumulation in rice leaves by using synchrotron radiation micro-X-ray fluorescence

(SXRF)¹ at the Institute of High Energy Physics, Chinese Academy of Sciences. Due to the limitations of the appointment system and the SXRF equipment system, which needs to be maintained during the summer holiday (August to mid-September), we only conducted one experiment by scanning part of the rice leaves. We found that the iron indicated by reddish brown intensity mainly accumulated around or in the main veins of E+ rice leaves, while it was distributed almost evenly around E- rice leaves. This result verified that the iron was redistributed in E+ rice leaves (figure below). This is consistent with the histological Fe³⁺ staining by Prussian blue in Supplementary Fig. 1b, i.e., the Fe³⁺ accumulation is associated with the stomatal apparatus, which was also mainly around the main veins of E+ rice leaves. In the same location, we detected the appearance of brown spots which can generally represent cellular senescence or death (Supplementary Fig. 1b). These results suggested that ferroptosis is involved in the cell death triggered by OsiSh-2. Since this is a preliminary study, we did not add these results to the revised manuscript. We very much appreciate the reviewer's suggestion. We will apply this efficient ion detection technology in our future studies.

Fig. The determination of iron distribution in rice leaves by using synchrotron radiation micro-X-ray fluorescence (SXRF). **a, b** Microscopic X-ray fluorescence (μ -XRF) analysis was performed at the 4W1B beamline in the Beijing Synchrotron Radiation Facility, which runs 2.5 GeV electrons with a current of 250 mA, to determine the iron distribution in E- (**a**) and E+ (**b**) rice leaves.

A polychromatic beam (pink beam) with an incident X-ray energy of 10-18 keV is used. The size (FWHM) of the beam spot is focused down to $\sim 50 \mu\text{m}$ by a polycapillary half-lens. The two-dimensional mapping is acquired by fly scan mode: the sample is held on a precision motor-driven stage, scanning $50 \mu\text{m}$ stepwise. The XRF raw data were processed by standard procedures using the PyMca package.

Reference

1. Solé, V. A. et al. A multiplatform code for the analysis of energy-dispersive X-ray fluorescence spectra. *SPECTROCHIM ACTA B.* **62**, 63-68 (2007).

Comment [3]: Although cell/tissue death on rice leaves was evident by the appearance of visible necrotic lesions or by the detection of trypan blue staining, there was no data to show whether/how these cells/tissues are endophytically infected/colonized by *S. hygroscopicus*. Authors should provide a direct comparison between the *S. hygroscopicus* colonization, accumulation of Fe^{3+} and ROS, and the cell death event in a spatiotemporal manner at the single cell level. Using a fluorescence protein-labeled *S. hygroscopicus* strain or selective staining of *S. hygroscopicus* would help the analysis.

Response [3]: This comment is greatly appreciated. We have tried for years to establish a stable genetic operating system for *Streptomyces* strains. For instance, another endophyte in our lab *S. albidoflavus* OsiLf-2^{1,2} has been successfully labelled by eGFP or mCherry proteins, as shown in the figure below. However, this operation failed in *S. hygroscopicus* OsiSh-2. The mutant strains of OsiSh-2 are either difficult to keep alive or hard to successfully transform. Unfortunately, we still cannot use fluorescence staining to detect the colonization and functional position of OsiSh-2 in host rice. Further study and improvement of the genetic operation of OsiSh-2 are ongoing.

Alternatively, we revealed the colonization of OsiSh-2 by detecting the relative biomass of OsiSh-2 in rice. This was calculated using the threshold cycle (C_T) of *ShRpoA* (a DNA-directed RNA polymerase subunit alpha of OsiSh-2) DNA against the C_T of *OsUbq* (a rice genomic ubiquitin gene) DNA as a ratio (*ShRpoA/OsUbq*), represented by the equation $2^{C_T(OsUbq) - C_T(ShRpoA)}$. This method referred to the commonly used DNA-based qPCR assay for relative quantification of the endophyte or pathogen

within its host plants. The assay has been confirmed to be highly specific and suitable to reliably detect amounts of endophytes in host plants^{3,4}.

Fig. Fluorescence detection of mCherry-labelled strains of *S. albidoflavus* OsiLf-2 by laser scanning confocal microscopy. **a** The red fluorescence signal of OsiLf-2 (pSN3: mCherry) in the primordia and tip of rice roots at 7 days posttreatment. No fluorescence was observed in the E- rice root. Bar = 10 μ m. **b** The red fluorescence signal in the hyphae and spores of OsiLf-2 (pSN3: mCherry). No fluorescence was observed in the unlabelled strains of OsiLf-2.

References

1. Gao, Y. et al Antagonistic activity against rice blast disease and elicitation of host-defence response capability of an endophytic *Streptomyces albidoflavus* OsiLf-2. *Plant Pathology*. **69**, 259-271 (2020).
2. Niu S. et al. The osmolyte-producing endophyte *Streptomyces albidoflavus* OsiLf-2 induces drought and salt tolerance in rice via a multi-level mechanism. *The Crop Journal*. **10**, 375-386 (2022).
3. Park, C. H. et al. The *Magnaporthe oryzae* Effector AvrPiz-t Targets the RING E3 Ubiquitin Ligase APIP6 to Suppress Pathogen-Associated Molecular Pattern–Triggered Immunity in Rice. *Plant Cell*. **24**, 4748–4762 (2012).

4. Kawano Y. et al. Activation of a Rac GTPase by the NLR family disease resistance protein pit plays a critical role in rice innate immunity. *Cell host & microbe*. **7**, 362-375 (2010).

Comment [4]: Rice leaves at 48 hours post inoculation (hpi) with *S. hygroscopicus* spores exhibited the lesion mimic symptoms (Figure 1a and 1b; Figure 2a). However, detection for both Fe³⁺ and ROS was performed within 24 hpi where the peak for Fe³⁺ level was detected at 6 hpi. At the similar inoculation stages, transcript profiling for immune-related genes (Figure 1e-h) and ferroptosis gene (Figure 1i) were performed, and GPX4 enzyme activity (Figure 1j) and GSH/GSSG levels (Figure 1k, 1l) were measured at 12 hpi. *Streptomyces* spp. are filamentous organisms that undergo several distinct stages of growth, involving spore germination, aerial hyphae emerge, and production of spore chains. At these early stages within 12 hours post inoculation, does *S. hygroscopicus* infect and colonize the host leaf cells/tissues? What are the developmental stages/forms present within the leaf tissues?

Response [4]: Thank you very much for this comment. As shown in Fig. 3a, we monitored the endophytic biomass of OsiSh-2 in rice in real time from 0.5 -168 hours post inoculation (hpi). We can see that from 0.5 hpi, OsiSh-2 colonized the host leaf cells/tissues. Because it is difficult to obtain a fluorescent protein-labelled *S. hygroscopicus* strain, we performed scanning electron microscopy (SEM) and optical microscopy to observe the colonization of OsiSh-2 at the surface and in the intercellular space of rice, respectively. We found that many OsiSh-2 spores or spore chains (without aerial hyphae) colonized the surface of rice leaves (Figure below a) or in the intercellular space of rice cells (Figure below b) at 12 hpi. The same phenomenon was also found in the rice suspension cells at 6 hpi (Figure below c). As the reviewer mentioned, *Streptomyces* have complex developmental cycles. However, we found that OsiSh-2 spores or spore chains exist only on the surface of rice leaves or in the intercellular space of rice cells. This is a very interesting factor that we did not pay enough attention to before. It was reported that the process involved in spore germination in *Streptomyces* is affected by a multitude of factors, including environmental signals, spore population, different *Streptomyces*, nutrient conditions,

etc¹. As these are yet unknown regulatory pathways, we hypothesized that the maintenance of the spore stage of OsiSh-2 when it colonized the host might be involved in a precise interaction relationship between the endophyte and host plant. In addition, a new mode of *Streptomyces* growth-exploration is promoted by interkingdom interactions. Diverging from the classic life cycle of *Streptomyces*, explorer cells grow as nonbranching vegetative hyphae and can rapidly traverse solid surfaces². Notably, iron availability plays an important role in the modulation of *Streptomyces*³. Whether *S. hygrosopicus* OsiSh-2 can apply an exploration model, that is closely related to the colonization and symbiosis of endophytes in the host remains to be further studied.

Fig. SEM and optical microscope image of OsiSh-2 spores in rice leaves and suspension cells.

a, b Spore morphology of OsiSh-2 on rice leaves at 12 hours post inoculation (hpi) observed with scanning electron microscope (**a**) and optical microscopy (**b**). Scale bars: 5 μm and 50 μm, respectively. **c** Spore morphology of OsiSh-2 in the coculture system of OsiSh-2 and rice suspension cells at 12 hpi. Scale bars: 50 μm.

References

1. Bobek, J., Šmídová, K. & Čihák, M. A Waking Review: Old and Novel Insights into the Spore Germination in *Streptomyces*. *Frontiers in microbiology*. **8**, 2205 (2017).
2. Jones, S. E. & Elliot, M. A. 'Exploring' the regulation of *Streptomyces* growth and development.

Current opinion in microbiology. **42**, 25-30 (2017).

3. Jones, S. E. et al. *Streptomyces* volatile compounds influence exploration and microbial community dynamics by altering iron availability. *mBio*. **10**, e00171–19 (2019).

Comment [5]: In Figure 6, the fluorescence signals for ROS detection by CM-H₂DCFDA staining at the late time points (48 and 72 hpi) are likely derived from the background noise, as the subject in focus is out of leaf tissues.

Response [5]: Thank you very much for this comment. In Figure 6, from 0.5-72 hpi, the fluorescence signals for ROS detection by CM-H₂DCFDA staining were detected on the rice leaves, and focused on leaf tissues. The misunderstanding that the subject in focus is out of leaf tissues is mainly due to the ROS burst in the stomata of rice leaves almost disappearing at late time points (48 and 72 hpi). We added three magnified images of E- M+, E+ (WT) M+, and E+ ($\Delta cchH$) M+ rice at 72 hpi (Figure below), in which the leaf tissues can be observed.

Fig. Fluorescence microscopy observations of ROS accumulation at 72 hpi (as indicated by CM-H₂DCFDA staining). a-c The magnified images of E- rice infected by *M. oryzae* (M+) (a), E+ (WT) M+ rice (b), and E+ ($\Delta cchH$) M+ rice (c) at 72 hpi. Scale bars: 50 μ m.

Comment [6]: In evaluating the defense responses triggered in rice leaves, authors have chosen *OsLOX2* and *OsWRKY70* in Figure 1e and 1f and *OsPBZ1* and *OsLOX2* in Figure 6d and 6e for transcript profiling by RT-qPCR. Why were different genes used in these similar analyses? In addition, the rationale for choosing these specific immune-related genes for qPCR analysis was not well justified. Transcriptional profiling by choosing more candidate defense genes that decode multiple defense response/signaling pathways or a genome-wide transcriptome profiling would be helpful to interpret the functional readout of the defense responses in the rice leaves upon *S. hygroscopicus* colonization.

Response [6]: We apologize for using different defence response-related genes in these similar analyses. We chose genes because the salicylic acid (SA) and jasmonic acid (JA)/ethylene (ET) signalling pathways are the most important pathways in plant immunity systems during plant–microbe interactions. Hence, to indicate the ferroptosis response triggered by *S. hygroscopicus* OsiSh-2 colonization, we tested the transcription levels of *OsLOX2*, *OsAOS2*, *OsPBZ1*, *OsPR1a*, and *OsWRKY70*. *OsLOX2* and *OsAOS2* are two JA biosynthesis-related genes¹. *OsPBZ1* and *OsPR1a* are important pathogenesis-related genes in the SA signalling pathway². *OsWRKY70* are two transcription regulation factor (TF) genes involved in cross-talk between SA and JA. The results showed that the expression of these five selected genes was mainly induced at 4-12 hpt in E+ rice (figure below). According to the reviewer’s suggestion to reference the genome-wide transcriptome profiling to choose more candidate defence genes, we analysed our proteomic profiling published previously³. Among the proteins related to the SA and JA/ET signalling pathways, we found that the expression of two proteins encoded by *OsPBZ1* and *OsLOX2* significantly varied in response to OsiSh-2 colonization and further *M. oryzae* infection. Indeed, during the defence priming tests in our current study, both *OsPBZ1* and *OsLOX2* were evoked nearly 4 h earlier in E+ M+ rice than in E- M+ rice. Therefore, in the revised manuscript, we chose *OsPBZ1* and *OsLOX2* to both reveal the ferroptosis response upon *S. hygroscopicus* colonization and the priming response upon *M. oryzae* infection. We replaced the expression level

image of *OsWRKY70* with that of *OsPBZ1* in Fig. 1h. Using transcriptome analysis to reveal the ferroptosis response in rice under OsiSh-2 colonization is an appreciated suggestion, and we will apply this technology in our further study.

Fig. The transcript levels of key genes involved in the rice immune system. **a** The expression of genes associated with the JA/ET signalling pathways (*OsLOX2* and *OsAOS2*). **b** The expression of genes associated with the SA signalling pathways (*OsPBZ1* and *OsPRIa*). **c** The expression of genes associated with crosstalk between SA and JA/ET signalling pathways (*OsWRKY70*). The values are the means of two independent experiments, with three replicates per experiment. The error bars indicate the SDs. The bars with asterisks are significantly different as determined by the Tukey–Kramer test (*, $p < 0.05$; **, $p < 0.01$; ***, $p < 0.001$).

Reference:

1. Peng, D. H. et al. Protein elicitor PemG1 from *Magnaporthe grisea* induces systemic acquired resistance (SAR) in plants. *Molecular plant-microbe interactions: MPMI*. **24**, 1239–1246 (2011).
2. Liu, X. et al. Rice *OsAAA-ATPase1* is Induced during Blast Infection in a Salicylic Acid-Dependent Manner, and Promotes Blast Fungus Resistance. *International Journal of Molecular*

Sciences. **21**, 1443 (2020).

3. Bu, Z. G. et al. The Rice Endophyte-Derived α -Mannosidase ShAM1 Degrades Host Cell Walls To Activate DAMP-Triggered Immunity against Disease. *Microbiology spectrum*. **11**, e04824-22 (2023).

Comment [7]: In the main text (line 89), it states that “a cell-death phenotype; i.e., the emergence of lesion mimics (LMs), was observed on rice leaves at 24 hours post treatment (hpt) (Fig. 1a)”, whereas in the Figure 1 legend, it says that “Image of LMs on rice leaves treated with OsiSh-2 at 48 hpt”. Further clarification is needed with regard to the time points on these investigations.

Response [7]: This comment is greatly appreciated. In our real-time observation after OsiSh-2 treatment on rice leaves, we found that the LMs could be observed by the naked eye approximately 24 hpt. At 36 and 48 hpt, the number of LMs was basically the same as that at 24 hpt. The image of LMs on rice leaves treated with OsiSh-2 was indeed taken at 24 hpt in Fig. 1a, b, and 2a. We apologize for our carelessness. We have revised this information in the legends as “Image of LMs on rice leaves treated with OsiSh-2 at 24 hpt”.

Reviewer #1 (Remarks to the Author):

First of all, I thank the authors for putting a lot of effort into conducting additional experiments. However, logical problems in the paper were discovered in the data presented by the authors and are described in detail below.

1) In the revised manuscript Fig 1c, the author replaced the existing photo at 24 hpt under E+ conditions and added new images of brown cell death at 24 and 36 hpt. In the newly added photos by the authors, same as the first version, I can see that Fe³⁺ increased at 6 hpt and then decreased significantly at 12 hpt. On the other hand, new photos showing brown changes similar to cell death replaced the first version manuscript photo at 24 hpt without any reason, and this cell death further increased at 36 hpt. Ferroptotic cell death occurs when high concentrations of Fe³⁺ and ROS persist, but the increase in Fe³⁺ at 6 hpt and then decrease significantly at 12 hpt means that, contrary to the author's claim, typical ferroptotic cell death caused by Fe³⁺ was not occurred at 6 and 12 hpt. Unlike the case where ferroptosis occurred rice cells, the cells at 12 hpt in Fig. 1c showed no accumulation of Fe³⁺ and almost no oxidative damages inside the cells. As evidence of cell death, the authors showed the trypan blue rice leaf photo presented in Fig. 1b of the first manuscript as 48 hpt, but in the revised manuscript of Fig. 1b of the same photo, the photo time was changed to 24 hpt in the figure legend without any explanation, making the data unreliable. In addition, the trypan blue photos in Supplementary Fig. 1a, which was newly added as evidence of cell death, was taken at 24 hpt. The new cell death photos confirmed by trypan blue staining in Supplementary Fig. 1a are consistent with the cell death seen like senescence at 24 hpt, which the author newly added to Fig. 1a, not at 6 hpt when Fe³⁺ accumulated. According to the paper by Dangol et al (2019), after ferroptosis, typical cell death occurs in rice cells due to strong hydroxyl radicals, and cell debris and Prussian blue color due to Fe³⁺ clearly appear and persist within the cells. Therefore, if ferroptosis occurred in cells with accumulated Fe³⁺ at 6 hpt, as the author claimed, cell death should naturally be confirmed at 6 hpt or 12 hpt by trypan blue. The newly added photos at 24 hpt and 36 hpt in Fig. 1c do not appear to be strong ferroptotic cell death, but appear to be cellular senescence or other type of cell death after colonization by OsiSh-2 as written in the text. The reason is that, looking at the results of inoculation by different concentration (107<108<109) of OsiSh-2 in Fig. 4g, the degree of cell death at 24 hpt is proportional to the concentration increase, but Fe³⁺ accumulation is highest at 108. These new Fig. 4g results show that the increase in cell death at 24 hpt is not proportional to Fe³⁺ concentration, but is proportional to OsiSh-2 biomass. This is evidence that the newly added cell death photos at 24 hpt in Fig. 4g is not ferroptotic cell death caused by Fe³⁺ accumulation. Ferroptosis is iron-and ROS-dependent cell death that is accompanied by cell membrane rupture and oxidative damage due to large severe lipid peroxidation (Dangol et al., 2019, Fig. 1b). Therefore, the rice immune response caused by the endophyte OsiSh-2 shown in the photo is not accompanied by ferroptotic cell death, so it can be considered an immune response different from ferroptosis. The author revised the new version discussion to say that ferroptosis is one of the factors, but it cannot be defined as ferroptosis because there is no evidence of typical ferroptotic cell death. Therefore, as mentioned above, I think that it is contradictory to the logic that ferroptosis caused by the endophyte OsiSh-2 is involved in rice disease resistance without true ferroptotic cell death.

2) The authors modified Fig. 3a to add OsiSh-2 biomass changes over 0.5-168 hpt instead of the previous 2-24 hpt. Interestingly, OsiSh-2 biomass showed a pattern of significantly increasing at 4 hpt, then decreasing, increasing again at 24 hpt, and then decreasing again. The authors explained that the reason why OsiSh-2 biomass increased at 4 hpt and then decreased sharply at 6 hpt was due to ferroptosis immune response. However, the Fe³⁺ accumulation presented by the authors in Fig. 1c cannot be defined as ferroptosis because, as mentioned above, only Fe³⁺ accumulation occurred without cell death. This is because the Trypan blue pictures presented in Fig. 1b and supplementary Fig. 1a were all taken at 24 hpt, not 6 hpt, the time of occurrence of ferroptosis as claimed by the author. This decrease in OsiSh-2 biomass is not due to ferroptosis, but is likely related to the increase in ROS, a key element of plant immunity, as shown in Fig. 1e and 1f.

3) In summary, the reviewer requested photos of direct evidence of ferroptosis, which is most important to the paper, but the authors were unable to provide them in the revised data. Thus, the logic of the paper is fundamentally contradictory because there was no occurrence of real ferroptosis at 6 or 12 hpt, which is a key logic in the "ferroptosis-mediated early immune response

model" claimed in this paper.

Reference)

Dangol S, Chen Y, Hwang BK, Jwa NS. Iron- and Reactive Oxygen Species-Dependent Ferroptotic Cell Death in Rice-Magnaporthe oryzae Interactions. Plant Cell. 2019 Jan;31(1):189-209.

Reviewer #2 (Remarks to the Author):

In general, this revised version of the manuscript addressed all the points issued before. However, I found the results obtained after erastin treatment very difficult to interpret and they should be further discussed.

It seems that Erastin only has an effect on rice in the presence of OsiSh-2. If OsiSh-2 is inducing ferroptosis, then an inducer of ferroptosis should behave similarly. However this is not the case. This should be discussed in all the experiments in which erastin is used.

Reviewer #4 (Remarks to the Author):

In the resubmitted version of this manuscript, the authors present a nearly air-tight rebuttal to my comments/concerns, as well as those of the other reviewers. It is clear that the authors made a serious effort to deal with the issues identified by the reviewers. This reviewer greatly appreciates the substantial amount of additional experiments and data that went into the revised version. In my view, this complete manuscript is suitable for a broader audience.

Minor comments:

Scale bars showed in Figure 1e seem to be inconsistent as images for 4, 6, 12, and 24 hpt display larger amplification than that of 36 hpt.

Response to Reviewers

To Reviewer #1:

Comment [1]:

1) In the revised manuscript Fig 1c, the author replaced the existing photo at 24 hpt under E+ conditions and added new images of brown cell death at 24 and 36 hpt. In the newly added photos by the authors, same as the first version, I can see that Fe³⁺ increased at 6 hpt and then decreased significantly at 12 hpt. On the other hand, new photos showing brown changes similar to cell death replaced the first version manuscript photo at 24 hpt without any reason, and this cell death further increased at 36 hpt.

Response: Thank you very much for this comment. In our previous manuscript, to avoid the interference of brown spots or blotches in the observation, we used Prussian blue staining images not around the brown spots or blotches. In the revised manuscript, we added new Prussian blue staining images with brown cell death at 24 hpt and 36 hpt following the opinion of reviewer #4 that “**In Figure 1a and 1b and Figure 2b, visible cell death shown on rice leaves upon *S. hygroscopicus* inoculation appears as randomly distributed brown necrotic spots or blotches (Figure 1a and Figure 2a) that were further revealed by Trypan blue staining (Figure 1b). However, the histological Fe³⁺ staining by Prussian blue is associated with the stomatal apparatus (Figure 1c; Figure 2b, 2d; Figure 4f; and Figure 5b). Similarly, ROS accumulation revealed by staining with a fluorescent probe CM-H₂DCFDA (Figure 1d, Figure 2c, 2e; Figure 4g, Figure 5c, Figure 6a) seems to be mostly associated with the stomatal cell areas. As the ferroptosis is strictly associated with the hallmark of iron and ROS accumulation within the dead cells, further cautious analysis is needed to confirm whether the cell death process and the Fe³⁺ and ROS accumulation are simultaneously occurring within the same cells/tissues**”. Therefore, to better show the process of cell death, we used Prussian blue staining images around the brown spots or blotches, which appeared in E+ rice from 24 hpt in Fig. 1c, 4g and 5b in the revised manuscript.

According to your kind suggestion (**comment 2**) for our original manuscript that **“In Fig. 1c, the image demonstrates that when OsiSh-2 is inoculated, there is a higher accumulation of iron at 6 hpt compared to the non-inoculated condition. However, at 12 and 24 hpt, the iron levels decrease and become similar to those of the non-inoculated rice. This finding makes it difficult to conclude that it leads to typical ferroptosis and suggests a higher possibility of cell recovery rather than the occurrence of typical ferroptotic cell death. If ferroptosis occurs after 24 hpt (eg. 36, 48 hpt), please add microscopic evidences to support this claim.”** We also added information on Fe³⁺ and ROS accumulation at 36 hpt in Fig. 1c, e and f. At this time, the levels of Fe³⁺ and ROS decreased, and Fe³⁺ accumulation around the stomata almost disappeared. The phenomenon of Fe³⁺ and ROS at 48 hpt is the same as that at 36 hpt in rice leaves, and LMs did not continue to increase and expand. Considering that the brown spots interfered with the images of Fe³⁺ accumulation, we did not show the images at 48 hpt.

We replied only to reviewer#4 for replacing the new images for E+ rice at 24 hpt, and we apologize for our thoughtlessness, as we did not explain the changes to you and the other reviewers.

2) Ferroptotic cell death occurs when high concentrations of Fe³⁺ and ROS persist, but the increase in Fe³⁺ at 6 hpt and then decrease significantly at 12 hpt means that, contrary to the author's claim, typical ferroptotic cell death caused by Fe³⁺ was not occurred at 6 and 12 hpt. Unlike the case where ferroptosis occurred rice cells, the cells at 12 hpt in Fig. 1c showed no accumulation of Fe³⁺ and almost no oxidative damages inside the cells.

Response: In this study, we found that Fe³⁺ and ROS accumulation could be simultaneously induced by OsiSh-2 at 6 hpt and then decreased significantly at 12 hpt around the stomata in E+ rice. We also demonstrated that OsiSh-2-triggered Fe³⁺ and ROS accumulation at the early stage of approximately 6 hpt was a ferroptotic cell death in host rice by ferroptosis inducers and inhibitors (Fig. 2, Supplementary Fig. 2). We agree with the reviewer's comment that typical ferroptotic cell death caused by Fe³⁺ accumulation did not occur at 12 hpt. To establish a symbiotic relationship between

OsiSh-2 and the host rice, OsiSh-2 needs to precisely regulate plant cell death to better reside symbiotically within plants. Therefore, OsiSh-2 had to suppress the level of ferroptosis, probably by secreting siderophores in the same way as the commonly used ferroptosis inhibitor DFO. This viewpoint was confirmed by the strong induction of *ShCchH* and *ShDesD* (the core synthesis genes of siderophores COE and DFO) expression at 6-12 hpt in E+ rice (Supplementary Fig. 5f). In addition, it should be noted that the occurrence of ferroptosis is an energy-consuming process, and continuous ferroptosis in rice might inhibit the growth of plants. Therefore, as shown in Fig. 1c, the ferroptosis level decreased significantly at 12 hpt. However, it is difficult to show Fe³⁺ accumulation with Prussian staining and cell death with trypan blue staining in stomata guard cells in rice leaves at the same time. The large amount of cell death observed in rice leaves requires a continuous accumulation process. To verify the occurrence of ferroptotic cell death simultaneously with high concentrations of Fe³⁺ and ROS at 6 hpt, we made more attempts using different methods:

(A) We carried out TUNEL (terminal deoxynucleotidyl transferase-mediated dUTP-biotin nick-end labelling) and propidium iodide (PI) staining assays to show cell death in rice leaf cells. The TUNEL assay can sensitively detect DNA fragmentation caused by programmed cell death (PCD) signalling cascades¹. PI is a versatile indicator dye for dead cells that acts by intercalating with cellular DNA². Therefore, the positive signals of TUNEL and PI staining are shown as green and red fluorescence, respectively, indicating DNA damage and cell death. As shown below, compared to E- rice leaves, much more TUNEL- and PI- positive signals were found in E+ rice leaves at 6 hpt, indicating that OsiSh-2 triggered DNA damage and stomatal cell death in E+ rice leaves. The related results of TUNEL and PI staining were added to Supplementary Fig. 1b, and described in the revised manuscript on pages 3-4, lines 98-103 as “In addition, the leaf tissues of E+ and E- rice at 6 hpt were subjected to terminal deoxynucleotidyl transferase-mediated dUTP-biotin nick-end labelling (TUNEL) and propidium iodide (PI) staining assays. As shown in Supplementary Fig. 1b, compared to the E- rice leaves, many more TUNEL- and PI- positive signals were found in E+ rice leaves at 6 hpt,

indicating that OsiSh-2 triggered DNA damage and stomatal cell death in E+ rice leaves.” The detailed experimental information for the TUNEL assay has been added to the “Methods” section of the revised manuscript on page 15, lines 484-489.

Supplementary Fig. 1 *Streptomyces hygrosopicus* OsiSh-2 triggers cell death in and around stomata and adjacent mesophyll cells in rice leaves. **b** DNA damage and cell death assays by terminal deoxynucleotidyl transferase-mediated dUTP-biotin nick-end labelling (TUNEL, green signals) and propidium iodide (PI, red signals) staining showed cell death in E+ rice leaves at 6 hpt. Scale bars: 20 μm .

(B) Compared with rice leaves, the suspension cells are single cells and more sensitive to external stimuli (i.e., OsiSh-2 treatment), and the occurrence of cell death can be observed more rapidly. Thus, we monitored the cell death of rice suspension cells in real-time by PI staining. As we mentioned in lines 135-143, pages 4-5: “We also added a spore suspension of OsiSh-2 (E+) to rice suspension cells at a final concentration of 10^8 spores mL^{-1} and then monitored cell death by PI staining and Fe^{3+} accumulation by Prussian blue staining. In E+ rice suspension cells, the number of dead cells increased by 213.64% compared with that of untreated (E-) rice suspension cells at 6 hpt (Supplementary Fig. 3a, b). Meanwhile, Fe^{3+} accumulation was observed only in the E+ rice suspension cells (Supplementary Fig. 3c). Moreover, the addition of the ferroptosis inducer erastin significantly increased the number of dead cells (45.32%) in E+ Erastin+ rice suspension cells, while the ferroptosis inhibitor Fer-1 alleviated cell death by 41.83% in E+ Fer-1+ rice suspension cells compared with E+ mock rice suspension cells (Supplementary Fig. 3a, b).”

(C) According to Dangol et al. (2019), we also tried to use the epidermal cells of rice leaf sheaths to detect cell death at 6 hpt. However, we failed to observe Fe^{3+} accumulation and cell death induction in leaf sheath epidermal cells in E+ rice. When we looked at the difference in tissue structures between rice leaves and leaf sheaths, we found that our results are reasonable. Many stomata exist on rice leaves to maintain water balance and photosynthesis. However, in the leaf sheath, there are relatively few stomata^{3,4}. Consistent with these reports, we found that in rice leaves, the stomata were neatly distributed on the epidermis, especially on both sides of the veins. In contrast, in the leaf sheath there are quite a few stomata that are mainly embedded inside the stomatal papillae, and the number of stomata was 80.33% less than that in rice leaves (Figure below a, b). Thus, when facing OsiSh-2 and *M. oryzae*, different responses might be induced in the leaf sheath. Our preliminary research showed that endophytic OsiSh-2 invaded rice cells mainly through stomata. As shown in the figure below by scanning electron microscopy observation (Figure below c), OsiSh-2 is mainly concentrated around the stomata of rice leaves. This is consistent with our observation that the ROS burst mainly occurred around the stomata of E+ rice leaves (Fig. 1e). From this viewpoint, few and unexposed stomata in rice sheath cells led to the failure of OsiSh-2 entrance in host rice as early as 6 hpt through stomata. This mode of action of OsiSh-2 is different from that of *M. oryzae*. The infection and proliferation of *M. oryzae* in the host needs to generate mature appressoria and then accumulate turgor to breach the rice cuticle via a rigid penetration peg in rice cells. Therefore, in the study of Dangol et al. (2019), ferroptotic cell death induced by *M. oryzae* in incompatible rice leaf sheaths was observed at 24 h after infection. Of course, OsiSh-2 can also secrete some cell wall hydrolases to destroy the cell wall structure of host rice, but this process might occur later. In our previous study, we reported that OsiSh-2 can trigger rice immune responses by secreting α -Mannosidase ShAM1 to digest the rice cell wall and release damage-associated molecular patterns, and cell death was also observed at 24 hpt⁵.

Overall, we concluded that OsiSh-2 can induce ferroptotic cell death in host rice at 6 hpt.

Fig. Morphological features of stomata in rice leaves and leaf sheaths. **a** The stomata in rice leaves and leaf sheaths were observed by optical microscopy. Yellow arrows indicate the stomata. Scale bar: 100 μ m. **b** Relative number of stomata in rice leaves and leaf sheaths. Data are represented as the means \pm SDs of ten independent views. **c** The morphology of rice leaves treated with OsiSh-2 was observed using scanning electron microscopy (SEM). Yellow arrow indicates the stomata, red arrow indicates the spores of OsiSh-2. Scale bars: 10 μ m.

Reference:

1. Cui, Y. et al. Disruption of *EARLY LESION LEAF 1*, encoding a cytochrome P450 monooxygenase, induces ROS accumulation and cell death in rice. *Plant J.* **105**, 942-956 (2021).
2. Krämer, C. E., Wiechert, W. & Kohlheyer, D. Time-resolved, single-cell analysis of induced and programmed cell death via non-invasive propidium iodide and counterstain perfusion. *Sci Rep.* **6**, 32104 (2016).
3. Yu, Q., Chen, L., Zhou, W. Q., An, Y. H., Luo, T. X., Wu, Z. L., Wang, Y. Q., Xi, Y. F. & Yan, L. F. RSD1 Is Essential for Stomatal Patterning and Files in Rice. *Front Plant Sci.* **11**, 600021 (2020).
4. Shi, S. J. et al. Bph30 confers resistance to brown planthopper by fortifying sclerenchyma in rice leaf sheaths. *Mol Plant.* **10**, 1714-1732 (2021).

5. Bu, Z. G. et al. The rice endophyte-derived α -Mannosidase ShAM1 degrades host cell walls to activate DAMP-triggered immunity against disease. *Microbiology spectrum*. 11, e04824-22 (2023).

3) As evidence of cell death, the authors showed the trypan blue rice leaf photo presented in Fig. 1b of the first manuscript as 48 hpt, but in the revised manuscript of Fig. 1b of the same photo, the photo time was changed to 24 hpt in the figure legend without any explanation, making the data unreliable.

Response: We apologize for our negligence in changing the figure legend of the trypan blue rice leaf photo presented in Fig. 1b of the first manuscript from 48 hpt to 24 hpt without any explanation. According to Reviewer #4's comment for the original manuscript: **“a cell-death phenotype; i.e., the emergence of lesion mimics (LMs), was observed on rice leaves at 24 hours post treatment (hpt) (Fig. 1a), whereas in the Figure 1 legend, it says that Image of LMs on rice leaves treated with OsiSh-2 at 48 hp. Further clarification is needed with regard to the time points on these investigations.”**, we realized our carelessness. In our real-time observation after OsiSh-2 treatment on rice leaves, we found that the LMs could be observed by the naked eye appeared at approximately 24 hpt. At 36 and 48 hpt, the amount of LMs was basically the same as that at 24 hpt. The image of LMs on rice leaves treated with OsiSh-2 was indeed taken at 24 hpt in Fig. 1a, 1b and 2a. Therefore, we revised the legend for Fig. 1b to “Image of LMs on rice leaves treated with OsiSh-2 at 24 hpt” and responded to reviewer #4 regarding this change. We apologize for our carelessness again.

4) In addition, the trypan blue photos in Supplementary Fig. 1a, which was newly added as evidence of cell death, was taken at 24 hpt. The new cell death photos confirmed by trypan blue staining in Supplementary Fig. 1a are consistent with the cell death seen like senescence at 24 hpt, which the author newly added to Fig. 1a, not at 6 hpt when Fe^{3+} accumulated. According to the paper by Dangol et al (2019), after ferroptosis, typical cell death occurs in rice cells due to strong hydroxyl radicals, and cell debris and Prussian blue color due to Fe^{3+} clearly appear and persist within the cells. Therefore,

if ferroptosis occurred in cells with accumulated Fe^{3+} at 6 hpt, as the author claimed, cell death should naturally be confirmed at 6 hpt or 12 hpt by trypan blue. The newly added photos at 24 hpt and 36 hpt in Fig. 1c do not appear to be strong ferroptotic cell death, but appear to be cellular senescence or other type of cell death after colonization by OsiSh-2 as written in the text. The reason is that, looking at the results of inoculation by different concentration ($10^7 < 10^8 < 10^9$) of OsiSh-2 in Fig. 4g, the degree of cell death at 24 hpt is proportional to the concentration increase, but Fe^{3+} accumulation is highest at 10^8 . These new Fig. 4g results show that the increase in cell death at 24 hpt is not proportional to Fe^{3+} concentration, but is proportional to OsiSh-2 biomass. This is evidence that the newly added cell death photos at 24 hpt in Fig. 4g is not ferroptotic cell death caused by Fe^{3+} accumulation. Ferroptosis is iron- and ROS-dependent cell death that is accompanied by cell membrane rupture and oxidative damage due to large severe lipid peroxidation (Dangol et al., 2019, Fig. 1b). Therefore, the rice immune response caused by the endophyte OsiSh-2 shown in the photo is not accompanied by ferroptotic cell death, so it can be considered an immune response different from ferroptosis. The author revised the new version discussion to say that ferroptosis is one of the factors, but it cannot be defined as ferroptosis because there is no evidence of typical ferroptotic cell death. Therefore, as mentioned above, I think that it is contradictory to the logic that ferroptosis caused by the endophyte OsiSh-2 is involved in rice disease resistance without true ferroptotic cell death.

Response: Trypan blue staining showing a large amount of cell death in rice leaves requires a continuous accumulation process. Therefore, we thought that the cell death that occurred at 6 hpt could only be detected at a later time in rice leaves by trypan blue staining, as shown in Supplementary Fig. 1a, or observed brown necrotic spots or blotches at 24 hpt and 36 hpt, as shown in Fig. 1c. However, we agree with the reviewer that the added photos do not appear to be strong evidence for ferroptotic cell death at 6 h. As we mentioned in response 2), we added the results of TUNEL detection, combined with the rice suspension cell assay, and we concluded that OsiSh-2 can induce ferroptosis occurrence in rice as an early immune response at 6 hpt.

In addition to inducing ferroptosis, OsiSh-2 can also induce other immune

responses, for example later at 24 hpt. After 24 hpt, we observed an increase in OsiSh-2 biomass, which then decreased sharply at 48 hpt. This indicated that an immune response was triggered by the increase in OsiSh-2 accumulation at 24 hpt. The phenomenon that OsiSh-2 induced LM generation in E+ rice leaves at 24 hpt and that the LMs did not continue developing or spreading to the new rice leaves confirmed our speculation. We have discussed this in the revised manuscript on pages 10-11, lines 340-345, and we added examples such as “The second regulation to inhibit the increase in OsiSh-2 after 24 hpt may be ascribed to other immune system processes but not ferroptosis. For example, there was still a strong ROS burst in the rice leaves of E+ rice at 24 hpt (Fig. 1e). In our previous study, we also reported that OsiSh-2 can trigger rice immune responses by secreting α -Mannosidase ShAM1 to digest the rice cell wall and release damage-associated molecular patterns, and cell death was also observed at 24 hpt”.

Comment [2]: The authors modified Fig. 3a to add OsiSh-2 biomass changes over 0.5-168 hpt instead of the previous 2-24 hpt. Interestingly, OsiSh-2 biomass showed a pattern of significantly increasing at 4 hpt, then decreasing, increasing again at 24 hpt, and then decreasing again. The authors explained that the reason why OsiSh-2 biomass increased at 4 hpt and then decreased sharply at 6 hpt was due to ferroptosis immune response. However, the Fe^{3+} accumulation presented by the authors in Fig. 1c cannot be defined as ferroptosis because, as mentioned above, only Fe^{3+} accumulation occurred without cell death. This is because the Trypan blue pictures presented in Fig. 1b and supplementary Fig. 1a were all taken at 24 hpt, not 6 hpt, the time of occurrence of ferroptosis as claimed by the author. This decrease in OsiSh-2 biomass is not due to ferroptosis, but is likely related to the increase in ROS, a key element of plant immunity, as shown in Fig. 1e and 1f.

Comment [3]: In summary, the reviewer requested photos of direct evidence of ferroptosis, which is most important to the paper, but the authors were unable to provide them in the revised data. Thus, the logic of the paper is fundamentally contradictory because there was no occurrence of real ferroptosis at 6 or 12 hpt, which is a key logic

in the “ferroptosis-mediated early immune response model” claimed in this paper. Reference: Dangol S, Chen Y, Hwang BK, Jwa NS. Iron- and Reactive Oxygen Species-Dependent Ferroptotic Cell Death in Rice-Magnaporthe oryzae Interactions. Plant Cell. 2019 Jan;31(1):189-209.

Response [2] and [3]: Thank you very much for these comments. As we mentioned in response [1], we added new and direct evidence and confirmed that OsiSh-2 can induce ferroptosis occurrence in rice as an early immune response at 6 hpt. The inhibition of the increase in OsiSh-2 after 24 hpt may be ascribed to other immune system processes. Thank you again for your suggestion, which prompted us to try some cell death experiments and obtain satisfactory results, such as the TUNEL assay. These results further directly proved that OsiSh-2 can induce ferroptotic cell death in host rice at 6 hpt, which also makes our manuscript more complete.

To Reviewer #2:

Comment [1]: In general, this revised version of the manuscript addressed all the points issued before. However, I found the results obtained after erastin treatment very difficult to interpret and they should be further discussed. It seems that Erastin only has an effect on rice in the presence of OsiSh-2. If OsiSh-2 is inducing ferroptosis, then an inducer of ferroptosis should behave similarly. However, this is not the case. This should be discussed in all the experiments in which erastin is used.

Response [1]: Thank you very much for this comment. The mechanism of action of erastin is causing a decrease in intracellular GSH content, leading to oxidative stress and DNA damage in cells¹. In animal tests, 1-5 μ M erastin is a commonly used concentration range, and it has almost no effect on cell viability^{2,3}. In plant and microbe tests, the most commonly used working concentration of erastin is 10 μ M, and erastin treatment at this concentration does not cause obvious cell death^{4,5}. For instance, erastin treatment at 10 μ M did not trigger iron-dependent ROS accumulation and HR cell death in the sheath epidermal layer of *OsFER2* (a rice ferritin 2, required for ferroptotic cell death) mutant rice⁶. In this study, we also used erastin at 10 μ M. Similarly, although treatment with erastin at 10 μ M induced low level accumulation of ROS, no

accumulation of iron and cell death was detected in E- rice. A higher concentration of erastin than 10 μ M might trigger cell death in rice. However, to avoid a negative effect on the normal growth of rice cells, we used erastin at 10 μ M. We have supplemented this description to the revised manuscript on page 4, lines 129-130 as “Notably, although erastin is an inducer of ferroptosis, this small molecule in 10 μ M did not trigger iron-dependent cell death alone in rice (Fig. 2a).” We also described the effect of erastin in rice suspension cells as “To rule out the possibility that the observed phenomenon was due to the action of the compound erastin or Fer-1 alone in rice cells and OsiSh-2, we supplemented the treatments of erastin and Fer-1 in E- rice or culture medium of OsiSh-2 as controls. However, erastin and Fer-1 had almost no effect on rice cell viability (Supplementary Fig. 3a, b) or OsiSh-2 growth (Supplementary Fig. 3d).” in the revised manuscript on page 5, lines 143-147.

Reference:

1. Zhao, Y., Li, Y., Zhang, R., Wang, F., Wang, T. & Jiao, Y. The Role of Erastin in Ferroptosis and Its Prospects in Cancer Therapy. *Onco Targets Ther.* **13**, 5429-5441 (2020).
2. Alim I. et al. Selenium Drives a Transcriptional Adaptive Program to Block Ferroptosis and Treat Stroke. *Cell.* **177**, 1262-1279.e25 (2019).
3. Jiang, L., Kon, N., Li, T., Wang, S. J., Su, T., Hibshoosh, H., Baer, R. & Gu, W. Ferroptosis as a p53-mediated activity during tumour suppression. *Nature.* **520**, 57-62 (2015).
4. Dangol, S., Chen, Y., Hwang, B. K. & Jwa, N. S. Iron- and Reactive Oxygen Species-Dependent Ferroptotic Cell Death in Rice-*Magnaporthe oryzae* Interactions. *Plant Cell.* **31**, 189-209 (2019).
5. Shen, Q., Liang, M. L., Yang, F., Deng, Y. Z. & Naqvi, N. I. Ferroptosis contributes to developmental cell death in rice blast. *New Phytol.* **227**, 1831-1846 (2020).
6. Nguyen, N. K., Wang, J., Liu, D., Hwang, B. K. & Jwa, N. S. Rice iron storage protein ferritin 2 (*OsFER2*) positively regulates ferroptotic cell death and defense responses against *Magnaporthe oryzae*. *Front Plant Sci.* **13**, 1019669 (2022).

To Reviewer #4:

Comment [1]: In the resubmitted version of this manuscript, the authors present a

nearly air-tight rebuttal to my comments/concerns, as well as those of the other reviewers. It is clear that the authors made a serious effort to deal with the issues identified by the reviewers. This reviewer greatly appreciates the substantial amount of additional experiments and data that went into the revised version. In my view, this complete manuscript is suitable for a broader audience.

Minor comments: Scale bars showed in Figure 1e seem to be inconsistent as images for 4, 6, 12, and 24 hpt display larger amplification than that of 36 hpt.

Response [1]: We greatly appreciate this suggestion. The images for 4, 6, 12, and 24 hpt indeed display larger amplification than that of 36 hpt. We apologize for our carelessness. We have replaced the same image at 36 hpt in consistent magnification with those at 4, 6, 12, and 24 hpt in the revised Figure 1e and displayed it as follows.

Fig. 1 *Streptomyces hygroscopicus* OsiSh-2 triggers an ROS- and Fe³⁺-accumulated cell death response in rice leaves. **e** CM-H₂DCFDA staining shows the accumulation of reactive oxygen species (ROS, green fluorescence) in E+ rice leaves at 6 hpt. Scale bars: 50 μ m. The yellow arrows indicate ROS bursts.

Reviewer #1 (Remarks to the Author):

I had some questions about the photos in Supplementary Fig. 1b additionally submitted by the author to prove cell death at 6 hpt after endophyte inoculation.

1. The authors need to add each transmitted light image so that the photos can indicate which cells are shown and which cells are undergoing death.
2. Looking at the 6hpt image in Fig. 1c, it appears that one cell has Fe accumulation, but in the newly provided Supplementary Fig. 1b photo, the TUNEL green signal appears to be distributed in several cells. The reason must be explained by comparing with a light image mentioned above.
3. It is not clear whether the TUNEL green signal is caused by plant cell death or dead endophytes. You must use DAPI, etc. to show whether the rice nuclear staining image matches the TUNEL green signals.

Response to Reviewers

To Reviewer #1:

Comment [1]: The authors need to add each transmitted light image so that the photos can indicate which cells are shown and which cells are undergoing death.

Response: According to the reviewer's kind suggestion, we supplemented each transmitted light image of E- and E+ rice leaves for Propidium iodide (PI, red) and the terminal deoxynucleotidyl transferase-mediated dUTP-biotin nick-end labelling (TUNEL, green) staining respectively, as well as the light field images. These added images are shown in Supplementary Fig. 1b.

Supplementary Fig. 1 b Cell death and DNA damage assays by propidium iodide (PI, red signals) and terminal deoxynucleotidyl transferase-mediated dUTP-biotin nick-end labelling (TUNEL, green signals) staining respectively showed cell death in the stomata of E+ rice leaves at 6 hpt. Xv, xylem vessel element; St, stomata. Scale bars: 20 μ m.

PI passes through damaged cell membranes and exhibits bright red fluorescence¹. As shown in the figure, the red signals displayed in the stomata (St) of E+ rice leaves, while the stomata of E- rice leaves were not stained. Not surprisingly, in E+ and E- rice cells, obvious red fluorescence was observed in the xylem vessel element (Xv), which comprises dead but functioning cells. The vascular plants possess a regulatory network to coordinate the different phases of xylem maturation, including DNA degradation, partial hydrolysis of the non-lignified primary cell walls, and cell death². So, the xylem vessel element could be stained by the PI.

The TUNEL assay can sensitively detect DNA fragmentation caused by cell death signaling cascades³, and the positive signal was shown with green fluorescence. There are more and stronger green TUNEL signals in E+ rice compared to those in E- rice leaves, especially in stomata. This is due to that OsiSh-2 could activate the plant immune system when it interacts with host rice, including ferroptosis and other immune responses.

In the merged image of PI and TUNEL signals, we observed red PI signal overlapped the green TUNEL signal at the stomata (yellow spot, white arrows), indicating the occurrence of DNA damage and cell death in the stomata of E+ rice leaves at 6 hpt. All these results are described in the revised manuscript on pages 3-4, lines 101-106 as “As shown in Supplementary Fig. 1b, the red signals displayed in the stomata (St) of E+ rice leaves, while the stomata of E- rice leaves were not stained. Meanwhile, the green TUNEL signals were stronger in the stomata of E+ rice leaves than those in the stomata of E- rice leaves. In the merged image of PI and TUNEL signals, we observed red PI signal overlapped the green TUNEL signal at the stomata (yellow spot, white arrows), indicating that OsiSh-2 triggered the DNA damage and stomatal cell death in E+ rice stomata.” The detailed experimental information for the TUNEL assay has been added to the “Method” section of the revised manuscript on page 15, lines 487-494.

From these newly added figures, we indicated the cell death more clearly, and this figure replaced the previous Supplementary Fig. 1b in the revised manuscript. We have to mention that we changed the image of E- rice leaves in the previous Supplementary Fig. 1b. Because, in the previous figure, only one stoma can be observed. To keep consistent with the E+ rice image, the new figure of E- rice with two stomata cells in the field was used. The changed figure should be more convincing. In addition, in the unrevised Supplementary Fig. 1b, the blue fluorescence was observed mainly due to that the merged diagrams contained multiple channels, including blue, red, and green fluorescence channels. In the revised Supplementary Fig. 1b, we only showed the PI and TUNEL signals, as well as their merged diagrams.

Comment [2]: Looking at the 6 hpt image in Fig. 1c, it appears that one cell has Fe accumulation, but in the newly provided Supplementary Fig. 1b photo, the green TUNEL signal appears to be distributed in several cells. The reason must be explained by comparing with a light image mentioned above.

Response: Thank you very much for this comment. In our study, the observation of Fe³⁺ and ROS accumulation was focused on the surface cells of leaf tissue. However, the PI and TUNEL staining assay was performed by embedding rice leaves in paraffin and cutting them into thin slices. Thus, we can also observe the inner cells of rice leaves. So, in Fig. 1c, it appears that Fe³⁺ accumulation was only observed in the stomata cells of rice leaves, while some green TUNEL signal appears to be distributed in several cells, especially in xylem vessel element, due to the normal programmed cell death. This was consistent with the report that a positive TUNEL signal has been observed in the xylem vessel elements of pea roots⁴ and tomato leaves⁵.

Anyway, the accumulation of Fe³⁺, ROS, and cell death were all detected in the stomata of E+ rice leaves at 6 hpt demonstrated that OsiSh-2 triggered Fe³⁺- and ROS-dependent ferroptosis in rice at 6 hpt.

Comment [3]: It is not clear whether the green TUNEL signal is caused by plant cell death or dead endophytes. You must use DAPI, etc. to show whether the rice nuclear staining image matches the green TUNEL signals.

Response: Thank you very much for this comment. We don't think green TUNEL signals are caused by dead endophytes. Because the spore of OsiSh-2 is only 0.5-1.0 μm in size, far less than the size of a stoma (approximately 20 μm). In addition, the green TUNEL signals in E+ rice leaves are relatively regularly shaped. Thus, we considered that the green TUNEL signals are mainly caused by plant cells.

PI and 4'-6-diamidino-2-phenylindole (DAPI) are the two most commonly used DNA dyes. PI is impermeable to cells with an intact plasma membrane, while DAPI tends to combine with relatively intact DNA molecules. Although both of them can be used to distinguish between dead cells and living cells, we think PI staining is more suitable in this study. Because the DAPI staining signal becomes increasingly weaker

correspondingly as a result of the destruction of DNA molecules into small fragments, that is to say, the staining signals of TUNEL and DAPI assays were contrary in intensity. This was confirmed in the study of Liu et al. (2007)⁶. As both DAPI and TUNEL staining were used to detect DNA damage, we preferred PI staining which directly indicates the dead cells to match the green TUNEL signals. Indeed, in the merged image of PI and TUNEL staining, we observed yellow overlaps at the stomata, indicating the occurrence of DNA damage matched cell death at this position (Supplementary Fig. 1b).

Reference:

1. Jones, K., Kim, D. W., Park, J. S. & Khang, C. H. Live-cell fluorescence imaging to investigate the dynamics of plant cell death during infection by the rice blast fungus *Magnaporthe oryzae*. *BMC Plant Biol.* **69**, 1-8 (2016).
2. Bollhöner, B., Prestele, J. & Tuominen, H. Xylem cell death: emerging understanding of regulation and function. *J Exp Bot.* **3**, 1081-1094 (2012).
3. Cui, Y. et al. Disruption of *EARLY LESION LEAF 1*, encoding a cytochrome P450 monooxygenase, induces ROS accumulation and cell death in rice. *Plant J.* **105**, 942-956 (2021).
4. Mittler, R. & Lam, E. In Situ Detection of nDNA Fragmentation during the Differentiation of Tracheary Elements in Higher Plants. *Plant Physiol.* **2**, 489-493 (1995).
5. Wang, H., Li, J., Bostock, R. M. & Gilchrist, D. G. Apoptosis: A Functional Paradigm for Programmed Plant Cell Death Induced by a Host-Selective Phytotoxin and Invoked during Development. *Plant Cell.* **3**, 375-391 (1996).
6. Liu, S. H., Fu, B. Y., Xu, H. X., Zhu, L. H. & Zhai, H. Q. Cell death in response to osmotic and salt stresses in two rice (*Oryza sativa* L.) ecotypes. *Plant Science*, **5**, 897-902 (2007).

Reviewer #1 (Remarks to the Author):

I carefully reviewed the photos in Supplementary Fig. 1b submitted by the author to demonstrate ferroptotic cell death around 6 hpi after OsiSh-2 inoculation. Comparison of the newly provided transmitted light image with the TUNEL green signals, it is believed that the dead nuclear green signals caused by ferroptotic cell death claimed by the author have been caused due to the experimental method. Looking at Figure 2B in the paper by Jones et al (2016), when a rice leaf cross section (white dotted line) cuts a cell, a passage is created where the plasma membrane is removed. Therefore, it was already proven that PI easily move into cells and stain the nucleus (white stars) (Jones et al., 2016). Therefore, in the photo of suppl. Fig. 1b +E, a large number of TUNEL green signals appeared around 6 hpt after OsiSh-2 inoculation, which is the result of staining when nuclei remain in the cell portion after cross section. In other words, cells with a positive TUNEL green signal were alive before cross section, so cell death was not observed at 6 hpi through Tryphan blue staining. As evidence, if the author enlarges the TUNEL photo of suppl. Fig. 1b -E, the green signals are weak, but they are clearly visible that they are distributed in a similar pattern to the +E photo just below. The brightness of all suppl. Fig. 1b -E photos is a bit low, but if the author raises it to the level of +E photos or more, the TUNEL green signals will become clearer. PI staining photos have too much background overall, making them difficult to use for data analysis. Considering the results submitted by the author so far, it seems clear that ferroptosis-mediated disease resistance induction is not logically correct.

Jones K et al. 2016. Live-cell fluorescence imaging to investigate the dynamics of plant cell death during infection by the rice blast fungus *Magnaporthe oryzae*. *BMC Plant Biol.* 16:69. doi: 10.1186/s12870-016-0756-x.

Response to Reviewer

To Reviewer #1:

Comment: I carefully reviewed the photos in Supplementary Fig. 1b submitted by the author to demonstrate ferroptotic cell death around 6 hpi after OsiSh-2 inoculation. Comparison of the newly provided transmitted light image with the TUNEL green signals, it is believed that the dead nuclear green signals caused by ferroptotic cell death claimed by the author have been caused due to the experimental method. Looking at Figure 2B in the paper by Jones et al (2016), when a rice leaf cross section (white dotted line) cuts a cell, a passage is created where the plasma membrane is removed. Therefore, it was already proven that PI easily move into cells and stain the nucleus (white stars) (Jones et al., 2016). Therefore, in the photo of suppl. Fig. 1b +E, a large number of TUNEL green signals appeared around 6 hpt after OsiSh-2 inoculation, which is the result of staining when nuclei remain in the cell portion after cross section. In other words, cells with a positive TUNEL green signal were alive before cross section, so cell death was not observed at 6 hpi through Tryphan blue staining. As evidence, if the author enlarges the TUNEL photo of suppl. Fig. 1b -E, the green signals are weak, but they are clearly visible that they are distributed in a similar pattern to the +E photo just below. The brightness of all suppl. Fig. 1b -E photos is a bit low, but if the author raises it to the level of +E photos or more, the TUNEL green signals will become clearer. PI staining photos have too much background overall, making them difficult to use for data analysis. Considering the results submitted by the author so far, it seems clear that ferroptosis-mediated disease resistance induction is not logically correct.

Jones K et al. 2016. Live-cell fluorescence imaging to investigate the dynamics of plant cell death during infection by the rice blast fungus *Magnaporthe oryzae*. *BMC Plant Biol.* 16:69. doi: 10.1186/s12870-016-0756-x.

Response: PI (propidium iodide) and TUNEL (terminal deoxynucleotidyl transferase-mediated dUTP-biotin nick-end labelling) staining techniques used in our experiments

are widely recognized detection techniques. PI was used for the detection of cell death and the TUNEL assay can sensitively detect DNA fragmentation caused by cell death signaling cascades (Ref. 67). So, to make sure the dead cells, we observed the red PI signals but not as reviewer indicated by “the dead nuclear green signals”. Although we have explained the working principle of the staining assay in the second and third round of reviews, unfortunately, reviewer misunderstands this detection technique and confused between the two different staining results.

First of all, we hardly agree with the reviewer’s comment that “PI staining photos have too much background overall, making them difficult to use for data analysis.”, although we agreed the reviewer that “PI easily move into cells and stain the nucleus”. The background of PI staining photos in this work was similar with those in the referred papers (Ref. 67; Bureau et al., *Plant Methods*. 2018). Most importantly, in the supplementary figure 1b, E+ showed obvious cell death (red signals) in stomata where the ferroptosis occurred, while E- did not have. Besides, no matter E- or E+ samples, the PI red signals could be observed around the xylem vessel element (Xv), which comprises dead but functioning cells, indicated that PI staining of dead cells was credible.

We didn’t deny the possibility that experiment method (cross section) might induce DNA damage, so it is normal as reviewer mentioned that “in the photo of suppl. Fig. 1b +E, a large number of TUNEL green signals appeared around 6 hpt after OsiSh-2 inoculation, which is the result of staining when nuclei remain in the cell portion after cross section”. But TUNEL green signals didn’t represented the dead cells. More importantly, compared with those of E-, the fluorescence intensity and density in E+ were much stronger, especially around the stomata, indicating that OsiSh-2 induced more DNA damage at 6 hpt. The similar pattern could be found in the previous reports (Re. 67 ; Qiu et al., *New Phytologist*. 2019; Wu et al., *Plant Physiology*. 2016). So, we cannot agree the reviewer mentioned “it is believed that the dead nuclear green signals caused by ferroptotic cell death claimed by the author have been caused due to the experimental method.”. In regard of the light intensity, the fluorescence photos of rice leaves in different treatments were taken under the same light intensity, and we did not

decrease the light intensity or change the exposure times. This is the most basic principle that we should follow in the experiment, and we have emphasized in the second round of revise. Moreover, the intensity of the Merge picture is sufficient to indicate that the experiments are at same light intensity.

Reference:

Bureau, C. et al. A protocol combining multiphoton microscopy and propidium iodide for deep 3D root meristem imaging in rice: application for the screening and identification of tissue-specific enhancer trap lines. *Plant Methods*. **14**, 96 (2018).

Cui, Y. et al. Disruption of *EARLY LESION LEAF 1*, encoding a cytochrome P450 monooxygenase, induces ROS accumulation and cell death in rice. *The Plant Journal*. **105**, 942-956 (2021).
Reference 67 in manuscript.

Koroleva, O. A. et al. Glucosinolate-accumulating S-cells in Arabidopsis leaves and flower stalks undergo programmed cell death at early stages of differentiation. *The Plant Journal*. **64**, 456–469 (2010).

Qiu, Z.N. et al. DNA damage and reactive oxygen species cause cell death in the rice local lesions 1 mutant under high light and high temperature. *New Phytologist*. **222**, 349–365 (2019).

Wu, L.W. et al. Down-Regulation of a Nicotinate Phosphoribosyl transferase Gene, OsNaPRT1, Leads to Withered Leaf Tips. *Plant Physiol*. **171**, 1085-1098 (2016).